# Contribution of leukocyte telomere length to cardiovascular disease onset from genome-wide cross-trait analysis

Jun Qiao [1,2,3,19], Qian Wang[4,19], Yuhui Zhao[5,19], Minjing Chang[6,19], Shuo Sun[7,19], Pengwei Zhang [1,2,3], Kaixin Yao[8], Miaoran Chen[8], Leilei Zheng[9], Xiaolong Xing[5], Liuyang Cai[1,2,3], Anil G. Jegga [10,11,12], Lei Jiang[7] ✉, Siim Pauklin [13] ✉, Rongjun Zou[14,15,16] ✉, Yining Yang[17,18] ✉ & Yuliang Feng [1,2,3] ✉

Telomere shortening is a well-established marker of cellular aging and genomic instability. While the relationship between leukocyte telomere length and cardiovascular diseases has long been of interest, their genetic interplay remains incompletely understood. In this study, we observe substantial genetic overlap beyond genome-wide correlations and identify a potential causal relationship between leukocyte telomere length and coronary artery disease. Specifically, we discover 248 pleiotropic loci, 22 of which show strong evidence of colocalization. Some shared loci implicate multiple pleiotropic genes across different trait pairs, including *ALDH2*, *ACAD10*, *TMEM116*, *SH2B3* (all at 12q24.12), *TMED6* (16q22.1), *SERPINF1* (17p13.3), and *XPO7* (8p21.3). Functional analysis highlights key pathways involved in DNA biosynthesis and telomere maintenance. Notably, SH2B3 is validated through proteome-wide Mendelian randomization analysis, suggesting its potential as a therapeutic target. Here we report the shared genetic basis between leukocyte telomere length and cardiovascular diseases, providing valuable insights into future therapeutic developments.

Telomeres are repetitive DNA-protein complexes that cap the ends of linear chromosomes, safeguarding genomic stability by preventing degradation and fusion[1]. Due to the end-replication problem, telomeres progressively shorten with each cell division, ultimately triggering cellular senescence or apoptosis once a critical length is reached. Telomere attrition is a hallmark of biological aging and has been linked to several age-related diseases, including cardiovascular diseases (CVDs)[2,3]. In the vascular system, accelerated telomere shortening promotes the senescence of smooth muscle cells and macrophages, contributing to the formation of atherosclerotic plaques[4]. These plaques, composed of a necrotic core surrounded by a fibrous cap made up of extracellular matrix, become unstable when senescent cells drive chronic inflammation and degradation of extracellular matrix, leading to thrombosis upon plaque rupture and

elevating the risk of serious cardiovascular events[5,6]. Endothelial dysfunction—an early feature of atherosclerosis—can further accelerate telomere erosion, establishing a vicious cycle of cellular aging and vascular damage[7]. Despite growing interest, the association between leukocyte telomere length (LTL) and CVD risk remains unclear. While some studies report that shorter LTL is linked to a higher risk of coronary artery disease (CAD)[8,9] and peripheral artery disease (PAD)[10], others find no association, particularly in stroke or PAD outcomes[11,12]. These discrepancies highlight the need for further research to clarify the underlying relationship between telomere biology and CVD pathogenesis.

A plausible explanation for the observed association between LTL and CVDs is a shared genetic architecture. LTL exhibits considerable inter-individual variability across the human lifespan but is

A full list of affiliations appears at the end of the paper. ✉e-mail: jianglei0731@gmail.com; siim.pauklin@ndorms.ox.ac.uk; zourj3@mail2.sysu.edu.cn; yangyn5126@xjrmyy.com; fengyl@sustech.edu.cn

highly heritable, with estimates ranging from 44 to 86%[13,14]. The largest genome-wide association study (GWAS) of LTL, based on UK Biobank data, has identified 138 associated loci[15]. Similarly, GWAS of various CVD phenotypes have revealed numerous risk loci, underscoring the contribution of common genetic variants to disease susceptibility[16–21]. These findings offer an opportunity to investigate the genetic overlap between LTL and CVDs. Shared genetic architecture may be mediated by vertical or horizontal pleiotropy. Vertical pleiotropy refers to genetic variants influencing one trait that causally affects another—a framework often examined using Mendelian randomization (MR). Previous MR studies have suggested potential causal associations between shorter LTL and CAD or stroke, whereas findings for atrial fibrillation (AF) and heart failure (HF) remain inconsistent[9,15,22–24]. In contrast, horizontal pleiotropy describes genetic variants that independently affect multiple traits, pointing to shared genes and biological pathways. Recent work by Gong et al. has demonstrated that horizontal pleiotropy plays a critical role in explaining the genetic architecture of complex traits[25]. However, studies of horizontal pleiotropy between LTL and CVDs remain limited, with most efforts focused on psychiatric disorders[26]. Given the extensive epidemiological evidence and emerging insights from genetic studies, this study aims to systematically evaluate the shared genetic architecture and underlying mechanisms between LTL and multiple CVD phenotypes through a genome-wide pleiotropic analysis.

Here, we show a thorough analysis of shared genetic architectures between LTL and CVDs by encompassing the available large GWAS datasets in individuals of European ancestry. First, by charting the landscape of genetic overlap beyond genetic correlation, we provide more granular insights into the unique and shared genetic architectures between LTL and CVDs. Then, we employ innovative statistical techniques to capture diverse forms of genetic pleiotropy, followed by thorough analyses to link the genomic findings to biological pathways, yielding profound implications for conceptualizing shared genetic risk for both LTL and CVDs.

## Results

### Genetic overlap beyond genetic correlation between LTL and six major CVDs

We obtained GWAS summary statistics for LTL and six major CVDs (Supplementary Data 1 provides an overview and acronyms; the Methods section list data sources; Fig. 1 illustrates the general workflow). Following the harmonization and filtering of SNPs shared across GWAS summary statistics, we employed cross-trait linkage disequilibrium (LD) score regression (LDSC)[27] to assess genome-wide genetic correlation ($r_g$) between LTL and six major CVDs, namely AF, CAD, venous thromboembolism (VTE), HF, PAD, and stroke. The results of bivariate LDSC indicated a range of weak to moderate genome-wide $r_g$ between LTL and CVDs, excluding LTL-AF. Notably, the most pronounced negative $r_g$ were observed for LTL-PAD ($r_g = -0.250$, SE = 0.040, $P = 3.83 \times 10^{-10}$) and CAD ($r_g = -0.171$, SE = 0.025, $P = 4.65 \times 10^{-12}$), while smaller but statistically significant $r_g$ were noted for stroke ($r_g = -0.104$, SE = 0.037, $P = 4.60 \times 10^{-3}$) and VTE ($r_g = -0.072$, SE = 0.025, $P = 4.20 \times 10^{-3}$). LTL was only moderately genetically correlated to HF ($r_g = -0.145$, SE = 0.037, $P = 8.20 \times 10^{-5}$). In contrast, no significant $r_g$ was observed between LTL and AF (Supplementary Fig. 2 and Supplementary Data 2). Although genome-wide $r_g$ offered valuable insight into the genetic overlap between phenotypes, it could not distinguish genetic overlap resulting from a mixture of concordant and discordant effects from the absence of genetic overlap, potentially yielding an estimated $r_g$ near zero in both scenarios. Therefore, using multiple methods with different model assumptions to identify and understand this "missing dimension" of genetic overlap was essential for comprehensively characterizing the shared genetic foundations across phenotypes.

To better investigate polygenic overlap beyond genetic correlation, we used the causal mixture modeling approach (MiXeR) to estimate the number of shared and unique trait-influencing variants (or 'causal' variant, defined as genetic variants with non-zero additive effects) necessary to explain at least 90% of heritability for each trait[28]. Unlike methods that rely on genome-wide genetic correlation, MiXeR assesses overlap without considering the direction of each variant's effect. This approach provides a clearer picture of localized correlations, which might otherwise be masked in genome-wide analyses due to opposing directional effects of shared variants that cancel each other out. Univariate MiXeR revealed that LTL exhibited a lower degree of polygenicity ($N = 380$, SD = 26). Among the six major CVDs, HF ($N = 2305$ 'causal' variants explaining 90% of HF's $h^2_{SNP}$, SD = 213) was the most polygenic, followed by CAD ($N = 1528$, SD = 311) and stroke ($N = 1055$, SD = 117). VTE, PAD, and AF demonstrated lower polygenicity, associated with 308 to 504 variants at 90% $h^2_{SNP}$ (Supplementary Data 3a). These findings highlighted a pattern of polygenicity distinct from $h^2_{SNP}$ estimates.

The result of bivariate MiXeR revealed substantial but distinct patterns of polygenic overlap between LTL and CVDs. Given the low polygenicity of LTL and these CVD phenotypes (including AF, VTE, and PAD), LTL was found to share less proportion of causal variants with these CVDs, ranging from 18.30% in AF to 22.63% in PAD. However, relatively large genetic overlaps were also observed between LTL and these CVDs (Fig. 2a, Supplementary Fig. 3 and Supplementary Data 3b). For example, polygenic overlap between LTL and PAD was particularly striking (Dice coefficient = 0.229, SD = 0.015), with 86 (SD = 7) shared variants, representing 22.63% LTL-influencing variants and 23.24% PAD-influencing variants, consistent with the strongest negative genome-wide genetic correlation ($r_g = -0.223$, SE = 0.014) and genetic correlation of shared variants (the genetic correlation of shared variants [$r_gs$] = −0.970, SE = 0.022). A total of 69 (SD = 19) variants were estimated to be shared between LTL and AF, representing 18.30% LTL-influencing variants and 13.79% AF-influencing variants (Dice coefficient = 0.157, SD = 0.041), despite weak negative genetic correlation. This pattern of extensive genetic overlap but weak $r_g$ indicated a predominance of mixed effect directions, supported by the MiXeR-estimated proportion of shared 'causal' variants with concordant effects (0.416, SD = 0.029). Considering the low polygenicity of LTL and high polygenic diseases such as CAD, HF, and stroke, significant disparities were observed in the number of shared and unique "causal" variants. In particular, MiXeR estimated that of the 380 LTL-influencing variants, 48.69%, 39.82%, and 28.07% also influence HF, CAD, and stroke, respectively. For example, LTL and HF shared the largest number of variants ($N = 185$, SD = 24), with many more unique variants of HF ($N = 2120$, SD = 208) than unique variants of LTL (195, SD = 30), representing 48.69% LTL-influencing variants and 8.02% HF-influencing variants. While they were moderately correlated at the genome-wide level ($r_g = -0.180$, SE = 0.017), shared variants were strongly correlated ($r_gs = -0.913$, SE = 0.087). A similar, although less pronounced, relationship was evident in LTL-CAD and LTL-stroke. Furthermore, the overlapping variants demonstrated a low level of effect direction concordance, highlighting the prevalence of mixed effect directions between LTL and CVDs. These observations suggested that the extent of polygenic overlap between LTL and CVDs was likely underestimated by genome-wide genetic correlations (Fig. 2b).

Local genetic correlations provide a more effective means of capturing genetic associations with mixed effect directions. Specifically, a pair of traits may display no genome-wide $r_g$ due to an equal number of positive and negative (opposite effect directions) local genetic correlations with comparable magnitudes. To prevent the potential masking of local genetic correlations when evaluating $r_g$ at the genome-wide level, we applied Local Analysis of [co]Variant Annotation (LAVA)[29] to perform local genetic correlations (loc-$r_gs$) between LTL and six major CVDs at genomic regions, where both

**Fig. 1 | Schematic representation of analyses performed for leukocyte telomere length and six major cardiovascular diseases in the current study.** This figure demonstrates the comprehensive pleiotropic analysis conducted for LTL and six major CVDs from multiple perspectives within this study. We first investigated the shared genetic architectures between LTL and six major CVDs by assessing pairwise genetic overlap beyond correlation. Extensive analyses were then conducted to investigate two types of pleiotropy: horizontal pleiotropy, whereby causal variants for two traits colocalize in the same locus, and vertical pleiotropy, whereby a variant exerts an effect on one trait through another. Notably, spurious pleiotropy was excluded from the analysis, whereby causal variants for two traits fall into distinct loci but are in LD with a variant associated with both traits. Therefore, we applied Mendelian randomization to evaluate the pairwise causal associations between LTL and the CVDs, primarily elucidating the contributions from vertical pleiotropy. We used several statistical tools to capture horizontal pleiotropy by characterizing the shared loci and their implications on genes, tissues, biological functions, and protein targets. This comprehensive pleiotropic analysis allowed us to construct an atlas of the shared genetic associations, enhancing our understanding of the complex interactions between LTL and cardiovascular health. The diagram was created using BioRender and included with permission for publication (Created in BioRender. Feng, Y. (2025) https://BioRender.com/gwhp1nw). LTL leukocyte telomere length, AF atrial fibrillation, CAD coronary artery disease, VTE venous thromboembolism, HF heart failure, PAD peripheral artery disease.

phenotypes had heritability estimates significantly different from zero (Supplementary Data 4). Overall, 54 local genomic regions were found significant for bivariate analysis after correcting for multiple testing using false discovery rate (FDR) (FDR < 0.05; Fig. 2a and Supplementary Data 5), with 61% and 39% of the regions showing negative and positive loc-$r_g$s, respectively. Corroborating the MiXeR findings, LAVA estimated correlated loci of LTL-PAD (2 positively correlated and 5 negatively correlated regions), LTL-VTE (5 positively correlated and 5 negatively correlated regions), and LTL-AF (9 positively correlated and 4 negatively correlated regions), adding further support for a shared genetic basis. Local correlations for LTL-CAD, comprising 14 negatively and 3 positively correlated regions, and for LTL-stroke, with 5 negatively correlated regions and 2 positively correlated, were consistent with their respective negative global genetic correlations. The absence of significant genomic regions between LTL and HF may be attributed to LAVA's propensity for identifying loci with extreme correlations, in contrast to MiXeR, thereby highlighting loci more likely to be statistically significant.

Interestingly, our investigation also identified LD block 1841 (chr12: 111,592,382–113,947,983) displaying significant correlations for

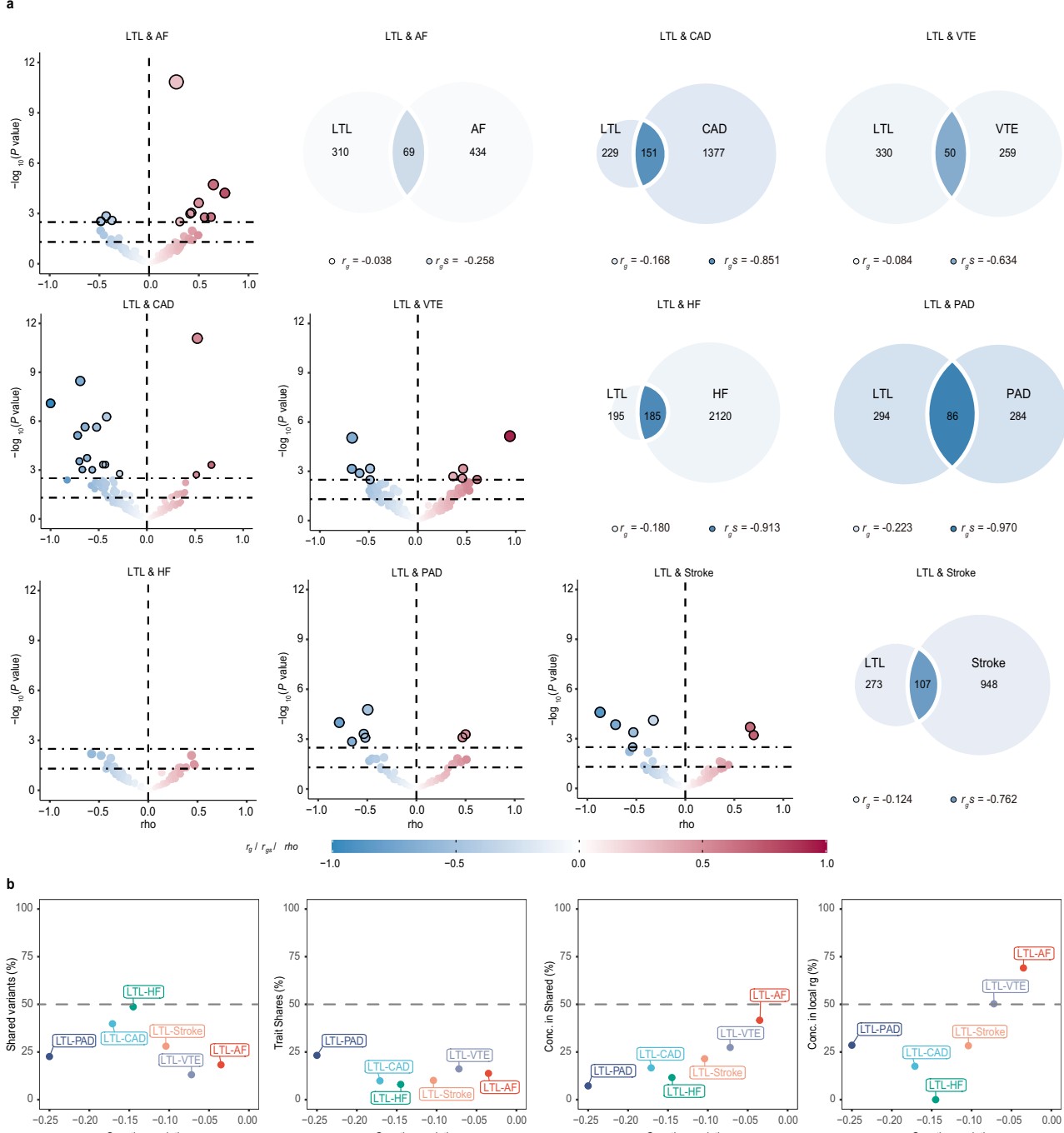

**Fig. 2 | Genetic overlap between leukocyte telomere length and six major cardiovascular diseases beyond genome-wide genetic correlation. a** MiXeR-modeled genome-wide genetic overlap and genetic correlations (top right) and LAVA local correlations (bottom left) between LTL and six major CVDs. Top right: MiXeR Venn diagrams showing the number of estimated 'causal' variants that are unique to LTL, unique to CVDs or shared between them. Genome-wide genetic correlation ($r_g$) and genetic correlation of shared variants ($r_gs$) are represented by the color of the trait-specific ($r_g$) and shared regions ($r_gs$), respectively. The circle size represents the extent of polygenicity of each trait, with larger circles corresponding to greater polygenicity and vice versa. Bottom left: Volcano plots of LAVA local genetic correlation coefficients (rho, *x*-axis) against -log10 (*P* values) for each pairwise analysis per locus. Larger dots with black circles represent significantly correlated loci after FDR correction (FDR < 0.05). MiXeR estimated $r_g$ and $r_gs$, and

LAVA estimated rho are represented on the same blue to red color scale. All statistical tests were two-sided unless stated otherwise. Note that the volcano plots were plotted at *P* values truncated by $1 \times 10^{-12}$ for better visualization, thus excluding a region (LD block 1581 on chromosome 10, ranges from 104,206,838 to 106,142,283) influencing LTL–CAD from volcano plots of LAVA. **b** Genetic correlation estimated by LDSC (*x*-axis) against the percentage of LTL variants that are shared with CVDs as estimated by MiXeR (first plot), the percentage of CVD variants that are shared with LTL (second plot), and the percentage of CVD variants that are shared with LTL that have concordant effect directions (third plot). The fourth plot shows the percentage of local genetic correlations from LAVA with concordant effect directions on the *y*-axis. LTL leukocyte telomere length, AF atrial fibrillation, CAD coronary artery disease, VTE venous thromboembolism, HF heart failure, PAD peripheral artery disease.

a majority of the trait pairs, with uniform negative correlation values between −0.692 and −0.530. Subsequent analyses utilizing Hypothesis Prioritisation in Multi-trait Colocalization (HyPrColoc)[30] revealed robust colocalization evidence for this locus between LTL and all CVDs, excluding AF, HF and PAD, with a posterior probability (PP) higher than 0.7, which encompassed the shared causal SNP (rs10774625, an intronic variant of the ataxin 2 (*ATXN2*) gene on 12q24.12) (Supplementary Data 19). The *ATXN2* rs10774625 polymorphism has been associated with various CVDs, notably CAD, alongside cardiometabolic markers such as blood pressure and blood lipids[31].

## The causal inference between LTL and six major CVDs

Despite these findings substantiating a shared genetic foundation between LTL and six major CVDs, there was uncertainty in relation to whether the complex interplay predominantly reflected horizontal pleiotropy or potentially involved a causal relationship (referred to as 'vertical pleiotropy'). Mendelian randomization (MR) harnesses vertical pleiotropy to deduce potential causal relationships, excluding any SNPs indicative of horizontal pleiotropy. Therefore, latent causal variable (LCV) analysis was utilized to elucidate the possible causal relationships underlying the genetic correlations observed[32]. Notably, none of the trait pairs demonstrated tendencies indicative of partial genetic causation under the stringent genetic causality proportion (GCP) threshold ($|GCP| > 0.6$, $P < 0.05$ / number of trait pairs = $8.33 \times 10^{-3}$) (Supplementary Fig. 4a and Supplementary Data 6a). Under a more lenient threshold ($|GCP| > 0.4$, $P < 0.05$), we detected a weak negative causal association between LTL and CAD, suggesting that genetically inferred LTL shortening was linked to a higher risk of CAD.

While LCV estimates causal relationships between traits by leveraging shared genetic architecture, it does not fully account for latent heritable confounders, which can introduce bias into causal estimates. To address this limitation, we conducted an additional analysis using the MRlap method to explore the bidirectional causal associations between LTL and CVDs[33]. The MRlap analysis confirmed the previously identified negative causal association between LTL and CAD (OR = 0.926, 95% CI: 0.899-0.955; Supplementary Fig. 4b and Supplementary Data 6b), which aligns with findings from traditional MR analysis (Supplementary Data 6c–e). Overall, our MR results suggest that LTL has only a weak negative causal impact on CAD, indicating that vertical pleiotropy contributes minimally to the shared genetic architecture between LTL and CVDs.

## Pleiotropic genomic loci identified for LTL and CVDs

The observed comorbidity between LTL and six major CVDs suggests that instead of a predominance of trait-specific risk variants, there may be a set of pleiotropic variants influencing the risk of both LTL and CVDs (i.e., horizontal pleiotropy). To identify pleiotropic SNPs within the union set of trait pairs exhibiting significant genetic correlation or overlap, we applied pleiotropic analysis under composite null hypothesis (PLACO) method[34]. PLACO pinpointed potential pleiotropic variants shared between LTL and CVDs, resulting in the identification of 12,604 SNPs comprising 10,008 unique variants. Functional Mapping and Annotation (FUMA)[35] further delineated 248 independent genomic risk loci as pleiotropic, spanning 122 unique chromosomal regions (Fig. 3a–f, Supplementary Fig. 5 and Supplementary Data 7–8). Among these, 194 loci were associated with LTL and 80 loci with CVDs. In total, 32 loci overlapped between LTL and CVDs, accounting for 16.49 and 40.00% of the total number of loci linked to these respective categories. A total of 188 pleiotropic loci exhibited genetic signals for multiple trait pairs, with 74 of them (39.36%) spanning 16 unique chromosomal regions, demonstrating this phenomenon in over half of the investigated trait pairs. For example, the pleiotropic locus 16q22.1 (mapped gene: *TMED6*) was jointly associated with LTL and all CVDs. A mixture of concordant and discordant allelic effects existed in these pleiotropic loci. Remarkably, the selected effect alleles at the top SNPs within or near 117 loci (47.18%) exhibited inconsistent effects on two traits within a pair. Essentially, these variants may concurrently increase LTL and diminish the risk of developing CVDs, aligning with their robust genome-wide genetic correlation. Conversely, the remaining SNPs demonstrated concordant associations with both LTL and CVDs, implying that these SNPs might influence the risk of both traits in the same direction.

ANNOVAR was used to identify the functional categories of the lead SNPs shared between LTL and CVDs based on their locations with respect to genes, including exonic, intronic, 5′ untranslated regions, 3′ untranslated regions, upstream, downstream, and intergenic[36]. ANNOVAR category annotation of candidate SNPs shared between LTL and CVDs revealed that 67 (27.02%) were in intergenic regions, 132 (53.23%) were in intronic regions, and only 18 (7.26%) were in exonic regions. For example, the index SNP rs1566452 at 16q22.1 locus ($P_{PLACO} = 4.46 \times 10^{-8}$ for LTL-HF) was associated with expression quantitative trait loci (eQTL) in the artery coronary and artery tibial ($P_{Artery\_Coronary} = 4.06 \times 10^{-5}$, $P_{Artery\_Tibial} = 3.32 \times 10^{-10}$; Supplementary Data 9–10) for WW domain-containing E3 ubiquitin protein ligase 2 (*WWP2*) gene encoding one of the E3 ubiquitin ligases, which critically participate in the development and progression of cardiovascular diseases[37]. Besides, numerous ubiquitin E3 ligases have also been documented to promote the degradation of human telomerase reverse transcriptase (hTERT), thereby reducing telomerase activity and potentially leading to decreased telomere length[38]. Furthermore, we identified 21 top SNPs with combined annotation-dependent deletion (CADD) scores[39] exceeding 12.37, and 7 mRNA exonic variants with even higher CADD scores, indicating potentially deleterious effects. Notably, rs11556924 within the zinc finger C3HC-type containing 1 (*ZC3HC1*) gene represents an exonic non-synonymous variant with a CADD score of 28 (variants with scores above 20 are predicted to be among the 1.0% most deleterious substitutions in the human genome). Furthermore, nine SNPs were assigned RegulomeDB scores of 1 f, 1 d, or 1a[40], indicating a likely influence on binding sites. We followed up this finding using the GTEx database to investigate the gene regulatory effects. For example, rs11779558 at 8p21.3 locus was significantly associated with eQTL functionality in artery tibial ($P_{Artery\_Tibial} = 5.79 \times 10^{-5}$) for exportin 7 (*XPO7*) gene.

Colocalization analysis[41] further revealed 22 out of 248 potential pleiotropic loci with PP.H4 greater than 0.7, wherein 14 top SNPs at the corresponding loci were identified as candidate shared causal variants (Fig. 3a–f, Supplementary Fig. 6 and Supplementary Data 7). Notably, the 12q24.12 locus, pleiotropic for all correlated trait pairs except for LTL-AF, exhibited evidence of colocalization between these trait pairs (PP.H4 ranging from 0.729 to 0.998). We further performed multi-trait colocalization analysis on each FUMA-annotated pleiotropic locus using the HyPrColoc method to identify potential shared causal variants and provide the posterior probability of colocalization. HyPrColoc analysis revealed colocalization evidence for this locus between LTL and all CVDs except AF and PAD, with a PP exceeding 0.7, identifying rs10774625 (an intronic variant of the *ATXN2* gene on 12q24.12) as the potential shared causal variant (Supplementary Data 19). Moreover, 46 pleiotropic loci were identified with PP.H3 exceeding 0.7, indicating the possibility of different causal variants within these loci.

## Pleiotropic genes associated with LTL and multiple CVDs

Despite the success of the above analyses in identifying disease risk loci, the biological significance of most identified variants remains unknown. To achieve a more comprehensive understanding of how genetic variation influences disease risk, we adopted approaches integrating SNPs across a spectrum of association significance to construct a cohort of predicted genes that could subsequently be

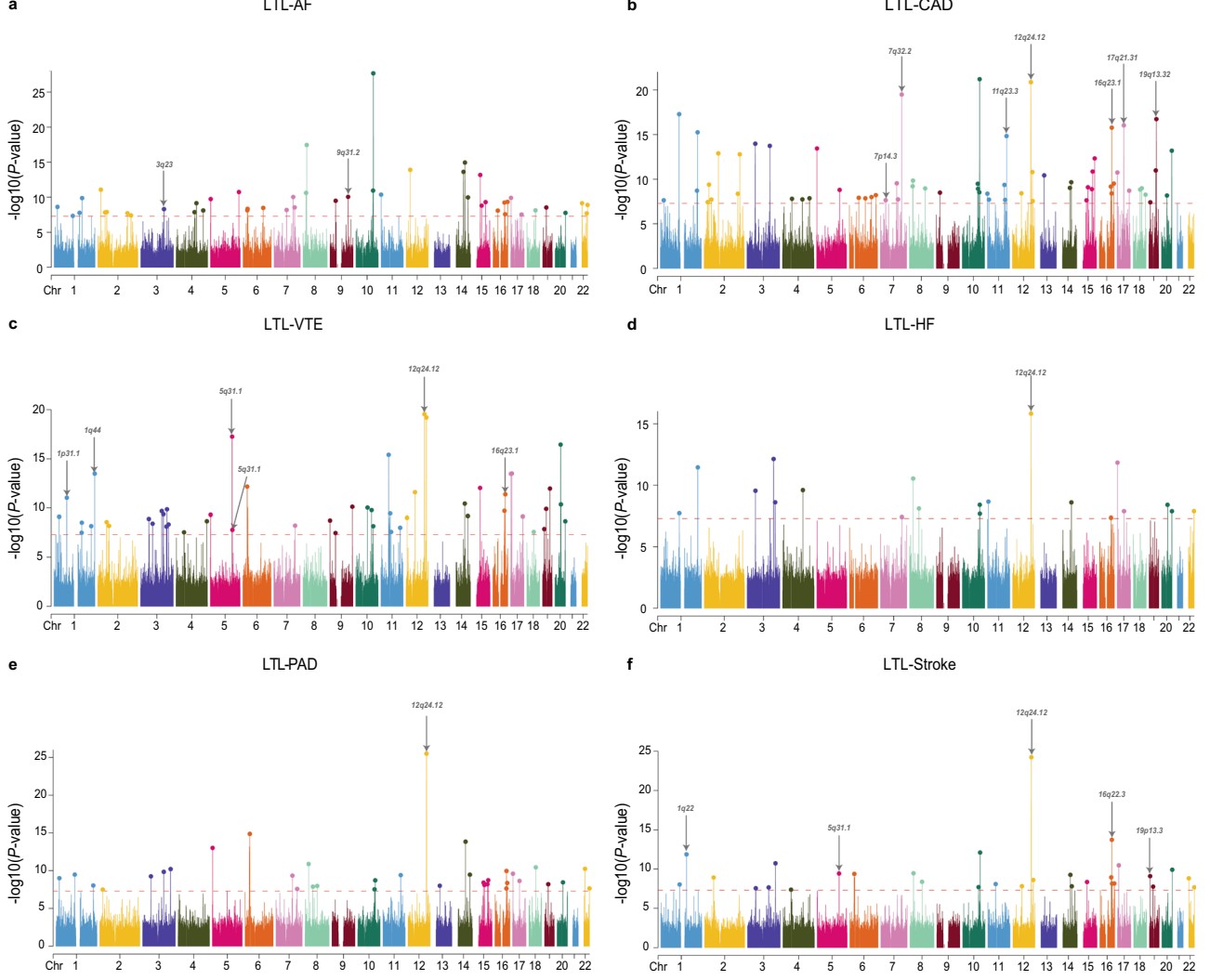

**Fig. 3 | Manhattan plots for the PLACO results of leukocyte telomere length and six major cardiovascular diseases.** Manhattan plots show the PLACO results for LTL-AF (**a**), LTL-CAD (**b**), LTL-VTE (**c**), LTL-HF (**d**), LTL-PAD (**e**), and LTL-Stroke (**f**). The *x*-axis reflects the chromosomal position, and the *y*-axis reflects negative log10 transformed *P* values for each SNP. The horizontal dashed red line indicates the genome-wide significant *P* value of -log10 ($5 \times 10^{-8}$). The independent genome-wide significant associations with the smallest *P* value (Top lead SNP) are encircled in a colorful circle. Only SNPs shared across all summary statistics were included. Labels are the chromosome regions where genomic risk loci with evidence for colocalization (PP.H4 > 0.7) are located. All statistical tests were two-sided. LTL leukocyte telomere length, AF atrial fibrillation, CAD coronary artery disease, VTE venous thromboembolism, HF heart failure, PAD peripheral artery disease.

mapped to functional pathways for analysis. We employed two distinct strategies for mapping SNPs to genes: Firstly, a genome-wide gene-based association study in Multi-marker Analysis of GenoMic Annotation (MAGMA)[42] and positional mapping in FUMA were utilized, mapping SNPs to genes based on their physical position in the genome. Secondly, eQTL-informed MAGMA (e-MAGMA)[43] and eQTL mapping in FUMA were employed to map SNPs to genes through their eQTL associations.

MAGMA analysis, utilizing 557 potential pleiotropic genes located within or overlapping with 248 pleiotropic loci, identified 478 significant pleiotropic genes (323 unique), in which 244 genes were detected in two or more trait pairs (Fig. 4 and Supplementary Data 11–13). For example, *SH3PXD2A*, *SH2B3*, *BRAP*, *ATXN2*, *PTPN11*, *NAA25*, *ALDH2*, and *ACAD10* were identified as significant pleiotropic genes in five pairs of traits. Remarkably, seven of eight genes (excluding *SH3PXD2A*) were located on the 12q24.12 locus, identified in all trait pairs except for LTL-AF. Of the pleiotropic genes identified, 98 (20.50%) were novel for LTL and 258 (53.97%) for CVDs. Only one pleiotropic gene, CD19 molecule (*CD19*), had not previously been reported to be associated with both traits. Furthermore, 470 genes

(98.33%) identified by MAGMA were confirmed by FUMA positional mapping (Supplementary Data 9).

To pinpoint tissues potentially integral to the biological processes of LTL and six major CVDs, we utilized LDSC applied to specifically expressed genes (LDSC-SEG) for tissue-specific enrichment analysis using single-trait GWAS summary statistics[44]. By linking genetic heritability to specific functional categories or tissues, LDSC-SEG evaluates whether the heritability of LTL and CVDs is enriched in SNPs that are active in specific tissues. Regarding multi-tissue gene expression, we found that expressions of LTL-associated loci were significantly enriched in spleen tissues, surpassing an FDR < 0.05 threshold (Fig. 5 and Supplementary Data 14). Additionally, AF showed significant enrichment in heart-related tissues, including heart left ventricle and heart atrial appendage, while CAD demonstrated enrichment in artery-related tissues, such as artery tibial, artery aorta, and artery coronary. Conversely, no significant tissue-specific enrichment was observed for VTE, HF, PAD, and stroke. These findings were corroborated by multi-tissue chromatin interaction results.

Given that MAGMA assigns SNPs to the nearest genes based on arbitrary genomic windows, and considering that the effects of a locus

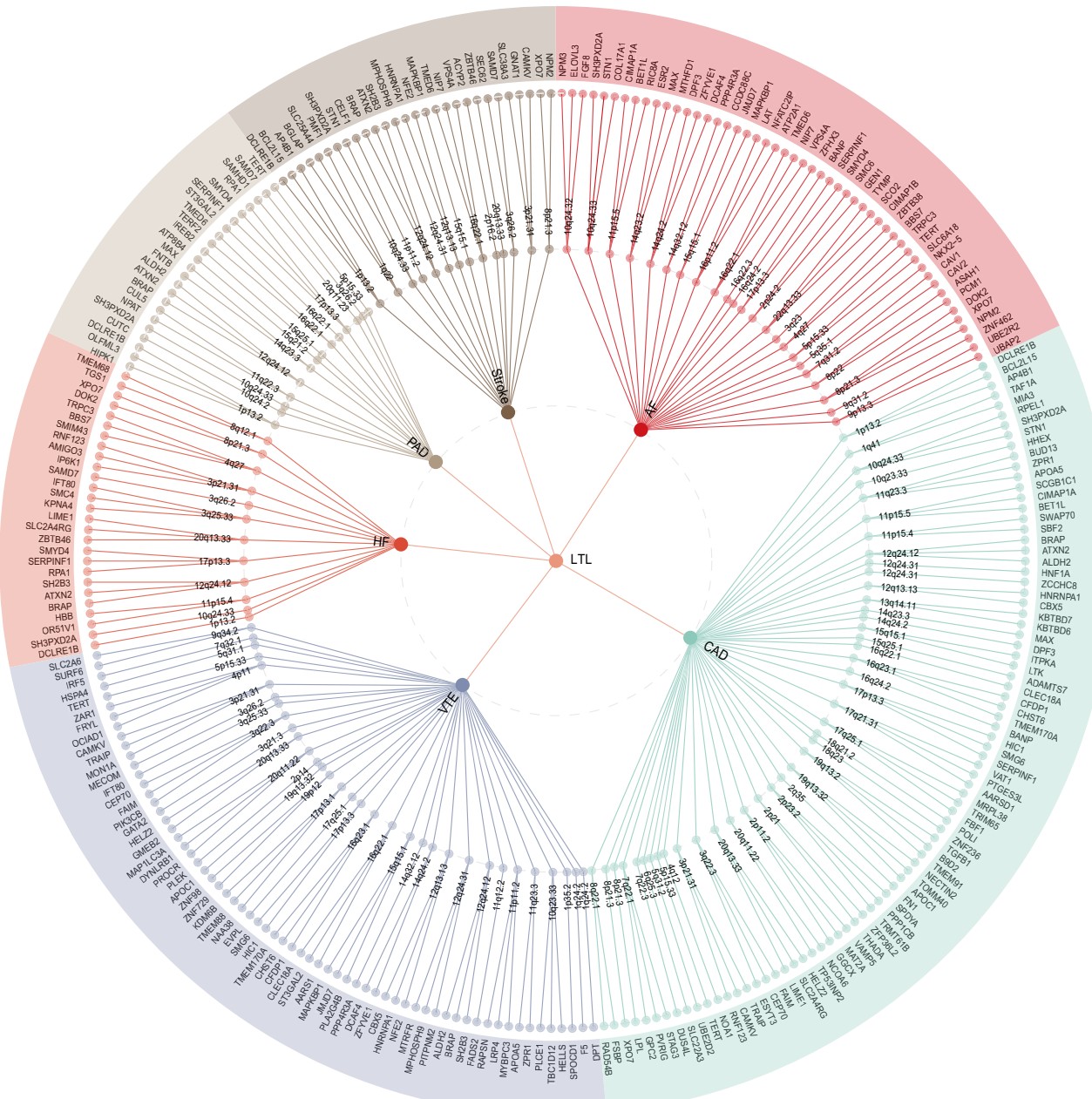

**Fig. 4 | The Overall landscape of the pleiotropic associations across leukocyte telomere length and six msjor cardiovascular diseases.** A circular dendrogram showing the shared genes between LTL (center circle) and each of six CVDs (first circle), resulting in six pairs. A total of 248 shared loci were identified across six trait pairs, mapped to 478 significant pleiotropic genes (323 unique) identified by multimarker analysis of GenoMic annotation (MAGMA). For the trait pairs with more than three pleiotropic genes, we only showed the top 3 pleiotropic genes according to the prioritization of candidate pleiotropic genes (fourth circle). Bonferroni correction was applied, and all reported *P* values were two-sided. LTL leukocyte telomere length, AF atrial fibrillation, CAD coronary artery disease, VTE venous thromboembolism, HF heart failure, PAD peripheral artery disease.

do not always operate through the nearest gene, a critical need remains to functionally link SNPs to genes (e.g., through genetic regulation) to enhance our understanding of potential underlying mechanisms. Consequently, e-MAGMA was conducted to uncover functional gene associations potentially overlooked by the proximity-based SNP assignment in MAGMA, thereby illuminating alternative causal pathways from SNPs to traits. e-MAGMA analysis revealed 1843 significant tissue-specific pleiotropic genes (419 unique) after Bonferroni correction, each strongly enriched in at least one tissue (Supplementary Data 15). Of these, 918 tissue-specific pleiotropic genes (75 unique) were significantly identified across multiple trait-related tissues in at least two trait pairs. For example, *MAPKAPK5, TMEM116,*

*HECTD4, ALDH2,* and *ACAD10,* all located on the 12q24.12 locus, were recognized as significant tissue-specific pleiotropic genes in all trait pairs except for LTL-AF. Notably, transmembrane protein 116 (*TMEM116*) gene was found to be highly tissue-specific, showing enrichment in eight trait-related tissues, including artery tibial, artery coronary, adipose visceral (omentum), adipose subcutaneous, heart atrial appendage, heart left ventricle, liver, and whole blood. *TMEM116* encoded a transmembrane protein involved in blood coagulation and had been identified as a potential risk gene for coronary atherosclerosis in previous studies[45]. However, its relationship with LTL remained less understood. In comparison with the transcriptome-wide association study (TWAS)[46] results for single-trait GWAS, we identified

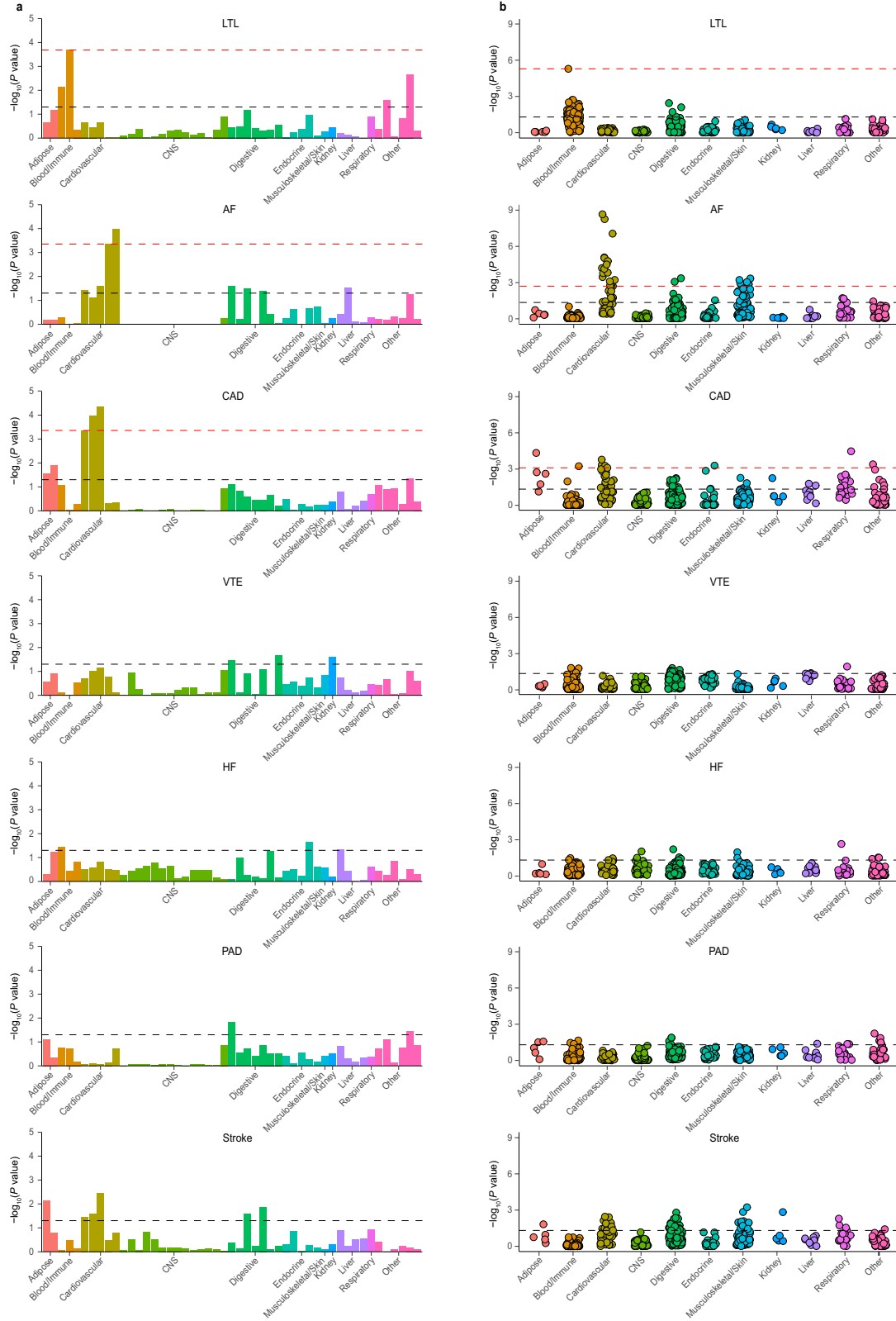

940 (51.00%) tissue-specific pleiotropic genes as novel for LTL and 1446 (78.46%) for CVDs (Supplementary Data 16). Finally, of the 1843 genes identified by e-MAGMA, 83.45% of these genes were replicated by FUMA eQTL mapping (Supplementary Data 9).

Finally, 289 pleiotropic genes (207 unique) were jointly identified by MAGMA and e-MAGMA analysis, in which 50 unique genes were detected in two or more trait pairs, further suggesting the tissue

specificity of these pleiotropic genes (Supplementary Data 11). For example, *ACAD10* (12q24.12), *ALDH2* (12q24.12), *HECTD4* (12q24.12), *MAPKAPK5* (12q24.12), *NAA25* (12q24.12), *SH2B3* (12q24.12), *TMEM116* (12q24.12), *SERPINF1* (17p13.3), *TMED6* (16q22.1), and *XPO7* (8p21.3) were identified as significant pleiotropic genes in more than half of the trait pairs. Remarkably, two of ten genes (including *ALDH2* and *ACAD10*) were located at the 12q24.12 locus, identified in five pairs of

**Fig. 5 | The results of multiple-tissue analysis using gene expression data and chromatin data for leukocyte telomere length and six cardiovascular diseases.** **a** Tissue type-specific enrichment of single nucleotide polymorphism (SNP) heritability for LTL and CVDs in 49 tissues from GTEx v8 estimated using stratified LDSC applied to specifically expressed genes (LDSC-SEG). Each bar represents a tissue from the GTEx dataset. The *x*-axis reflects tissue types, and the *y*-axis reflects negative log10 transformed *P*values. **b** Each point represents a peak for DNase I hypersensitivity site (DHS) or histone marks (including H3K27ac, H3K36me3, H3K4me1, H3K4me3, and H3K9ac) in a tissue type. Tissues types were classified into 11 distinct categories, i.e., 'Adipose,' 'Blood/Immune,' 'Cardiovascular,'' 'CNS,' 'Digestive,' 'Endocrine,' 'Musculoskeletal/Skin,' 'kidney,' 'Liver,' 'Respiratory,' and 'Other.' The black dotted line represents the significance threshold of *P* < 0.05, and the red line indicates the significant *P* value after conducting FDR correction (FDR < 0.05). In both (**a**) and (**b**), different colors represent different tissue types. All statistical tests were two-sided. LTL leukocyte telomere length, AF atrial fibrillation, CAD coronary artery disease, VTE venous thromboembolism, HF heart failure, PAD peripheral artery disease.

traits except for LTL-AF. Acyl-CoA dehydrogenase family member 10 (*ACAD10*), a gene encoding an enzyme crucial for fatty acid beta-oxidation in mitochondria, critically regulates cellular lipid synthesis with significant expression in the human brain[47]. Previous data show that homozygous loss of function of *ACAD10* results in perturbed lipid synthesis that potentially influences the development of CVDs, whereas common gene variants have been associated with CAD, stroke, and hypertension (a common CVD risk factor)[48]. Additionally, aldehyde dehydrogenase 2 family member (*ALDH2*) gene encoding a vital mitochondrial enzyme critical for cardiac function, has been associated with exacerbated myocardial remodeling and contractile dysfunction in aging. This association is possibly through mitochondrial damage mediated by the AMPK/Sirt1 pathway.

### Shared biological pathways between LTL and six major CVDs
To investigate the concept that a group of genes might collectively fulfill specific biological functions through shared pathways or functional category enrichments, we employed various analytical strategies, including gene-set analysis of genes identified via MAGMA and functional enrichment analysis targeting tissue-specific genes. After rigorously adjusting for 7744 gene sets (biological process sets from the Molecular Signatures Database [MSigDB, v.2023.1; C5: GO BP])[49], we noted minimal overlap in gene sets between LTL and six major CVDs. Only three gene sets associated with chromatin organization, the negative regulation of nucleobase-containing compound metabolic processes, and the negative regulation of miRNA maturation were identified. No gene set demonstrated significant enrichment across more than one trait pair (Supplementary Data 17a). Remarkably, many genes shared between LTL and AF were most significantly linked to the 'negative regulation of the nucleobase-containing compound metabolic process.' Subsequently, we identified several biological processes overrepresented among the 50 pleiotropic genes detected by MAGMA and e-MAGMA analysis in two or more trait pairs, using the ToppGene Functional Annotation tool (ToppFun)[50] (Supplementary Fig. 7 and Supplementary Data 17b). All identified significant biological processes, except for the DNA biosynthetic process, were directly associated with telomere maintenance processes such as 'telomere maintenance via telomerase,' 'telomere maintenance via telomere lengthening,' 'telomere capping,' and 'telomere organization.' Interestingly, the nucleobase-containing compound metabolic process identified in the MAGMA gene-set analysis encompassed both DNA biosynthetic processes and telomere maintenance mechanisms, implying a key role in shared biological pathways between LTL and CVDs.

### Shared causal proteins between LTL and six major CVDs
We undertook a comprehensive analysis of the MR associations between 1,922 unique proteins and the risk of LTL and six major CVDs using the summary data-based MR (SMR)[51], each protein with index cis-acting variants (cis-pQTL) obtained from the UK Biobank Pharma Proteomics Project (UKB-PPP). Following the exclusion of associations failing the Heterogeneity in Dependent Instrument (HEIDI) test, and after conducting sensitivity analysis with multi-SNPs-SMR and applying multiple testing corrections via Bonferroni adjustment, the genetically predicted levels of 85 proteins were found to be

significantly associated with the risk of LTL and CVDs (Fig. 6 and Supplementary Data 18). Specifically, 12, 9, 26, 24, 2, 6, and 6 proteins were significantly associated with LTL, AF, CAD, VTE, HF, PAD, and stroke, respectively. Notably, SH2B adaptor protein 3 (SH2B3) emerged as significantly associated with LTL, CAD, and VTE, also showing colocalization evidence in HyPrColoc analysis, with rs10774625 pinpointed as a shared causal variant (Supplementary Data 19). The index SNP rs10774625, located at the 12q24.12 locus (an intronic variant of the *ATXN2* gene), was associated with eQTLs in whole blood ($P_{Whole\_Blood} = 3.48 \times 10^{-4}$) and was also linked to pQTLs in whole blood ($P_{Whole\_Blood} = 1.08 \times 10^{-3}$) for the SH2B3. SH2B3 acted as an adaptor protein, playing a crucial role in negatively regulating cytokine signaling and cell proliferation. Prior research indicated that SH2B3 was associated with lifespan, with missense alleles within SH2B3 influencing life expectancy through predisposition to cardiovascular events[52]. SH2B3 mutations can lead to chronic inflammation and a higher susceptibility to CVDs, underscoring the importance of this protein in cardiovascular health.

## Discussion
In this comprehensive genome-wide pleiotropy association study, we systematically examined the shared genetic architecture between LTL and six major CVDs, extending beyond traditional genome-wide genetic correlations. Our subsequent causal inference analyses further supported a causal relationship between LTL and CAD, particularly from the perspective of vertical pleiotropy. Further in-depth analyses identified pleiotropic genetic variants and loci, pleiotropic genes, biological pathways, and protein targets from a horizontal pleiotropy perspective, all reinforcing the involvement of the DNA biosynthesis and telomere maintenance in the shared genetic etiology of these traits. Overall, these findings provide insights into the relationship and shared genetic mechanisms underlying LTL and six major CVDs.

Consistent with prior robust epidemiological evidence, our findings reveal weak to moderate, yet statistically significant, negative genome-wide genetic correlations between LTL and six major CVDs, except for LTL-AF. Beyond mere genetic correlations, our analysis uncovers a more extensive degree of genetic overlap between LTL and CVDs using MiXeR and LAVA, involving a broad mixture of both concordant and discordant effect sizes. Employing MiXeR, we demonstrated that the polygenicity of LTL and CVDs presents fundamental distinctions beyond SNP-based heritability. Briefly, LTL was substantially less polygenic than three CVD phenotypes (CAD, HF, and stroke), yet exhibited similar polygenicity to other CVD phenotypes, including AF, VTE, and PAD. Despite notable differences in polygenicity, we observed extensive genetic overlaps between LTL and all CVDs, supported by LAVA local correlations. This pattern emerged in cases of weak or non-significant genome-wide genetic correlations, such as between LTL and AF, and in strong genome-wide correlations, such as between LTL and PAD. For example, despite the absence of genome-wide genetic correlations, MiXeR indicated that a pronounced fraction of the genetic risk underlying LTL overlaps with AF. The findings correspond with the discovery of a similar number of positively and negatively correlated genomic regions between LTL and AF by LAVA, alongside further detecting much more shared

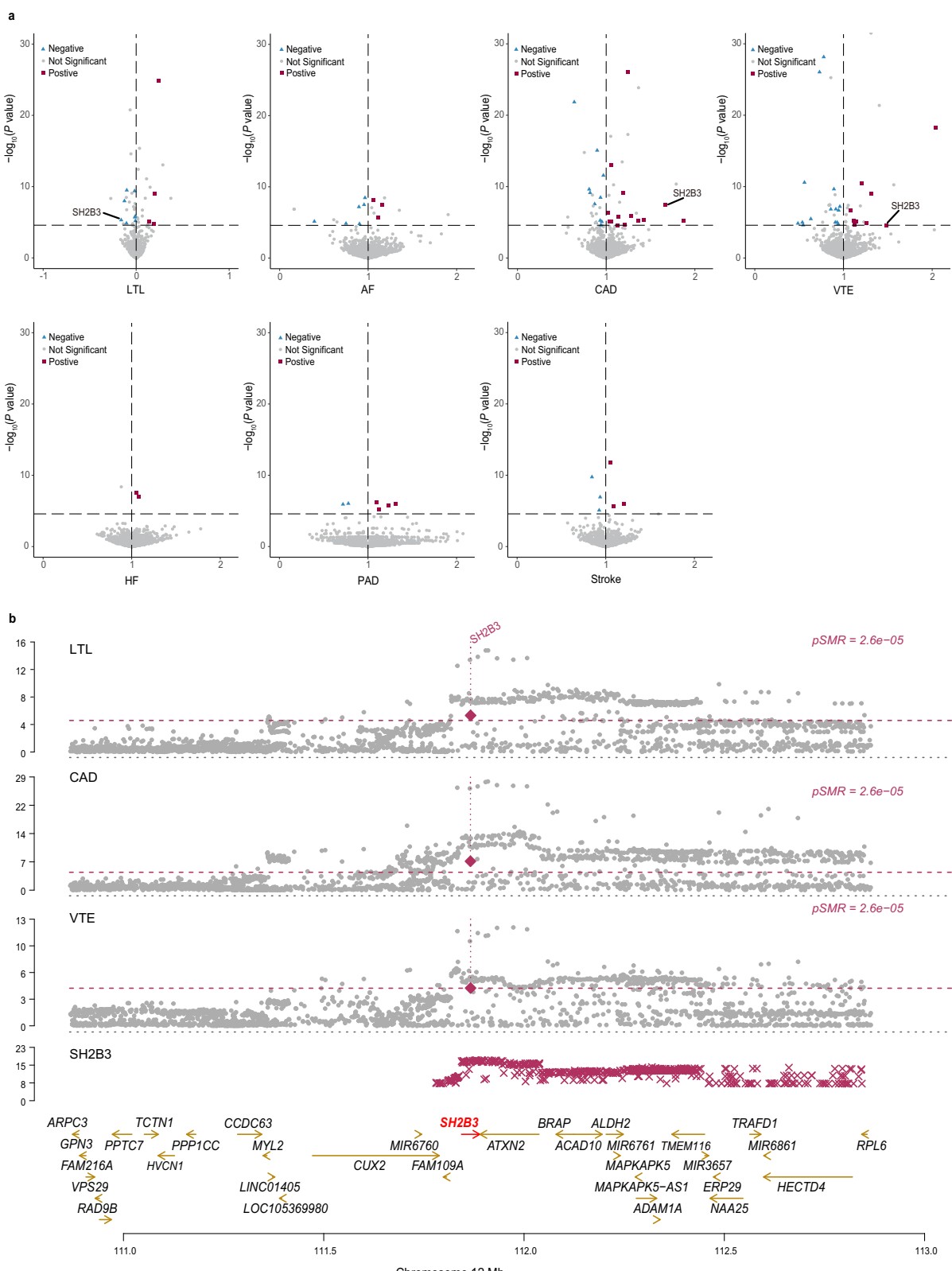

pleiotropic loci below the genome-wide significance threshold. Although the local genetic correlations encompassed genomic regions averaging approximately 1 megabase (Mb) in width, this resolution required refinement to minimize the impact of heterogeneous effects on estimation accuracy. For example, we noted inconsistent effects across lead variants in the LTL and AF pleiotropic analysis. Specifically, out of 47 top lead variants, 28 demonstrated the same effect direction for LTL and AF, whereas the remaining 19 exhibited opposite effect directions. This indicates that pleiotropic analysis at the level of single variants was necessary to offer further, more detailed insights into the shared genetic underpinnings across complex traits. These findings indicate that genetic overlap between LTL and CVDs was greatly underestimated due to the patterns of mixed effect directions concealed by estimates of genome-wide genetic correlations.

**Fig. 6 | Result of summary data-based Mendelian Randomization analysis on the associations between leukocyte telomere length and six cardiovascular diseases. a** Volcano plots based on SMR showing circulating proteins (red squares, blue diamonds, or gray dots) with the associations between circulating protein levels and each of the traits (x-axis) and the corresponding Pvalue (y-axis). Red squares indicate positive causal relations. Blue diamonds indicate negative causal relations. Gray dots indicate insignificant causal relations. Note that the volcano plots were plotted at Pvalues truncated by $1 \times 10^{-30}$ for better visualization, thus

excluding a protein, CELSR2, influencing CAD from volcano plots of SMR. **b** Miami plots of the 12q24.12 region. Genes that were mapped under the locus at 12q24.12 were highlighted in the analysis. The protein SH2B3 is labeled as significantly associated with LTL, CAD, and VTE. The red horizontal dashed line corresponds to Bonferroni correction ($P < 2.60 \times 10^{-5}$). All statistical tests were two-sided. LTL leukocyte telomere length, AF atrial fibrillation, CAD coronary artery disease, VTE venous thromboembolism, HF heart failure, PAD peripheral artery disease.

The intricate relationship between LTL and six major CVDs may be further elucidated by examining the vertical and horizontal pleiotropy mechanisms that underpin their shared genetic basis. The causal relationships between LTL and six major CVDs were then mainly explained by the effects attributed to vertical pleiotropy. We found only the negative causal relationship between LTL and CAD using LCV and MRlap methods. This finding is in stark contrast to existing research that suggested such effects were present. Previous two-sample MR studies have reported associations between genetic liability to LTL shortening and the increased risk of CAD and stroke, but the results for HF risk have been inconsistent[22,23]. Another study even demonstrated that genetically predicted AF contributes to LTL shortening rather than the reverse[24]. It is important to note that the interpretation of the MR estimate in this case was quite complicated by the fact that both LTL and CVDs were time-varying outcomes with a late age of onset. Taken together, our findings suggest that the associations between LTL and CVDs observed in prior research may have been overstated due to a combination of reverse causation, surveillance bias, or unmeasured confounding[53–55], indicating that pleiotropic and common biological pathways may be a better explanation for their association.

The horizontal pleiotropic analyses, covering various levels such as pleiotropic genetic variants and loci, pleiotropic genes, biological pathways, and protein targets, revealed significant genetic overlaps between LTL and CVDs from another perspective. At the SNP level, pleiotropic variants linking LTL and CVDs were broadly distributed, with a particular focus on shared pleiotropic loci among certain trait pairs, including 16q22.1 (*TMED6*), 8p21.3 (*XPO7*), 17p13.3 (*SERPINF1*), and 12q24.12 (*ATXN2*). For example, the transmembrane emp24 domain-containing protein 6 precursor (*TMED6*), highly and selectively expressed in pancreatic islets, was found to be associated with LTL and all CVDs, which belonged to the EMP24_GP25L superfamily and played a crucial role in protein trafficking and secretion. Knockdown of the *TMED6* gene in Min6 β-cells and INS1 cells led to a reduction in glucose-stimulated insulin secretion, suggesting that dysregulation of *TMED6* may play a critical role in the onset of type 2 diabetes[56,57]. To date, there have been no reports of an association between *TMED6* and either LTL or CVDs, hinting at a potential biological mechanism that may be mediated through CVD risk factors, particularly type 2 diabetes. Exportin-7 (*XPO7*) is a bidirectional transporter regulating the nuclear-cytoplasmic shuttling of a wide array of substrates, yet its function remains relatively obscure[58]. Recent studies have identified *XPO7* as a regulator of cellular senescence. Aging significantly reduces peak oxygen consumption (V̇o2peak), a good predictor of health, and low V̇o2peak is a strong independent risk factor for CVD morbidity and mortality[59]. Transcriptomic analyses have further revealed that *XPO7* reduces V̇o2peak by inhibiting protein translation, underscoring its potential role in CVD development[60]. The role and malfunction of the Serpin Family F Member 1 (*SERPINF1*), also known as the pigment epithelium-derived factor (*PEDF*) gene, in the aging process has currently been a hot topic. Briefly, the reduction in *PEDF* expression levels, both directly and indirectly, can induce cellular senescence via the modulation of various signaling pathways[61,62]. Studies have also demonstrated that *PEDF* possesses insulin-sensitizing effects in the liver and adipose tissues and exhibits anti-inflammatory, anti-thrombogenic, and vasculoprotective properties in vivo, offering

protection against metabolic syndrome and cardiovascular diseases[62,63]. Overall, this study extends previous findings on shared genetic architecture by offering a more comprehensive characterization of specific pleiotropic loci.

At the gene level, we employed two gene-mapping strategies to identify credible mapped genes for all jointly associated pleiotropic loci. Seven genes, namely *ALDH2*, *ACAD10*, *TMEM116*, *SH2B3* (all located at 12q24.12), *TMED6* (16q22.1), *SERPINF1* (17p13.3), and *XPO7* (8p21.3), have been identified as the most pleiotropic, influencing over half of the trait pairs. For example, *ALDH2* and *ACAD10*, located at the 12q24.12 locus, were associated with LTL and all CVDs except AF. Aldehyde dehydrogenase 2 (*ALDH2*) is the gene with the highest number of genetic polymorphisms in humans. It encodes a mitochondrial enzyme critical for detoxifying reactive aldehydes. For example, the *ALDH2* rs671 inactivating polymorphism, found in up to 8% of the global population and up to 50% of the East Asian population, is associated with an elevated risk of several CVDs, such as CAD. While numerous studies have connected aldehyde accumulation, due to alcohol consumption, ischemia, or heightened oxidative stress, to elevated CVD risk, this accumulation alone does not fully account for their complex interactions[64,65]. Moreover, previous studies indicated that the *ALDH2* enzyme might also exacerbate myocardial remodeling and contractile dysfunction during aging, potentially via AMPK/Sirt1-mediated mitochondrial damage. Acyl-CoA dehydrogenase family member 10 (*ACAD10*) gene encodes an enzyme involved in fatty acid beta-oxidation in mitochondria[66]. *ACAD10* is predominantly expressed in the human brain and is believed to play a role in physiological functions within the central nervous system, such as the regulation of cellular lipid synthesis. *ACAD10* has previously been identified as one of the candidate causal genes for CAD, stroke, and hypertension, a common risk factor for CVDs[48]. Collectively, these findings imply that *ACAD10* has a significant role in regulating lipid synthesis through distinct molecular and cellular pathways, potentially influencing the development of CVDs. Further research is required to elucidate the precise mechanisms of interaction between ACAD10 and both LTL and CVDs.

The 12q24.12 locus was a top hit region identified as pleiotropic for all correlated trait pairs except for LTL-AF, showing evidence of colocalization between these trait pairs. Besides, HyPrColoc further revealed robust colocalization evidence for this locus between LTL and all CVDs, excluding AF and PAD, highlighting the shared causal SNP (i.e., rs10774625). LAVA results supported these findings and showed consistent negative local genetic correlations between LTL and all CVDs, except for AF, HF, and PAD. The index SNP rs10774625 polymorphism has been reported to be associated with a variety of CVDs, notably CAD, alongside cardiometabolic markers such as blood pressure and blood lipids. The index SNP rs10774625, located at the 12q24.12 locus (an intronic variant of the *ATXN2* gene), was associated with eQTLs and pQTLs for *SH2B3* in whole blood. The SMR revealed that genetically predicted *SH2B3* protein levels were significantly associated with both LTL-CAD and LTL-VTE. *SH2B3* is a member of the adapter protein family and plays a critical role in negatively regulating cytokine signaling and cell proliferation. It was originally described as a regulator of hematopoietic and lymphocyte differentiation and was implicated in the transduction and regulation of growth factors and inflammation-related cytokine receptor-mediated signaling. *SH2B3*

missense variants may affect lifespan through their effects on CVDs[67]. Specifically, *SH2B3* can lead to increased production of IFNγ, which acts as a pro-inflammatory mediator to induce the polarization of macrophages into different states. This process is crucial for mediating inflammatory regulation and fibrosis post-myocardial infarction. On the other hand, IFNγ is released and activated by CD4 T helper cells, specifically Th1 cells, which are instrumental in coordinating immune cell infiltration and inflammation. The above immune cells and inflammatory signals are pivotal in the progression of non-ischemic heart failure in patients[68]. Notably, increased cardiomyocyte size and fibrosis are critical characteristics of cardiac hypertrophy and remodeling, which ultimately lead to heart failure[69]. Further studies have demonstrated that cardiac-specific *SH2B3* overexpression exacerbates pressure overload, leading to cardiac hypertrophy, fibrosis, and dysfunction by activating focal adhesion kinase, which subsequently triggers the downstream phosphoinositide 3-kinase-AKT-target of Programmed Death-1 (PD-1). Significant overexpression of the *SH2B3* gene promotes the activation of the Akt signaling pathway, which can promote cardiac hypertrophy and fibrosis and lead to the deterioration of cardiac function[70]. Meanwhile, age-related telomere dysfunction is a core driver of inflammation[71]. Therefore, *SH2B3* may become one of the most promising therapeutic targets for CVDs. In contrast, *TMEM116*, a member of the TMEM family of proteins that spans the plasma membrane of cells to facilitate intercellular communication, demonstrates a different aspect of disease association. Multiple members of the TMEM family may be up- or down-regulated in tumor tissues, and some of them are used as cancer prognostic biomarkers[72]. However, in studies of cardiovascular diseases, this protein has only been found to be related to coronary atherosclerosis[73], and the specific mechanism is unclear and requires further research.

At the pathway level, functional analyses of the pleiotropic loci between LTL and CVDs have implicated genes involved in the metabolic processes of nucleobase-containing compounds, including DNA biosynthesis and telomere maintenance. There is a close relationship between telomere maintenance, telomerase expression, and extension of cell lifespan. Telomeres undergo shortening during repeated cell divisions, and when their length diminishes to a critical point, the resultant genomic instability can lead to further genetic abnormalities that promote cell death or apoptosis, which is a hallmark of cellular senescence. Estrogen, stress accumulation from oxidative damage, hypertension, and other factors are believed to significantly impact telomere homeostasis and contribute to the development of CVDs[74]. This finding supports the theory that progressive telomere shortening contributes to the pathogenesis of age-related human diseases such as CVDs. Specifically, Minamino et al. documented the presence of vascular endothelial cells exhibiting age-related phenotypes within human atherosclerotic lesions[75]. Excessive vascular smooth muscle cells are stimulated to proliferate and migrate, leading to the growth of atherosclerosis. Besides, telomerase activation and telomere maintenance are critical in increasing the proliferation and growth of vascular smooth muscle cells. Activation through the telomerase reverse transcriptase component (TERT) extends the lifespan of cultured vascular smooth muscle cells. Conversely, telomerase inhibition can extend the lifespan and reduce the proliferation of cultured vascular smooth muscle cells, thereby decreasing the risk of atherosclerosis[76]. Therefore, these findings highlight the role of telomeres in cardiovascular health, suggesting that modulating telomerase activity may be critical for developing effective interventions for related CVDs.

Our study highlights the critical role of telomere maintenance in the shared genetic etiology of LTL and CVDs, offering potential avenues for therapeutic intervention. Previous research has demonstrated that telomerase can counteract early plaque-associated vascular smooth muscle cell (VSMC) senescence, enabling cell proliferation even when telomeres are critically short[77,78]. However, telomere shortening may also contribute to the accumulation of senescent

endothelial cells and VSMCs, exacerbating atherosclerosis and inflammation—two key drivers of coronary heart disease. These findings underscore the potential for therapeutic strategies aimed at preserving or restoring telomere length to mitigate cardiovascular risk. For example, pharmacological interventions that maintain telomere integrity or enhance telomerase activity may help reduce CVD susceptibility in high-risk individuals. Additionally, our findings suggest that LTL and its associated genetic variants could serve as valuable biomarkers for early cardiovascular risk detection. Identifying individuals with shorter telomeres or those carrying genetic variants linked to LTL may help healthcare providers to stratify patients at elevated risk for CVDs before clinical symptoms emerge. This proactive approach could enable earlier intervention strategies, such as lifestyle modifications, personalized risk management, or more frequent monitoring, to slow or prevent disease progression.

There were some limitations to the current study. Firstly, the main analysis focused solely on individuals of European ancestry due to the scarcity of sufficiently powered GWAS involving other ancestries. Nevertheless, we utilized GWAS summary data from East Asian ancestries for replication, partially confirming the consistency of the genetic foundation identified in the European sample (Supplementary Note 1). Future studies with a cross-ancestral approach are necessary to evaluate the universality of these findings. Secondly, the analysis focused on common genetic variants that account for only a small fraction of overall disease risk. The remaining variance was likely attributable to many undetected SNPs, rare variants, or gene interactions. Therefore, they need to be further studied to achieve a more complete understanding. Thirdly, although we uncovered the potential shared genetic architecture, the mechanisms of shared biological pathways still require further experimental validation. In particular, loci such as *ATXN2*, which have been associated with multiple metabolic traits—including blood pressure and blood lipid levels—may function as pleiotropic loci that influence a broad spectrum of traits through complex biological mechanisms. Future studies should aim to elucidate these pleiotropic effects more comprehensively, potentially uncovering shared regulatory pathways that contribute to the pathogenesis of diverse cardiometabolic conditions. Fourthly, due to the overlap between proteomic data from the UK Biobank and GWAS summary data for LTL and CVDs, causal inference based on the shared causal protein fraction between LTL and CVDs may be influenced to some extent. Finally, our analysis of GWAS summary data encompassed six major CVDs, representing a significant portion of the genetic risk architecture for these conditions, though not comprehensively. As GWAS datasets expand, it will be crucial to undertake cross-trait analyses incorporating more varied datasets and additional diseases to enhance our understanding.

In conclusion, we identified significant polygenic overlap between LTL and CVDs, characterized by distinct genetic correlation patterns and varying effect directions. Our MR results suggest that genetically inferred LTL shortening was linked to a higher risk of CAD. Furthermore, we demonstrated that LTL and six major CVDs share pleiotropic genetic variants, loci, genes, biological pathways, and protein targets, highlighting the role of DNA biosynthesis and telomere maintenance in their common genetic etiology. These findings provide valuable insights into the interconnected mechanisms underlying LTL and CVDs, with potential implications for guiding targeted therapies and informing clinical practice.

## Methods
### Ethics
The study was conducted in full compliance with ethical requirements. Ethical approval for all GWAS was obtained from the relevant ethics committees, and written informed consent was acquired from all participants. The study design and conduct complied with all relevant regulations regarding the use of human study participants and was

conducted in accordance with the criteria set by the Declaration of Helsinki.

## Data sources and quality control

Figure 1 and Supplementary Fig. 1 outline the workflow for our study. Due to the confounding effects of ancestral differences in linkage disequilibrium (LD) structure and the scarcity of sufficiently large multi-ancestry samples, we limited our main analysis to individuals of European ancestry. We sourced genome-wide association study (GWAS) summary statistics from the most comprehensive and recent publicly available datasets of European ancestry. Specifically, GWAS summary statistics for leukocyte telomere length (LTL) were derived from a published GWAS comprising 472,174 individuals from the UK Biobank[15], of whom 95% were of European ancestry. LTL was quantified as the ratio of telomere repeats copy number (T) to a single copy gene (S) in a mixed leukocyte population, measured via a multiplex quantitative polymerase chain reaction (qPCR) assay, and subsequently log-transformed to achieve an approximation to a normal distribution. Our selection criteria for GWAS included studies with sample sizes exceeding 50,000 to ensure adequate statistical power. Accordingly, we included GWAS summary statistics for six major cardiovascular diseases (CVDs): atrial fibrillation (AF)[16], coronary artery disease (CAD)[17], venous thromboembolism (VTE)[18], heart failure (HF)[19], peripheral artery disease (PAD)[20], and stroke[21]. AF GWAS summary statistics were sourced from a genome-wide meta-analysis of six studies (The Nord-Trøndelag Health Study [HUNT], deCODE, the Michigan Genomics Initiative [MGI], DiscovEHR, UK Biobank, and the Atrial Fibrillation Genetics [AFGen] Consortium), encompassing 60,620 AF cases and 970,216 controls of European ancestry. For CAD, we utilized GWAS summary statistics from a genome-wide meta-analysis by the CARDIoGRAMplusC4D Consortium and the UK Biobank, which included 181,522 cases and 984,168 controls. VTE GWAS summary statistics were extracted from a meta-analysis of 81,190 cases and 1,419,671 controls of European ancestry across 7 cohorts (the Copenhagen Hospital Biobank Cardiovascular Disease Cohort [CHB-CVDC], Danish Blood Donor Study [DBDS], deCODE, Intermountain Healthcare, UK Biobank, FinnGen, and Million Veterans Program [MVP] Consortium). GWAS summary statistics for HF came from the Heart Failure Molecular Epidemiology for Therapeutic Targets (HERMES) Consortium, including 47,309 cases and 930,014 controls. GWAS summary statistics for PAD were derived from a genome-wide meta-analysis of 11 independent GWASs, totaling 12,086 cases and 499,548 controls. Lastly, GWAS summary statistics for stroke were obtained from the GIGASTROKE consortium, which comprised 73,652 cases and 1,234,808 controls of European ancestry. Detailed information about these GWAS summary statistics and their original publication sources is available in Supplementary Data 1.

Prior to further analysis, stringent quality control measures were applied to the GWAS summary statistics, encompassing several key steps: (i) alignment with the hg19 genome build, referencing the 1000 Genomes Project Phase 3 Europeans; (ii) restriction of the analysis to autosomal chromosomes; (iii) removal of single nucleotide polymorphisms (SNPs) lacking a rsID or presenting duplicated rsIDs; and (iv) exclusion of rare or low-frequency variants, defined by a minor allele frequency (MAF) less than 1%. To ensure robust and interpretable comparisons between LTL and CVDs, we standardized the summary statistics to include only SNPs present across all analyzed phenotypes, resulting in a cohesive dataset of 6,923,146 SNPs. Additionally, in subsequent analyses, we implemented further data processing techniques tailored to the specific requirements of various statistical tools.

## Genetic overlap

To explore the shared genetic foundations between LTL and six major CVDs, we evaluated genetic overlap across genome-wide, polygenic, and local levels.

## Genome-wide genetic correlation analysis between LTL and CVDs

At the genome-wide level, we analyzed the genetic correlations ($r_g$) between LTL and six major CVDs using cross-trait LD score regression (LDSC)[27]. LDSC facilitates the estimation of the average genetic effect sharing across the entire genome between two traits, leveraging GWAS summary statistics. This includes the contribution of SNPs below the threshold of genome-wide significance and accounts for potential confounding factors such as polygenicity, sample overlap, and population stratification. This analysis utilized pre-computed LD scores from the European reference panel in the 1000 Genomes Project Phase 3, excluding SNPs that did not overlap with the reference panel. Notably, the major histocompatibility complex (MHC) region (chr6: 25–34 Mb), known for its intricate LD structure, was omitted from the main analysis. LDSC analysis estimated the genetic correlations between LTL and the six major CVDs. This method utilizes a weighted linear model, where the product of Z-statistics from two traits is regressed against the LD score across all genetic variants genome-wide. Genetic correlations with $P$ values below the Bonferroni-adjusted threshold ($P = 0.05 /$ number of trait pairs $= 0.05 / 6 = 8.33 \times 10^{-3}$) were deemed statistically significant.

To elucidate the biological underpinnings of the shared genetic predisposition to LTL and six major CVDs, we employed stratified LDSC applied to specifically expressed genes (LDSC-SEG)[44] to identify relevant tissue types and chromatin modification. This analysis incorporated multi-tissue gene expression data from the Genotype-Tissue Expression (GTEx) project[79] and the Franke lab[80,81]. Additionally, we utilized chromatin-based annotations associated with six epigenetic marks (DNase hypersensitivity, H3K27ac, H3K4me1, H3K4me3, H3K9ac, and H3K36me3) for validation purposes. These annotations included 93 labels from the Encyclopedia of DNA Elements (ENCODE) project[82] and 396 labels from the Roadmap Epigenomics database[83]. We adjusted the $P$ values for the significance of the coefficients using the false discovery rate (FDR) method. An FDR threshold of $< 0.05$ was established as the criterion for statistical significance.

## Polygenic overlap analysis between LTL and CVDs

To augment the genome-wide genetic correlation analysis, we engaged the causal mixture modeling approach (MiXeR) to quantify the polygenic overlap between LTL and six major CVDs, independent of the directions of genetic correlations. MiXeR estimates the quantity of shared and phenotype-specific "causal" variants that exert non-zero additive genetic effects, accounting for 90% of SNP-heritability in each trait[28]. This 90% SNP-heritability threshold minimizes the influence of variants with negligible effects. Importantly, MiXeR's capacity to evaluate polygenic overlap without regard to the directional effects of variants offers a nuanced view of local genetic associations that might be obscured in traditional genome-wide genetic correlation estimations due to opposing variant effects. Initially, univariate MiXeR analyses estimated the count of "causal" variants for LTL and six major CVDs, assessing both polygenicity and the average magnitude of additive genetic effects among these variants. The LD structure was determined using the genotype reference panel from the 1000 Genomes Project Phase 3. The MHC region (chr6: 25–34 Mb) known for its intricate LD structure, was excluded from the main analysis. Subsequently, bivariate MiXeR analysis quantified the polygenic overlap between LTL and the six CVDs. This analysis delineated the additive genetic effects across four categories: (i) SNPs with zero effect on both traits; (ii) trait-specific SNPs with non-zero effects on the first trait; (iii) trait-specific SNPs with non-zero effects on the second trait; and (iv) SNPs affecting both traits. MiXeR also estimates dice coefficient scores (i.e., the proportion of shared SNPs between two traits out of the total number of SNPs estimated to influence both traits) to quantify polygenic overlap. Additionally, MiXeR calculated the overall genetic correlations ($r_g$), the genetic correlation of shared variants ($r_gs$), and the

fraction of variants with concordant effects in the shared component. The model's predictive accuracy was evaluated by comparing the modeled versus actual data, utilizing conditional quantile-quantile (Q-Q) plots, log-likelihood plots, and the Akaike information criterion (AIC). Positive AIC differences are interpreted as evidence that the best-fitting MiXeR estimates are distinguishable from the reference model. A negative AIC value indicates that the MiXeR model fails to distinguish effectively between maximal and minimal overlap scenarios.

## Local genetic correlation between LTL and CVDs

To investigate whether there are any genomic loci with pronounced genetic correlations despite negligible genome-wide $r_g$, we utilized the Local Analysis of [co]Variant Annotation (LAVA)[29] method to estimate the localized genetic correlation between LTL and six major CVDs. LAVA, based on a fixed-effects statistical model, enables the estimation of local SNP heritability (loc-$h^2_{SNP}$) and genetic correlations (loc-$r_g$s) across 2495 semi-independent genetic regions of approximately equal size (-1 Mb). This method is adept at identifying loci with mixed effect directions, offering a nuanced measure of genome-wide genetic overlap, albeit influenced by the statistical power of the underlying GWAS data. For this analysis, we adopted the genomic regions defined by Werme et al. as autosomal LD blocks, characterized by minimal inter-LD block linkage with an average size of 1 million bases, each containing at least 2500 variants. The LD reference panel from the 1000 Genomes Project Phase 3 for European samples was employed, and consistent with LDSC and MiXeR analyses, the MHC region (chr6: 25–34 Mb) was excluded. Identifying meaningful loc-$r_g$s requires a significant local genetic signal; thus, a stringent P value threshold ($P < 1 \times 10^{-4}$) was applied to filter out non-significant loci. Subsequent bivariate testing was conducted on selected loci and traits with notable univariate genetic signals. For these loci, P values for loc-$r_g$s were adjusted using the FDR method, with an FDR < 0.05 establishing statistical significance. LAVA incorporated an estimate of sample overlap (genetic covariance intercept) from bivariate LDSC analyses to account for potential sample overlap effects.

In cases where shared risk loci were identified across multiple phenotypes, the Hypothesis Prioritisation for multi-trait Colocalization (HyPrColoc) method was employed to assess whether association signals across more than one trait were colocalized[30]. HyPrColoc, an efficient deterministic Bayesian clustering algorithm, leverages GWAS summary statistics to identify clusters of colocalized traits and potential causal variants within a genomic locus, providing a posterior probability (PP) of colocalization for each cluster. Loci with a PP > 0.7 were deemed colocalized, enhancing our understanding of shared genetic architectures across traits.

## Pleiotropy insights to dissect genetic overlap

**Causal inference between LTL and CVDs.** To elucidate the potential causal relationships underlying the genetic correlations observed between LTL and six major CVDs, we employed latent causal variable (LCV) analysis. This method posits that the genetic correlation between two traits operates through a latent factor, allowing for the distinction between genetic causality (vertical pleiotropy) and both correlated and uncorrelated horizontal pleiotropy[32]. It achieves this by estimating the genetic causality proportion (GCP) across all genetic variants, where GCP quantifies the proportion of each trait's heritability explained by a mutual latent factor. It is essential to emphasize that GCP does not indicate the magnitude of causal effects but only implies a causal relationship between traits. This method provides insights into whether the impact of one trait on the second exceeds the evidence in the reverse direction. The sign of genetic correlation can be employed to infer the consequence of the partial genetic causality of one trait on another. A GCP greater than zero suggests a partial genetic causal relationship from trait 1 to trait 2 and vice versa, with

values closer to |GCP| = 1 indicating stronger evidence of vertical pleiotropy. Conversely, a GCP of zero implies horizontal pleiotropy. We considered an absolute GCP estimate (|GCP|) greater than 0.60 as indicative of substantial genetic causality, applying a Bonferroni-corrected significance threshold of $P < 8.33 \times 10^{-3}$ to account for multiple comparisons across trait pairs. The LCV analysis, while robust, assumes a singular, unidirectional latent variable driving the genetic correlation, a premise potentially confounded by bidirectional causal effects or multiple latent factors.

To address the limitations inherent to LCV, we performed a sensitivity analysis for pairs of CVDs that showed partial genetic causality using the MRlap method[33]. MRlap mitigates weak instrument bias and winner's curse while accounting for sample overlap and its potential impact on these biases. The method first calculates the observed MR-based effect values (IVW) and then applies corrections based on genetic covariance, computed through LDSC. MRlap also calculates a test statistic to assess whether the corrected estimate significantly deviates from the IVW estimate ($P < 0.05$). If no significant difference is detected, the IVW estimate remains valid. However, if a significant difference is observed, the corrected MRlap estimate is preferred due to its reduced bias and independence from sample overlap. For sensitivity analyses, we employed traditional MR methods, including MR Egger, weighted median, IVW, simple mode, and weighted mode[84–86]. We adjusted for multiple testing with a threshold of $P < 4.17 \times 10^{-3}$, considering both the number of trait pairs and the number of tests conducted.

**Pairwise pleiotropic analysis between LTL and CVDs.** To explore the role of horizontal pleiotropy between LTL and six major CVDs, we utilized the pleiotropic analysis under composite null hypothesis (PLACO) method to conduct a comprehensive genome-wide identification of pleiotropic SNPs that concurrently influence the risk of both traits. PLACO operates on the composite null hypothesis that asserts a genetic variant is either associated with just one or neither trait, thereby distinguishing between pleiotropic effects and singular trait associations[34]. The method evaluates this hypothesis by examining the product of the Z statistics derived from the GWAS summary statistics of both traits, formulating a null distribution of the test statistic as a mixture distribution. This allows for the identification of SNPs that may be linked to only one or none of the phenotypes under study. The threshold for identifying pleiotropic SNPs with significant evidence of genome-wide pleiotropy was set at $P_{PLACO} < 5 \times 10^{-8}$. Additionally, to adjust for possible sample overlap, we decorrelated the Z-scores using a correlation matrix directly estimated from the GWAS summary statistics, ensuring a more accurate interpretation of pleiotropic effects.

**Characterization of pleiotropic loci and functional annotation.** The Functional Mapping and Annotation of Genome-Wide Association Studies (FUMA) platform was employed to identify independent genomic loci and conduct functional annotation for pleiotropic SNPs revealed by PLACO analysis[34]. FUMA, leveraging data from 18 biological databases and analytical tools, annotates GWAS findings to highlight probable causal genes through positional and eQTL mapping[35]. The 1000 Genomes Project Phase 3 European-based LD reference panels were utilized for LD structure correction. Initially, FUMA distinguishes independent significant SNPs (meeting genome-wide significance at $P < 5 \times 10^{-8}$ and $r^2 < 0.6$), further defining a subset as lead SNPs based on mutual independence ($r^2 < 0.1$). LD blocks within 500 kb of lead SNPs are merged to delineate distinct genomic loci, with the SNP exhibiting the lowest P value in each locus designated as the top lead SNP. The analysis then assesses directional effects between LTL and six major CVDs by comparing Z-scores of these top lead SNPs. SNPs achieving genome-wide significance ($P < 5 \times 10^{-8}$) in individual GWAS for each trait were annotated using FUMA for comparative analysis. The identified pleiotropic loci were considered novel if they

did not coincide with the loci previously reported in the original GWAS for LTL or any of the six major CVDs. In other words, to be deemed 'novel,' a locus identified through FUMA should not have exhibited statistical significance in the single-trait GWAS.

To elucidate the biological underpinnings of the observed statistical associations, lead SNPs were annotated using Annotate Variation (ANNOVAR)[36] for their proximity to genes and potential impact on gene function. The Combined Annotation-Dependent Depletion (CADD) score[39], which aggregates insights from 67 annotation resources, was used to assess the deleteriousness of variants. Variants with CADD scores greater than 12.37 were deemed likely to exert deleterious effects. Furthermore, the RegulomeDB score provided a categorical assessment of an SNP's regulatory potential based on expression quantitative trait loci (eQTL) and chromatin marks, ranging from 1 (strong evidence of regulatory functionality) to 7 (minimal evidence)[40]. The highest regulatory potential is indicated by a score of 1a, while a score of 7 suggests the least regulatory significance. For the identification of putative causal genes, SNPs were mapped using two approaches: positional mapping within a 10-kb window around the SNP and eQTL mapping. eQTL mapping used information from GTEx v8.

**Colocalization analysis.** For pleiotropic loci identified and annotated by FUMA, we conducted a colocalization analysis using COLOC to pinpoint potential shared causal variants across pairwise traits within each locus. COLOC evaluates five mutually exclusive hypotheses for each pair of traits at a locus[41]: H0 posits no association with either trait; H1 and H2 suggest an association with only one of the traits; H3 indicates that both traits are associated due to different causal variants; and H4 implies a shared association for both traits stemming from the same causal variant. The analysis was performed using default COLOC prior probabilities: p1 and p2, each set at $1 \times 10^{-4}$ for an SNP's association with the first and second trait, respectively, and p12 at $1 \times 10^{-5}$ for an SNP associated with both traits. A Posterior Probability for Hypothesis 4 (PP.H4) greater than 0.7 was considered evidence for colocalization, suggesting the presence of shared causal variants at the locus. The SNP exhibiting the highest PP.H4 was identified as a candidate causal variant.

**Gene level analyses.** Building on the insights from PLACO, we delved into the shared biological processes and pathways involving the identified pleiotropic loci. Through gene-level analysis using Multi-marker Analysis of GenoMic Annotation (MAGMA)[42], we assessed genes within or intersecting the pleiotropic loci, integrating data from both PLACO and single-trait GWAS. Unlike permutation-based approaches, MAGMA employs a multiple regression model that incorporates principal component analysis to evaluate gene associations. This model calculates a Pvalue for each gene, aggregating the impact of all SNPs linked to that gene while considering gene size, SNP count per gene, and linkage disequilibrium (LD) among the markers. SNPs were attributed to genes based on their location within the gene body or within a 10 kb range upstream or downstream. The LD calculations leveraged the 1000 Genomes Project Phase 3 European population as the reference panel, with SNP locations determined using the human genome Build 37 (GRCh37/hg19) and focusing on 17,636 autosomal protein-coding genes. A gene was deemed significant if its Pvalue was below 0.05 after applying a Bonferroni correction for the total number of protein-coding genes and the six trait pairs analyzed ($P = 0.05 / 17,636 / 6 = 4.73 \times 10^{-7}$). Due to complex LD patterns, the MHC region (chr6: 25–34 Mb) was excluded from MAGMA's gene-based analysis.

To overcome the limitations of MAGMA, which assigns SNPs to their nearest genes based on arbitrary genomic windows potentially missing functional gene associations due to long-range regulatory effects, we employed eQTL-informed MAGMA (e-MAGMA)[43] for a more nuanced investigation of tissue-specific gene involvement based on PLACO results. e-MAGMA retains the statistical framework of MAGMA, using a multiple linear principal component regression model, but enhances gene-based association analysis by incorporating tissue-specific cis-eQTL information for SNP assignment to genes, which yields more biologically relevant and interpretable findings. For our analysis, we utilized eQTL data from tissues provided by the GTEx v8 reference panel, as available on the e-MAGMA website. Guided by the principle that analyses focused on disease-relevant tissues yield more pertinent insights, we selected ten relevant tissues for our study. These included three artery tissues, two adipose tissues, two heart tissues, and three additional tissues (whole blood, liver, and EBV-transformed lymphocytes), selected based on their significant enrichment in the LDSC-SEG analysis. The LD reference data for our analysis came from the 1000 Genomes Project Phase 3 European panel. We calculated tissue-specific Pvalues for each gene across the selected tissues, with significance determined after Bonferroni correction for the number of tissue-specific protein-coding genes and trait pairs examined. For instance, the significance threshold for adipose subcutaneous tissue was set at $P = 0.05 / 9613 / 6 = 8.67 \times 10^{-7}$. Similar to MAGMA, e-MAGMA analysis results within the MHC region (chr6: 25–34 Mb) were excluded to avoid confounding due to complex LD patterns. Additionally, we conducted a transcriptome-wide association study (TWAS)[46] based on single-trait GWAS results using the functional summary-based imputation software FUSION, applying tissue-specific Bonferroni corrections to determine significance. The FUSION approach integrates GWAS summary statistics with pre-computed gene expression weights, referencing the same tissues analyzed in the e-MAGMA study from the GTEx v8 dataset. This integration takes into account the LD structures to identify significant relationships between gene expression levels and specific traits.

**Pathway level analyses.** To investigate the genetic pathways underlying the comorbidity of LTL and six major CVDs, we utilized MAGMA for gene-set analysis. This analysis employs a competitive approach, where test statistics for all genes within a gene set, such as a biological pathway, are aggregated to produce a joint association statistic. Gene sets were sourced from Gene Ontology biological processes (GO_BP) via the Molecular Signatures Database (MsigDB)[49], with gene definitions and association signals derived from MAGMA gene-based analysis. We adjusted for multiple testing using a Bonferroni correction, setting the threshold at $P = 0.05 / 7744 / 6 = 1.08 \times 10^{-6}$. We then conducted functional enrichment analysis on the genes overlapping across more than one trait pair, which were significantly identified by both MAGMA and e-MAGMA analyses. For this purpose, the ToppGene Functional Annotation tool (ToppFun) was employed to identify significantly represented biological processes and enriched signaling pathways, considering the entire genome as the background[50]. ToppFun performs Functional Enrichment Analysis (FEA) on the specified gene list, leveraging a broad spectrum of data sources, including transcriptomics, proteomics, regulomics, ontologies, phenotypes, pharmacogenomics, and bibliographic data. The list of candidate genes was submitted to the ToppFun tool within the ToppGene Suite, with an FDR < 0.05 set as the threshold for statistical significance.

**Proteome-wide Mendelian randomization study analysis.** To explore potential common causal factors at the proteomic level, we utilized summary data-based Mendelian Randomization (SMR)[51] to examine associations between protein abundance and disease phenotype. This analysis leveraged index cis-acting variants (cis-pQTLs) identified in the UK Biobank Pharma Proteomics Project (UKB-PPP), which includes plasma samples from 34,557 European individuals with the measurement of 2940 plasma proteins using the Olink

Explore platform. Cis-pQTLs were defined as SNPs located within a 1 Mb radius from the transcription start site (TSS) of the gene encoding the protein. Only index cis-pQTLs associated with plasma protein levels at a genome-wide significance threshold ($P < 5 \times 10^{-8}$) were considered for inclusion in the SMR analysis. SMR is designed to prioritize genes for which expression levels are potentially causally linked to an outcome trait, utilizing summary statistics within a Mendelian Randomization framework. To differentiate between pleiotropy and linkage (where protein abundance and a phenotype manifestation could be influenced by two separate causal variants in strong linkage disequilibrium with one another), the Heterogeneity in Dependent Instrument (HEIDI) test was employed. A HEIDI test *P* value below 0.01 signifies the presence of two distinct genetic variants in high linkage disequilibrium, explaining the observed associations. Additionally, to address potential biases from analyzing single SNPs, a multi-SNP approach (multi-SNPs-SMR) was utilized as a sensitivity analysis, enhancing the reliability of the statistical evidence. A *P* value less than 0.05 in the multi-SNPs-SMR analysis was deemed significant. Furthermore, HyPrColoc analysis was employed to ascertain whether the associations identified between proteins and various diseases stemmed from the same causal variant or were due to linkage disequilibrium. A posterior probability of a shared causal variant (PP.H4) greater than 0.7 signifies evidence of colocalization between proteins and multiple diseases.

### Reporting summary

Further information on research design is available in the Nature Portfolio Reporting Summary linked to this article.

## Data availability

All analyses in this study were conducted using publicly available datasets. Download links for publicly available GWAS summary statistics used as inputs in this study are provided in Supplementary Data 1. The study used only openly available GWAS summary statistics on leukocyte telomere length and six major cardiovascular diseases that have originally been conducted using human data. GWAS summary statistics on LTL are available at https://figshare.com/s/caa99dc0f76d62990195. Genome-wide summary statistics for AF, HF, and stroke are available from the GWAS Catalog under accession codes GCST006414, GCST009541, and GCST90104539, respectively. GWAS summary statistics for CAD and PAD are publicly available for download from the Cardiovascular Disease Knowledge Portal (CVDKP) at https://cvd.hugeamp.org/datasets.html. Genome-wide summary statistics for VTE were obtained from the deCODE Genetics website: https://www.decode.com/summary data/. Blood-based cis-pQTL from UKB-PPP are obtained from https://www.synapse.org/ Synapse:syn51365303. Source data used for generating the figures are available in the Source Data file with this paper. Source data are provided with this paper.

## Code availability

All software (and version, where applicable) used to conduct the analyses in this paper are freely available online: LDSC (v1.0.1; https://github.com/bulik/ldsc), MiXeR (v1.3; https://github.com/precimed/mixer), LAVA (v0.1.0; https://github.com/josefin-werme/LAVA), LCV (https://github.com/lukejoconnor/ LCV), MRlap (v0.0.3; https://github.com/n-mounier/MRlap), TwoSampleMR (v0.6.8; https://mrcieu.github.io/TwoSampleMR), PLACO (v0.1.1; https://github.com/RayDebashree/PLACO), FUMA (v1.5.4; http://fuma.ctglab.nl/), HyPrColoc(v1.0; https://github.com/jrs95/hyprcoloc), MAGMA (v.1.08; https://ctg.cncr.nl/ software/magma), e-MAGMA (https://github.com/eskederks/eMAGMA-tutorial), TWAS (http://gusevlab.org/projects/fusion/), SMR (v1.31; https://yanglab.westlake.edu.cn/ software/smr/), COLOC (v5.2.1; https://github.com/chr1swallace/coloc), and R (v.4.1.3; https://www.r-project.org/).

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

## Acknowledgements

This work was funded by the Natural Science Foundation of China Excellent Young Scientists Fund (Overseas) (Grant No. K241141101), Guangdong Basic and Applied Basic Research Foundation for Distinguished Young Scholars (Grant No. 2024B1515020047), Shenzhen Basic Research General Projects of Shenzhen Science and Technology Innovation Commission (Grant No. JCYJ20230807093514029), National Natural Science Foundation of China (Grant No. 82470452), Natural Science Foundation of Xinjiang Uygur Autonomous Region (Grant No. 2024D01D15) (To Y.F.); National Natural Science Foundation of China (Grant No. 82300315), Guangdong Province Basic and Applied Basic Research Fund Project (Grant No. 2024A1515012174; 2025A1515012690), Research Project of Guangdong Provincial Bureau of Traditional Chinese Medicine (Grant No. 20241120), Excellent Young Talents Program of Guangdong Provincial Hospital of Traditional Chinese Medicine (Grant No. SZ2024QN05) and Basic Clinical Collaborative Innovation Program of Guangdong Provincial Hospital of Traditional Chinese Medicine and School of Biomedical Sciences, The Chinese University of Hong Kong (Grant No. YN2024HK01), Noncommunicable Chronic Diseases-National Science and Technology Major Project (Grant No. 2024ZD0528206; 2024ZD0528200) (For R.Z.); National Natural Science Foundation (Grant no. 82170339 and 82270241), NSFC Incubation Project of Guangdong Provincial People's Hospital (Grant no. KY0120220021), Natural Science Foundation of Guangdong Province (Grant no. 2023B1515020082) (To L.J.); Cancer Research UK Career Development Fellowship (Grant no. C59392/A25064) (To S.P.), National Natural Science Foundation of China (Grant no. 82260073); Tianshan Talent Cultivation Program Project of Xinjiang Uygur Autonomous Region (Grant no. 2022TSYCLJ0028) (To Y.Y.), Shenzhen Science and Technology Program, Shenzhen, China (Grant No. GJHZ20240218111401002), National Science and Technology Major Project (Grant no. 2023ZD0505902) (To Jin-Song Bian), and Center for Computational Science and Engineering at Southern University of Science and Technology. The funder had no role in the design, implementation, analysis, interpretation of the data, approval of the manuscript, and decision to submit the manuscript for publication.

## Author contributions

J.Q., Y.F., Y.Y., R.Z., S.P., and L.J. conceptualized and supervised this project and wrote the manuscript. J.Q., Q.W., and Y.Z. performed the main analyses and wrote the manuscript. J.Q., M.C. (Minjing Chang), S.S., P.Z., K.Y., M.C. (Miaoran Chen), L.Z., and X.X. performed the statistical analysis and assisted with interpreting the results. L.C. and A.J. provided expertise in cardiovascular biology and GWAS summary statistics. All authors discussed the results and commented on the paper.

## Competing interests

The authors declare no competing interests.

## Additional information

[1]Department of Pharmacology, SUSTech Homeostatic Medicine Institute, School of Medicine, Southern University of Science and Technology, Shenzhen, China. [2]Department of Geriatrics, Shenzhen People's Hospital (The First Affiliated Hospital of Southern University of Science and Technology), Shenzhen, China. [3]Joint Laboratory of Guangdong-Hong Kong Universities for Vascular Homeostasis and Diseases, Shenzhen, China. [4]Department of Nephrology, The Fifth Clinical Medical College of Shanxi Medical University, Taiyuan, China. [5]Department of Rheumatology, Shanxi Key Laboratory of Immunomicroecology, Second Hospital of Shanxi Medical University, Taiyuan, China. [6]School of Public Health and Emergency Management, Southern University of Science and Technology, Shenzhen, China. [7]Department of Cardiology, Guangdong Cardiovascular Institute, Guangdong Provincial People's Hospital (Guangdong Academy of Medical Sciences), Southern Medical University, Guangzhou, China. [8]Department of Nephrology, Shanxi Kidney Disease Institute, Second Hospital of Shanxi Medical University, Taiyuan, China. [9]Department of Otolaryngology-Head and Neck Surgery, Second Hospital of Shanxi Medical University, Taiyuan, China. [10]Department of Pediatrics, University of Cincinnati College of Medicine, Cincinnati, USA. [11]Department of Computer Science, University of Cincinnati College of Engineering, Cincinnati, USA. [12]Department of Internal Medicine, College of Medicine, University of Cincinnati, Cincinnati, USA. [13]Botnar Research Centre, Nuffield Department of Orthopaedics, Rheumatology and Musculoskeletal Sciences, University of Oxford, Headington, Oxford, UK. [14]State Key Laboratory of Traditional Chinese Medicine Syndrome, Guangdong Provincial Hospital of Chinese Medicine, Guangdong, China. [15]State Key Laboratory of Dampness Syndrome of Chinese Medicine, Guangzhou, China. [16]Guangdong Provincial Key Laboratory of TCM Emergency Research, Guangzhou, China. [17]Department of Cardiology, People's Hospital of Xinjiang Uygur Autonomous Region, Urumqi, China. [18]Xinjiang Key Laboratory of Cardiovascular Homeostasis and Regeneration Research, Urumqi, China. [19]These authors contributed equally: Jun Qiao, Qian Wang, Yuhui Zhao, Minjing Chang, Shuo Sun. ✉e-mail: jianglei0731@gmail.com; siim.pauklin@ndorms.ox.ac.uk; zourj3@mail2.sysu.edu.cn; yangyn5126@xjrmyy.com; fengyl@sustech.edu.cn

