## [Transparent Peer Review file · Nature Communications]

Contribution of leukocyte telomere length to cardiovascular disease onset from genome-wide cross-trait analysis

Corresponding Author: Professor Siim Pauklin

Version 0:

Reviewer comments:

Reviewer #1

(Remarks to the Author)

Qiao and colleagues investigated the link between telomere shortening and several CVD traits using genetic correlation analysis and Mendelian randomization. The research question is interesting. However, the results are presented superficially without a clear purpose or importance of the analyses performed in addressing the research question.

Major comments

1. Both ITL and CVDs are age related, the authors should describe how this is handled in their genetic correlation analysis.
2. The authors mentioned within the results section that “Overall, MR analysis revealed that the 250 shared genetic foundation between LTL and CVDs cannot be ascribed to vertical pleiotropy”. The authors should consider providing more details in the results section to convince the reader that their statement is evidence-based. Also, how do the authors interpret their findings of the causal impact of CAD on LTL?
3. It is not clear how the authors ensured that their MR analyses were robust. There is no mention of tests such as MR-Egger anywhere in describing the results using the MR analysis.
4. Within the FUMA analysis, the authors continuously mention the number of variants linked to CVD as well as ITL. This is, however, not quite informative or interesting. The authors should consider providing approaches that provide information on genetic susceptibilities that link ITL and CVDs.
5. There is no coherent approach within the results section to provide any insight into the potential mechanisms or genetic architecture that could link ITL and CVDs. Results are superficial without providing any specific details of any important genes or biological pathways for any specific CVD trait.
6. In line 278 without any prior context, the authors introduce eQTLs. It is not clear as part of which analysis and to which purpose, the eQTL analysis has been performed.
7. In line 296, it is not clear how, why and for which variants the colocalization analysis was performed.
8. It is not clear how the many different analyses in this work come together.
9. The authors should consider providing details of the methods in relation to LDSC including the software and their data sources.
10. The authors should discuss how the results could help CVD patients.
11. The authors should consider providing an appropriate interpretation for the findings of the MiXeR analysis and the latent causal variable (LCV) analysis.

Minor comments

1. Throughout the paper and especially within the results section, there are plenty of abbreviations that have not been defined before.
2. Within the results section, the authors should briefly describe the purpose of the MiXeR, LHC-MR, ANNOVAR, HyPrColoc, LDSC-SEG, and PLACO analyses.
3. It is not clear how the input data was selected in the MAGMA analysis.

Reviewer #2

(Remarks to the Author)

Jun Qiao et al have performed genetic correlation analyses and further pleiotropic effect and causality evaluations between LTL and CVD outcomes. Results suggest no causality between LTL and these CVD outcomes but rather extensive genetic pleiotropy between LTL and CVD outcomes. They highlight on numerous potential genes and pathways that may underlie this pleiotropy.

Some clarifications and suggestions.

1) The analyses, particularly the analyses of pleiotropic effects, was extensive. But, it is not clear or hard to see what are the consistent results or "consensus" among these different results. And, it would also be important to comment on the "inconsistence" among some of these results.

2) Furthermore, in the Pleiotropic genomic loci identified for LTL and CVDs and Pleiotropic genes associated with LTL and multiple CVDs sections the authors attempt to connect the identified SNPs to potential functional genes for pathway-based evaluations. While it is very difficult to make this connection without extensive and detailed functional data relevant for both TL attrition and CVDs and the authors attempted to evaluate multiple scenarios (closes genes through MAGMA analysis, inclusion of QTL and other tissue level mics data), it makes the results difficult to crystallize. Would suggest including only the results of genes from either the overlapping list (MAGMA and E-MAGMA) or only MAGMA based results (it has been suggested recently that usually the closest gene is likely functional gene from GWAS studies and using the closest gene to the SNP perform relatively well in subsequent pathway evaluations).

3) As the authors highlight, it is well known that reduced LTL is an increased risk for various CVDs. Given this negative correlation would it be necessary to perform subsequent pathway-based enrichment analyses separately for variants that follow this known discordant relationship (overlapping variants that decrease LTL and also increase CVD risks) from variants that show a positive correlation (overlapping variants that either both decrease or increase LTL and CVD risks).

4) Following on this, would there be subsets of variants (and subsequent dysfunctional pathways) that follow this known negative correlation for a particular trait pair (i.e LTL – CAD pair) but not is another pair (i.e LTL – STROKE pair).

5) The authors indicate that additional East-Asian GWAS summary data was utilized for replications, but it is not clear which datasets were used and which results were replicated in the results section.

6) The authors correctly highlighted issues with previous studies, especially causal inference studies, that may have a level of bias due to sample overlap between traits evaluated. Would the causal inference studies performed in the present study in The Causal inference between LTL and six major CVDs and Shared causal proteins between LTL and six major CVDs sections also be similarly affected by some level of bias due to the extensive overlap of the UKBiobank data for LTL, CVDs, proteomics and other traits. This would be especially important to suggest that lack of causality between these traits.

7) Previous work by the ENAGAGE consortium have utilized similar datasets to suggest that insulin and potential glucose metabolism may be a strong mediator between LTL and CVD (PMID: 28515044 DOI: 10.1161/CIRCRESAHA.116.310517). It was quite striking that this was not picked up strongly in the present study. Could the authors evaluate if similar genes were identified in the present study and discuss their findings in relation to these previous data?

Reviewer #3

(Remarks to the Author)

In this paper Qiao et al. throw lots of methods at a complex question investigating a shared genetic biology of telomere length and the risk of six major cardiovascular diseases (CVDs). The authors focus on an investigation of horizontal pleiotropic effects that are likely to reveal shared biology but does not manage to create a clear narrative when exploring the established TL CVD relationship.

Due to the sheer number of methods used a figure detailing the strategy and logical process through these analyses would help to elucidate the meaning for some of these tests and their findings, which is currently a bit lost within the text. It is my impression that there are two strategies. The first being correlation leading to causal inference. The second being colocalization leading to gene/protein set analyses. I believe both have serious flaws.

Performed LDSC regression to look at rg. Looking at results in the supplement the authors found no significant evidence of genome-wide correlation with all but PAD, which was only nominally associated ($p=0.047$) in East Asians (s. table 1). The analysis presented in the rest of the paper is for Europeans only where there was evidence of a genetic correlation with LTL in all but AF. S Table 1 does not show the studies included in the presented analysis. This contradictory information puts the strategy in place in a weak position as it would appear results weren't as expected in East Asians so the analysis side-stepped to Europeans. These genetic correlations are also not novel.

The LTL data from UK Biobank are of predominantly European ancestry but, importantly, also include non-Europeans. The same is true of data for many of the CVD GWAS included here and without the information in ST1 it is unclear which sample sizes were used in this analysis to know if they are European only (I assume not). This has an impact on many analyses, such as those utilising the 1000g EU reference panel data.

Then look at MiXeR that aims to look at overlapping 'causal' variants that is not bound by a constant shared direction of effect. This highlighted an overlap of signal at numerous locations thus supporting an analysis of specific loci. LAVA was then applied and HyPrColoc. The threshold for PP of 0.7 is liberal and non-conclusive but suggestive that there may be a pleiotropic loci present.

ATXN2 loci came up as linked to all CVDs using HyPrColoc. Though this is also linked to blood pressure and lipids suggesting a pleiotropic relationship larger than that which is analysed here as LTL also links to these cardiovascular risk factors.

Causal analyses are highly limited and attempt to address the difference between a direct and indirect causal association in a bi-directional approach that fails to convince in the findings of no association. This is due to the nature of a genome-wide instrument (importantly, this is not detailed here) and goes against current knowledge. Without detail it makes it very hard to believe. Comprehensive evidence linking TL to CVDs utilising non-pleiotropic instruments were shown in the publication where the TL data come from. Again, supplementary tables are presented for a different East Asian analysis that makes it impossible to assess this analysis.

Authors heavily utilised genetic correlation methods, and to show "causal" part chose latent causal variable (LCV) analysis and Latent Heritable Confounder Mendelian Randomization (LHC-MR). Both LCV and LHC-MR use whole-genome summary statistics to explore bi-directional causal associations between complex traits and yield effect estimates accounting for bi-directional genetic effects, direct heritability, and confounder effects. Unfortunately, in LCV and LHC-MR settings it is difficult to assess which genetic variants are potentially causal, which have direct, indirect, or null effects, which are biologically plausible. Moreover, LHC-MR is a relatively new method and might not perform as well as advertised (PMID: 35576237). Using LCV and LHC-MR on the whole-genome may incorporate information that may not be relevant in association of just two traits – potentially, they may be correlated via some other traits, Thus the first strategy arm, correlation to causal inference, struggles to convince in the final outcome.

The authors, on lines 268-271, state it as remarkable that 47.2% of the loci have concordant effects that increase LTL and decrease CVD risk yet this is entirely within the understood relationship between the traits.

The evidence supporting signals as causal for both is not present, as shown in the colocalization results where very few of these signals (only 22/248) pass a too liberal threshold of 0.7. As a combined score for 2 traits it appears evident that the signals are predominantly driven by 1 of the traits and likely not truly overlapping signals.

This is where the second strategy arm, coloc to gene/protein sets, fails to convince as all downstream analyses were then performed on all loci despite no evidence of a colocalized signal for the huge majority. This makes the interpretation of any output wildly speculative that does not reflect the hypothesis of a shared biology and combines genes from different traits together with no strong justification that these genes are shared.

The FUMA analysis only considered results from the PLACO analysis novel if not found in the single trait GWAS. This is counterintuitive given the hypothesis that the variant/loci is linked to both traits through shared biology. Colocalization $PP > 0.7$ is only suggestive and is not 'strong evidence' (as stated in line 300), reserved for more stringent criteria > 0.9 .

In identifying SH2B3 as a shared 'causal' protein and ATXN2 as a shared pleiotropic loci I am reminded that these are 2 top key drivers of hypertension (PMID:25882670) and wonder if they are simply 2 genes involved in other complex pleiotropic associations that are unrelated to the LTL CVD relationship and these are simply shadows of other mechanisms.

Missing reference 16, where shorter genetic TL and its association with CVDs is discussed. Possibly PMID: 32109421.

In conclusion, lots of analyses are performed that do not create a narrative of clear shared biology and through their own investigation do not see a causal relationship. I am open to the concept of other factors linking TL to CVD and TL being a surrogate but these analyses ventured through analysis toolkits that fall down at key points (no observed colocalization and no detail on key MR analyses) and is thus certainly not 'proved' as stated in the abstract and conclusion.

Reviewer #4

(Remarks to the Author)

Version 1:

Reviewer comments:

Reviewer #1

(Remarks to the Author)

1. The introduction is still out of focus and contains multiple unnecessary, nearly out of context content such as "Hayflick limit", "fibrous cap", "matrix degradation" that are never used again anywhere in the manuscript and do not help the flow of the argument. The authors should consider rewriting a more to-the-point Introduction to ease the flow of the section.

2. An example problem within the Introduction is the following sentence “These changes contribute to the thinning of the fibrous cap, thereby heightening the risk of CVDs.” The authors should provide context into what fibrous cap is and why its thinning increases risk of CVD.
3. The Introduction superficially presents various arguments and jumps back and forth between pure biology and data science studies without an obvious logic that links these different arguments. Thus create a disjointed and unpleasant reading experience.
4. In another place within the Introduction section, the authors mention “although the relationship with other CVDs remains more ambiguous”. What other CVDs are considered and what is the reason for this ambiguity? And what the authors exactly mean by ambiguous? Is it because these “Other CVDs” have not been studied, or the results were not exactly conclusive, and if so why?
5. Within the Introduction, lines 99 to 104 provides several claims without mentioning a scientific source.
6. The authors stated: “This body of evidence highlighted the potential impact of LTL on the development and progression of various CVDs “. The evidence is presented vaguely and is a brief mention of previous meta-analyses linking ITL and CVD. The authors do not present how these meta-analyses have an impact of ITL on both development and progression of ITL. The authors should provide clear, relevant, explicit evidence for their claims within the Introduction section.
7. Towards the end of the first paragraph within the Introduction section, the authors introduce a new angle of “inflammation and oxidative stress” that is presented superficially and vague making it a difficult reading experience and hard to pinpoint what the authors are suggesting as their central research question.
8. In the second paragraph of the Introduction, the authors start by repeating the same arguments as the first paragraph, this time instead of meta-analyses linking ITL and CVD, they vaguely and broadly mention epidemiological studies linking ITL and CVD. Without presenting these epidemiological studies, the authors start discussing genetics as the explanation for this link and briefly mention various genetic dimensions such as GWAs, vertical and horizontal pleiotropy. Yet at this point, it is unclear what a central research question is. All these issues and unclarity remain unaddressed from the previous version and the authors should consider a serious revision of this section to improve the readability, flow, and clarity.
9. In line 134, please explain the study by Ging et al. and its relevance to the research question. Also ensure to clarify your central research question. What are you seeking to study between ITL and CVD and why? This is unclear throughout the Introduction section.
10. In line 136, the authors mention: "Therefore, in light of the widespread associations reported in epidemiological studies and the existing evidence of a shared genetic basis between LTL and CVDs, the necessity for a comprehensive, large-scale analysis is clear." large scale analysis on what topic precisely? Please explicitly state what large-scale analysis you are seeking to perform and to prove what hypothesis?
11. The six major CVDs mentioned in line 153, should be explicitly listed in brackets.
12. The abbreviation VTE in line 168 is not previously defined. Also PAD in line 155.
13. Line 161-164 states: “Although genome-wide rg offered valuable insight into the genetic overlap between phenotypes, it could not distinguish genetic overlap resulting from a mixture of concordant and discordant effects from the absence of genetic overlap, potentially yielding an estimated rg near zero in both scenarios. “ which two scenarios the authors are referring to?
14. In line 170, the authors mentioned “These variants are referred to as ‘causal’ variants”. It is not clear what “these” refer to. Also a citation might be necessary here for any criteria used to refer to a variant a “causal variant”.
15. In line 172, the authors mention “This approach provides a clearer picture of localized correlations, which might otherwise cancel each other out at a genome-wide level.”. What do the author mean by “cancel each other out”. The authors should be explicit about the phrase “cancel each other out”.
16. In line 173, the authors should state the input and output used in the MiXeR analysis.
17. Explain calculation of the Dice Coefficient under the methods section.
18. In line 181, the authors should present the results that revealed distinct patterns of polygenic overlap. What were these patterns of polygenic overlap and how was it concluded that they were distinct.
19. Sometimes the authors use one and sometimes 2 decimal points within the manuscript.
20. In line 382, the authors mentioned we successfully replicated 1,117 genes (60.57%) using FUMA eQTL mapping. What is the outcome variable for these replications? Did the author perform any meta-analysis? Was there any overlap between the original cohort where these genes were found and the UKbiobank?
21. In line 289, the authors say: “289 pleiotropic genes (207 unique) were jointly identified by MAGMA and e-MAGMA analysis” in what sense these genes were unique? If you mean that they were only found in association with one trait, then pleiotropy doesn't make sense.

Reviewer #2

(Remarks to the Author)

Most of my comments have been addressed. Particularly, they have 1) acknowledged potential bias in MR due to sample overlaps as a limitation in the discussion section (and used newer methods to account for these bias), 2) performed coloc with glucose related traits to highlight a ATXN2 variant as a shared genetic locus for LTL, CVD and glycemic traits (this will be a validation of previous work by ENGAGE consortium), 3) added in the replications using East-Asian data in supplementary note and 4) showed that if they focused on overlapping genes from Magma and e-magma gene based evaluations, they will have a smaller set of 50 genes that explain much of the pleiotropic effects between the LTL-CVD trait pairs. And, they have modified the conclusions slightly, stating that there is some level of causal effects and vertical pleiotropy between LTL-CVD traits (mainly CAD), however much of these associations are due to horizontal pleiotropic effects.

Reviewer #3

(Remarks to the Author)

Authors made several adjustments, improved clarity and performed additional analyses. This improves the paper a lot. However, due to a broad and quite liberal approach that includes all possible connections despite the weak signals the strength of the evidence linking these traits is still a bit lost to me. The authors have kindly added a figure (Supplementary Figure 1) that somewhat details the research questions and methods applied for each this is helpful and helps to improve the readers understanding of the authors strategy. However, the results are hard to follow and, while presented in detail (Figures and Supplementary tables), I believe that the paper lacks clarity in a final summary of findings that would create the authors stated "atlas of the shared genetic associations". One suggestion for improvement is a figure or a table that summarises the key findings making clear connections between the results, i.e. the most important genetic variants/genes alongside the methods they were detected and confirmed with.

The reference you give in Supp Table 1 for the LTL GWAS was run on the whole of UKB where multiple ethnicities were included. Therefore you have not "selected exclusively the European subset of the LTL and CVD GWAS data". Please correct this in the text.

Your colocalisation at ATXN2 justifies the previous comment that this locus is involved in other mechanisms and is therefore pleiotropic. Perhaps this should be made clearer in the text as a limitation, that the ATXN2 loci (and similar) that are linked to multiple CVDs, blood pressure, and lipids, may have larger pleiotropic relationships that are not covered in this study.

I believe you are overstating the impact of sample overlap, especially when there is good evidence that this does not introduce bias in two-sample MR methods (except for MR Egger) when using Biobank sized GWAS (Minnelli et al. PMID: 33899104).

Supplementary Table 6 shows your MR analysis with LTL as the exposure uses an instrument with only 51 SNPs associated with LTL. The referenced paper these data come from generated an exact conditionally independent instrument that excluded pleiotropic variants and contained 130 variants. Why is this instrument not being used?

I appreciate the argument defending the use of PLACO despite weak evidence being based on a published strategy. I would still comment, without requesting a response, that just because it wasn't picked up for a previous paper it doesn't make the strategy correct.

I thank the authors for their detailed response to my previous comment (R3.10), it was very helpful.

Reviewer #4

(Remarks to the Author)

Version 2:

Reviewer comments:

Reviewer #1

(Remarks to the Author)

The authors have made considerable improvement in their manuscript and have addressed all my comments. I have no further comments.

Reviewer #3

(Remarks to the Author)

We thank the authors for their considered responses and believe the paper has greatly improved. We have a small number of additional minor comments.

1) In response to R3.4 the authors re-performed some MR analysis to show the effects of overlap bias, and in lines 513-527 make a point about sample overlap, however, most of the previous papers they used to source genetic instruments were based on large datasets suggesting sample overlap bias is highly unlikely and thus this point has little value.

2) Lines 144-146 Previous MR studies do not contain the reference to the key paper by Codd et al. 2021 (your ref 15, from where the instrument was taken, where several CVDs have been assessed). It should also add a reference to Li et al. (PMID: 32109421); they use an instrument from an entirely independent meta-GWAS (also showing several CVD MR

estimates).

3) Line 531 suggests that all previous research on TL and CVDs is overstated - perhaps this should be supported with additional references.

4) The authors have conducted HyPrColoc but we are unable to find the results, which should be added to Supplementary Data and referenced in the paper where appropriate.

Reviewer #4

(Remarks to the Author)

**Re: NCOMMS-24-58910**

**Title: Contribution of leukocyte telomere length to major cardiovascular diseases onset: phenotypic**
**and genetic insights from a large-scale genome-wide cross-trait analysis**

We thank the Reviewers and the Editor for the helpful feedback and insightful suggestions. The response
from the reviewers is encouraging and constructive. Below, we provide point-by-point responses that
address all the comments with additional analyses. By incorporating these new data and findings, our
manuscript has been improved substantially.

-----

**Reviewer #1:**

Qiao and colleagues investigated the link between telomere shortening and several CVD traits using genetic
correlation analysis and Mendelian randomization. The research question is interesting. However, the
results are presented superficially without a clear purpose or importance of the analyses performed in
addressing the research question.

**Major comments**

1. Both LTL and CVDs are age related, the authors should describe how this is handled in their genetic
correlation analysis.

*Re: Thank you for your insightful comment. We acknowledge that both LTL and CVDs are age-related*
*traits. However, the GWAS summary statistics used in our study have already undergone rigorous quality*
*control and age adjustment in their original analyses, ensuring that age-related confounding effects have*
*been appropriately accounted for. Specifically, all included GWAS datasets, such as those for LTL, AF,*
*CAD, VTE, HF, PAD, and Stroke, adjusted for age and sex as covariates using standard statistical methods.*
*Additionally, population structure was controlled using principal component analysis across all studies.*
*Given these rigorous adjustments, further age correction in our genetic correlation analysis was unnecessary.*

2. The authors mentioned within the results section that “Overall, MR analysis revealed that the shared
genetic foundation between LTL and CVDs cannot be ascribed to vertical pleiotropy”. The authors should
consider providing more details in the results section to convince the reader that their statement is evidence-
based. Also, how do the authors interpret their findings of the causal impact of CAD on LTL?

Re: We thank the reviewer for the insightful comment. First, we would like to clarify an error. In our
analysis of the causal relationship between LTL and six major CVDs, we considered the potential effects
of sample overlap. To address this, we previously employed LCV and LHC-MR methods, rather than
relying on traditional MR approaches. In response to Reviewer 3's comments, we further investigated the
robustness of LHC-MR by comparing causal inferences using GWAS data with and without overlapping
samples. Our findings revealed that the results obtained from LHC-MR were not robust and may not be
suitable for causal inference in the presence of sample overlap. To mitigate potential biases from sample
overlap, we further conducted an inverse variance weighting (IVW) Mendelian randomization analysis
using the MRlap approach (Mounier *et al.*, 2023). MRlap corrects for biases arising from sample overlap,
weak instruments, and winner's curses, offering more reliable causal estimates. The MRlap analysis
confirmed the previously identified negative causal relationship between LTL and CAD, aligning with the
results of traditional MR analysis. Moreover, there was a high degree of agreement between MRlap and
LCV results, with both methods inferring causality in a manner consistent with existing epidemiological
evidence. Based on these findings, we considered the MRlap conclusions to be the primary outcome in the
sensitivity analysis of LCV results. In contrast, LHC-MR appeared to overestimate reverse causality,
leading to inconsistent results.

Our study aimed to explore the shared genetic basis between LTL and CVDs, considering both horizontal
and vertical pleiotropy. Horizontal pleiotropy occurs when a genetic variant influences multiple traits
independently, while vertical pleiotropy arises when one phenotype causally affects another—a key aspect
addressed by MR. Our MR results indicated that LTL exerts only a weak negative causal effect on CAD,
suggesting that vertical pleiotropy accounts for only a small fraction of the shared genetic basis between
LTL and CVDs, with horizontal pleiotropy being the predominant contributor. This supports our statement
that vertical pleiotropy cannot primarily explain the shared genetic basis.

Moreover, multiple biological mechanisms support this relationship regarding the causal impact of CAD
on LTL. Inflammation and oxidative stress, key drivers of atherosclerosis, are known to accelerate telomere
shortening. Telomerase activity, which helps maintain telomere length, is downregulated in vascular
smooth muscle cells within atherosclerotic plaques (Hägg *et al.*, 2018; Farzaneh-Far *et al.*, 2008).
Furthermore, increased reactive oxygen species (ROS) levels in CAD-affected arterial walls contribute to
DNA damage, including strand breaks and base modifications, leading to premature telomere attrition
(Matthews *et al.*, 2006).

We will then discuss the potential biological mechanisms underlying the causal effect of LTL on CAD, as
outlined below. Telomere shortening may contribute to atherosclerosis through multiple biological aging
pathways. One key mechanism is the accumulation of senescent cells, which is a hallmark of atherosclerotic
plaques. These senescent cells reduce the regenerative capacity of affected tissues and promote apoptosis,
thereby exacerbating inflammation and endothelial dysfunction. Both endothelial dysfunction and
increased vascular inflammation are critical events in the progression of atherosclerotic plaques (Minamino
*et al.*, 2002). Furthermore, cellular senescence diminishes the proliferative potential of vascular smooth
muscle cells, which weakens the fibrous cap and increases the instability of atherosclerotic plaques,
ultimately contributing to the development of CAD (Matthews *et al.*, 2006).

**Reference:**

Mounier N, Kutalik Z. Bias correction for inverse variance weighting Mendelian randomization. *Genet*
*Epidemiol.* 2023;47(4):314-331. doi:10.1002/gepi.22522

Hägg S. Telomere Length Dynamics and Atherosclerotic Disease. *Circ Res.* 2018 Feb 16;122(4):546-547.
doi: 10.1161/CIRCRESAHA.118.31256

Farzaneh-Far R, Cawthon RM, Na B, Browner WS, Schiller NB, Whooley MA. Prognostic value of
leukocyte telomere length in patients with stable coronary artery disease: data from the Heart and Soul
Study. *Arterioscler Thromb Vasc Biol.* 2008 Jul;28(7):1379-84. doi: 10.1161/ATVBAHA.108.167049

Matthews C, Gorenne I, Scott S, Figg N, Kirkpatrick P, Ritchie A, Goddard M, Bennett M. Vascular smooth
muscle cells undergo telomere-based senescence in human atherosclerosis: effects of telomerase and
oxidative stress. *Circ Res.* 2006 Jul 21;99(2):156-64. doi: 10.1161/01.RES.0000233315.38086.bc

Minamino T, Miyauchi H, Yoshida T, Ishida Y, Yoshida H, Komuro I. Endothelial cell senescence in
human atherosclerosis: role of telomere in endothelial dysfunction. *Circulation.* 2002 Apr 2;105(13):1541-
4. doi: 10.1161/01.cir.0000013836.85741.17

Matthews C, Gorenne I, Scott S, Figg N, Kirkpatrick P, Ritchie A, Goddard M, Bennett M. Vascular smooth
muscle cells undergo telomere-based senescence in human atherosclerosis: effects of telomerase and
oxidative stress. *Circ Res.* 2006 Jul 21;99(2):156-64. doi: 10.1161/01.RES.0000233315.38086.bc

3. It is not clear how the authors ensured that their MR analyses were robust. There is no mention of tests
such as MR-Egger anywhere in describing the results using the MR analysis.

Re: Thank you for your thoughtful and constructive feedback. To ensure the robustness of our causal
inferences, we performed a sensitivity analysis for pairs of CVDs that showed partial genetic causality using
the MRlap method. MRlap corrects for biases such as sample overlap, weak instruments, and winner's

courses, offering more reliable causal estimates. The method first calculates the observed MR-based effect
values and then applies corrections based on genetic covariance, computed through LDSC. By comparing
the results from the IVW method before and after correction, we can assess whether sample overlap
significantly affects the findings. If a significant impact is detected, the corrected P value is considered the
primary result. Moreover, to further validate the robustness of our findings, we conducted sensitivity
analyses using conventional MR methods (including MR-Egger). In summary, MRlap provides a more
refined causal analysis by mitigating the influence of sample overlap and genetic confounders, and the
consistency of our findings across multiple MR methods further supports their validity. (From Page 30, line
849 to line 861)

4. Within the FUMA analysis, the authors continuously mention the number of variants linked to CVD as
well as LTL. This is, however, not quite informative or interesting. The authors should consider providing
approaches that provide information on genetic susceptibilities that link ITL and CVDs.

Re: Thank you for your comment. We mainly used the FUMA platform to identify independent genomic
loci and functionally annotate the top pleiotropic SNPs linked to both LTL and CVDs. Specifically, among
the 248 independent genomic risk loci shared between LTL and CVDs, 188 loci exhibited pleiotropic
effects across multiple CVD traits, suggesting the involvement of shared biological pathways. Notably, 74
of these loci (39.36%) were located in 16 chromosomal regions with pleiotropic associations across more
than half of the examined trait pairs, indicating genetic hubs that may play a critical role in the LTL-CVD
interactions. Moreover, we identified 21 SNPs with potentially deleterious effects (CADD scores > 12.37),
indicating these variants may influence protein structure or function, potentially contributing to disease
mechanisms. Furthermore, 9 SNPs were mapped to transcription factor binding sites (RegulomeDB scores
1a/1d/1f), suggesting a regulatory role in gene expression. Most of the variants were located in intronic
(45.9%) and intergenic (27.0%) regions, implying they likely modulate gene regulation rather than directly
altering protein-coding sequences. These findings offer insights into the genetic susceptibilities linking LTL
and CVDs, revealing potential shared genetic mechanisms that influence both telomere maintenance and
cardiovascular health.

5. There is no coherent approach within the results section to provide any insight into the potential
mechanisms or genetic architecture that could link ITL and CVDs. Results are superficial without providing
any specific details of any important genes or biological pathways for any specific CVD trait.

Re: Thank you for your detailed and constructive comments. In this comprehensive pleiotropic association
study, we leveraged large-scale GWAS summary data and applied a variety of statistical genetic methods
to uncover the shared genetic architecture and mechanisms between LTL and CVDs. We began by
exploring genetic overlap at the genome-wide level, identifying pleiotropic associations between LTL and
CVDs using methods such as LDSC, MiXeR, and LAVA. These approaches enabled us to assess the
directionality of genetic correlations across loci, providing deeper insights into the complex genetic
interactions that go beyond traditional correlation measures. At the SNP level, we employed the PLACO
method to identify genome-wide significant pleiotropic SNPs shared between LTL and CVDs. This was
followed by FUMA annotation and colocalization analyses to pinpoint causal variants within these loci.
Moving to the gene level, we applied the MAGMA method to identify pleiotropic genes, followed by tissue-
specific enrichment analysis to determine which tissues are most relevant to these genetic interactions.
Additionally, we used e-MAGMA and TWAS to identify tissue-specific genes involved in both LTL and
CVDs. At the pathway level, we conducted pathway enrichment analysis using MAGMA and ToppFun to
explore relevant biological pathways.

A joint analysis using MAGMA and e-MAGMA identified 289 pleiotropic genes, of which 207 were unique.
Among these, 50 unique genes were detected in two or more trait pairs, providing further evidence of the
tissue specificity of these pleiotropic genes. Notably, Seven genes, namely *ALDH2*, *ACAD10*, *TMEM116*,
*SH2B3* (all located at 12q24.12), *TMED6* (16q22.1), *SERPINF1* (17p13.3), and *XPO7* (8p21.3), have
emerged as the most pleiotropic; they are thought to have important regulatory functions, influencing over
half of the trait pairs examined.

[revised manuscript text omitted]

this protein has only been found to be related to coronary atherosclerosis (Zhang S *et al.*, 2021), and the
specific mechanism is unclear and requires further research.

To better understand the biological processes underlying these pleiotropic loci, we performed gene-set
enrichment analysis using Gene Ontology biological processes pathways, focusing on genes identified by
MAGMA. This analysis revealed three significant gene sets: chromatin organization, negative regulation
of nucleobase-containing compound metabolic processes, and negative regulation of miRNA maturation.
Interestingly, no gene set showed significant enrichment across more than one trait. Among these, the genes
shared between LTL and AF were most strongly associated with the 'negative regulation of nucleobase-
containing compound metabolic processes.' Next, we used ToppFun to further analyze the gene sets
identified in both the MAGMA and e-MAGMA analyses, particularly those overlapping across multiple
trait pairs. This additional analysis revealed that most of the significant biological processes, except for
DNA biosynthetic processes, were directly related to telomere maintenance. These processes included
'telomere maintenance via telomerase,' 'telomere maintenance via telomere lengthening,' 'telomere capping,'
and 'telomere organization.' Interestingly, both the DNA biosynthetic processes and telomere maintenance
mechanisms identified in ToppFun were part of the lower layers of the 'nucleobase-containing compound
metabolic process' identified in the MAGMA gene-set analysis. This finding suggests a crucial role for
telomere-related biological pathways in the shared genetic etiology between LTL and CVDs. Specifically,
telomeres undergo shortening during repeated cell divisions, and when their length diminishes to a critical
point, the resultant genomic instability can lead to further genetic abnormalities that promote cell death or
apoptosis, which is a hallmark of cellular senescence. Estrogen, stress accumulation from oxidative damage,
hypertension, etc., are believed to significantly impact telomere homeostasis and contribute to the
development of CVDs (Sinner MF *et al.*, 2014). This finding bolsters the theory that progressive telomere
shortening contributes to the pathogenesis of age-related human diseases such as CVDs. Specifically,
Minano et al. documented the presence of vascular endothelial cells exhibiting age-related phenotypes
within human atherosclerotic lesions (Fuster JJ *et al.*, 2006). Excessive vascular smooth muscle cells are
stimulated to proliferate and migrate, leading to the growth of atherosclerosis. Besides, telomerase
activation and telomere maintenance are critical in increasing the proliferation and growth of vascular
smooth muscle cells. Activation through the telomerase reverse transcriptase component (TERT) extends
the lifespan of cultured vascular smooth muscle cells. Conversely, telomerase inhibition can extend the
lifespan and reduce the proliferation of cultured vascular smooth muscle cells, thereby decreasing the risk
of atherosclerosis (Minamino T *et al.*, 2002).

**Reference:**

Zhang J, Guo Y, Zhao X, et al. The role of aldehyde dehydrogenase 2 in cardiovascular disease. *Nat Rev*
*Cardiol.* 2023;20(7):495-509. doi:10.1038/s41569-023-00839-5

Chen CH, Sun L, Mochly-Rosen D. Mitochondrial aldehyde dehydrogenase and cardiac diseases.
*Cardiovasc Res.* 2010;88(1):51-57. doi:10.1093/cvr/cvq192

Lee JY, Lee BS, Shin DJ, et al. A genome-wide association study of a coronary artery disease risk variant.
*J Hum Genet.* 2013;58(3):120-126. doi:10.1038/jhg.2012.124

Islam T, Rahman MR, Khan A, Moni MA. Integration of Mendelian randomisation and systems biology
models to identify novel blood-based biomarkers for stroke. *J Biomed Inform.* 2023;141:104345.
doi:10.1016/j.jbi.2023.104345

Kuo CL, Joaquim M, Kuchel GA, et al. The Longevity-Associated SH2B3 (LNK) Genetic Variant: Selected
Aging Phenotypes in 379,758 Subjects [published correction appears in *J Gerontol A Biol Sci Med Sci.*
2020 Sep 16;75(9):1686. doi: 10.1093/gerona/glz268.]. *J Gerontol A Biol Sci Med Sci.* 2020;75(9):1656-
1662. doi:10.1093/gerona/glz191

Goswami SK, Ranjan P, Dutta RK, Verma SK. Management of inflammation in cardiovascular diseases.
*Pharmacol Res.* 2021;173:105912. doi:10.1016/j.phrs.2021.105912

Nevers T, Salvador AM, Velazquez F, et al. Th1 effector T cells selectively orchestrate cardiac fibrosis in
nonischemic heart failure. *J Exp Med.* 2017;214(11):3311-3329. doi:10.1084/jem.20161791

Flister MJ, Hoffman MJ, Lemke A, et al. SH2B3 Is a Genetic Determinant of Cardiac Inflammation and
Fibrosis. *Circ Cardiovasc Genet.* 2015;8(2):294-304. doi:10.1161/CIRCGENETICS.114.000527

Zhu X, Fang J, Jiang DS, et al. Exacerbating Pressure Overload-Induced Cardiac Hypertrophy: Novel Role
of Adaptor Molecule Src Homology 2-B3. *Hypertension.* 2015;66(3):571-581.
doi:10.1161/HYPERTENSIONAHA.115.05183

Nassour J, Schmidt TT, Karlseder J. Telomeres and Cancer: Resolving the Paradox. *Annu Rev Cancer Biol.*
2021;5(1):59-77. doi:10.1146/annurev-cancerbio-050420-023410

Zhang S, Dai H, Li W, et al. TMEM116 is required for lung cancer cell motility and metastasis through
PDK1 signaling pathway. *Cell Death Dis.* 2021;12(12):1086. Published 2021 Nov 16. doi:10.1038/s41419-
021-04369-1

Sinner MF, Tucker NR, Lunetta KL, et al. Integrating genetic, transcriptional, and functional analyses to
identify 5 novel genes for atrial fibrillation. *Circulation.* 2014;130(15):1225-1235.
doi:10.1161/CIRCULATIONAHA.114.009892

Fuster JJ, Andrés V. Telomere biology and cardiovascular disease. *Circ Res.* 2006;99(11):1167-1180.
doi:10.1161/01.RES.0000251281.00845.18

Minamino T, Miyauchi H, Yoshida T, Ishida Y, Yoshida H, Komuro I. Endothelial cell senescence in
human atherosclerosis: role of telomere in endothelial dysfunction. *Circulation*. 2002;105(13):1541-1544.
doi:10.1161/01.cir.0000013836.85741.17

6. In line 278 without any prior context, the authors introduce eQTLs. It is not clear as part of which analysis
and to which purpose, the eQTL analysis has been performed.

Re: Thank you for your insightful comment. In our analysis, we conducted eQTL mapping to complement
positional mapping, providing deeper insights into the regulatory roles of pleiotropic SNPs associated with
LTL and CVDs. FUMA offers two distinct mapping strategies: (i) positional mapping, which links SNPs
to genes based on their physical proximity, and (ii) eQTL mapping, which associates SNPs with genes
based on cis-eQTL relationships, even if the SNPs are not located near the gene. While positional mapping
plays a crucial role, it alone may be insufficient, particularly since many of the identified variants lie within
non-coding regions. In contrast, eQTL mapping provides valuable insights into how allelic variations at
specific SNPs can regulate gene expression in relevant tissues. Of the 248 shared loci identified, 50 index
SNPs were found to be eQTLs, highlighting their potential regulatory roles in gene expression. For example,
the index SNP rs1566452 at the 16q22.1 locus was associated with eQTL signals for the *WWP2* gene
(encoding an E3 ubiquitin ligase) in both coronary and tibial arteries. This suggests that the rs1566452 locus
regulates *WWP2* expression, providing evidence for its functional role in vascular smooth muscle cells.

7. In line 296, it is not clear how, why and for which variants the colocalization analysis was performed.

Re: I appreciate your thoughtful comments. We performed Bayesian colocalization analysis to identify
shared causal variants between LTL and CVDs at the 248 potential pleiotropic loci identified by FUMA.
COLOC evaluates five mutually exclusive hypotheses for each pair of traits at a locus: H0 posits no
association with either trait; H1 and H2 suggest an association with only one of the traits; H3 indicates that
both traits are associated due to different causal variants; and H4 implies a shared association for both traits
stemming from the same causal variant. The analysis was performed using default COLOC prior
probabilities: p_1 and p_2 , each set at 1×10^{-4} for an SNP's association with the first and second trait,
respectively, and p_{12} at 1×10^{-5} for an SNP associated with both traits. A Posterior Probability for
Hypothesis 4 (PP.H4) greater than 0.7 was considered strong evidence for colocalization, suggesting the
presence of shared causal variants at the locus. The SNP exhibiting the highest PP.H4 was identified as a
candidate causal variant. Finally, among the 248 genomic risk loci, we identified 22 loci with PP.H4 > 0.70,

indicating the presence of shared causal variants between LTL and CVDs. Additionally, 40 loci exhibited
$PP.H3 > 0.70$, suggesting that while both traits are associated, they are influenced by distinct causal variants.

8. It is not clear how the many different analyses in this work come together.

Re: We thank the reviewer for the comment. For the genome-wide level, we estimated the genetic
correlation (r_g) between LTL and CVDs using LDSC. A global r_g represents the average of the shared
association signals across the genome. However, it does not differentiate between (i) genetic overlap with
a mixture of concordant and discordant effects and (ii) an absence of genetic overlap, as both return an
estimate close to 0. To address this “missing dimension” of genetic overlap, the MiXeR was employed to
demonstrate extensive genetic overlap and mixed effect directions between LTL and CVDs. While global
r_g provides a broad overview, local genetic correlations offer a more effective approach for identifying and
capturing associations with mixed effect directions. We applied LAVA to perform local genetic correlations
($loc-r_{gs}$) between LTL and six major CVDs across 2,495 semi-independent genetic regions of
approximately equal size (~1 Mb). By charting the landscape of genetic overlap beyond r_g , we provide
insights into shared genetic architectures between LTL and six major CVDs.

These shared genetic bases reflect genetic variants that influence multiple complex phenotypic traits, which
can occur through vertical or horizontal pleiotropy. Horizontal pleiotropy arises when two phenotypes are
affected by the same genetic variant, shedding light on shared biological pathways linking complex traits.
In contrast, vertical pleiotropy occurs when a genetic variant influences one phenotype, which in turn causes
a secondary phenotype—this is the type of relationship primarily targeted by MR analysis. To distinguish
these mechanisms, we conducted extensive analyses of both horizontal and vertical pleiotropy, testing
potential causal genetic effects using LCV and MRlap methods and focusing on elucidating contributions
from vertical pleiotropy.

To capture horizontal pleiotropy, we applied various statistical tools to characterize shared loci and examine
their functional implications for genes, tissues, biological processes, and protein targets. At the SNP level,
PLACO was conducted to identify pleiotropic variants shared between LTL and CVDs, considering only
those reaching genome-wide significance ($P < 5 \times 10^{-8}$). FUMA was applied to characterize potential
pleiotropic loci, based on which a Bayesian colocalization analysis was performed to further identify shared
causal variants in each pleiotropic locus. At the gene level, MAGMA was used to identify candidate
pleiotropic genes. LDSC-SEG identified the tissue types specificity of these genes, which helped refine the
understanding of gene function and expression patterns. Furthermore, e-MAGMA and TWAS were applied

to identify pleiotropic genes specific to particular tissue types. At the pathway level, MAGMA gene-set
analysis was also conducted to explore the biological functions of these pleiotropic genes, testing 7,744
gene sets, including Gene Ontology biological processes from the MSigDB. Additionally, pathway
enrichment analysis of mapped genes was carried out using ToppFun. Finally, to elucidate the biological
underpinnings of the shared genetic predisposition to LTL and six major CVDs, we performed proteome-
wide MR and colocalization analyses to identify potential druggable targets.

In summary, this study addressed five central questions regarding the shared genetic basis of LTL and six
major CVDs: (i) Can we identify shared genetic architectures across LTL and CVDs? (ii) Are these shared
genetic architectures driven by causal associations, specifically vertical pleiotropy? (iii) Can we detect
genomic loci shared by LTL and multiple CVDs, and do some loci exhibit opposite allelic effects between
LTL and CVDs? (iv) Can we identify functional features of pleiotropic loci that explain their broad impact
on LTL and cardiovascular pathology? (v) Can we identify causal plasma proteins as therapeutic targets
affecting LTL and major CVDs?

9. The authors should consider providing details of the methods in relation to LDSC including the software
and their data sources.

Re: We thank the reviewer for the comment. We analyzed the r_g between LTL and six major CVDs using
LDSC. LDSC facilitates the estimation of the average genetic effect sharing across the entire genome
between two traits, leveraging GWAS summary statistics (available at <https://github.com/bulik/ldsc>). This
includes the contribution of SNPs below the threshold of genome-wide significance and accounts for
potential confounding factors such as polygenicity, sample overlap, and population stratification. This
analysis utilized pre-computed LD scores from the European reference panel in the 1000 Genomes Project
Phase 3, excluding SNPs that did not overlap with the reference panel. Notably, the MHC region (chr 6:
25-35 Mb), known for its intricate LD structure, was omitted from the main analysis. LDSC analysis
estimated the genetic correlations between LTL and the six major CVDs. This method utilizes a weighted
linear model, where it regresses the product of Z-statistics from two traits against the LD score across all
genetic variants genome-wide. Genetic correlations with P -values below the Bonferroni-adjusted threshold
($P = 0.05 / \text{number of trait pairs} = 0.05 / 6 = 8.33 \times 10^{-3}$) were deemed statistically significant.

10. The authors should discuss how the results could help CVD patients.

Re: Thank you for your insightful comment. Our study highlights the critical role of telomere maintenance
in the shared genetic etiology of LTL and CVDs, offering new potential avenues for therapeutic intervention.
Previous research has demonstrated that telomerase can counteract early plaque-associated vascular smooth
muscle cell (VSMC) senescence, enabling cell proliferation even when telomeres are critically short (Aung
N *et al.*, 2023; Chen B *et al.*, 2023). However, telomere shortening may also contribute to the accumulation
of senescent endothelial cells and VSMCs, exacerbating atherosclerosis and inflammation—two key drivers
of coronary heart disease. These findings underscore the potential for therapeutic strategies aimed at
preserving or restoring telomere length to mitigate cardiovascular risk. For example, pharmacological
interventions that maintain telomere integrity or enhance telomerase activity may help reduce CVD
susceptibility in high-risk individuals. Additionally, our findings suggest that LTL and its associated genetic
variants could serve as valuable biomarkers for early cardiovascular risk detection. Identifying individuals
with shorter telomeres or those carrying genetic variants linked to LTL may allow healthcare providers to
stratify patients at elevated risk for CVD before clinical symptoms emerge. This proactive approach could
enable earlier intervention strategies, such as lifestyle modifications, personalized risk management, or
more frequent monitoring, to slow or prevent disease progression. (From Page 23, line 640 to line 656)

**Reference:**

Aung N, Wang Q, van Duijvenboden S, Burns R, Stoma S, Raisi-Estabragh Z, Ahmet S, Allara E, Wood
385 A, Di Angelantonio E, Danesh J, Munroe PB, Young A, Harvey NC, Codd V, Nelson CP, Petersen SE,
Samani NJ. Association of Longer Leukocyte Telomere Length With Cardiac Size, Function, and Heart
Failure. *JAMA Cardiol.* 2023 Sep 1;8(9):808-815. doi: 10.1001/jamacardio.2023.2167

Chen B, Yan Y, Wang H, Xu J. Association between genetically determined telomere length and health-
related outcomes: A systematic review and meta-analysis of Mendelian randomization studies. *Aging Cell.*
2023 Jul;22(7):e13874. doi: 10.1111/acel.13874

11.The authors should consider providing an appropriate interpretation for the findings of the MiXeR
analysis and the latent causal variable (LCV) analysis.

Re: Thanks for the comment. We applied MiXeR to verify genetic overlap between LTL and each CVDs.
MiXeR estimates the total number of both shared and trait-specific causal variants (i.e., variants with
nonzero additive genetic effects on a given trait). To quantify the polygenic overlap, we used the Dice
coefficient (DC), which measures the proportion of shared SNPs between two traits relative to the total
number of SNPs associated with either trait. Bivariate MiXeR analysis revealed extensive genetic overlap
between LTL and CVDs, with notable variation across specific traits. The polygenic overlap between LTL

and PAD was particularly striking (Dice coefficient = 0.229, SD = 0.015). Additionally, MiXeR estimates
the effect directions of all shared 'causal' variants and quantifies the mixed genetic effects by calculating
the proportion of these variants with concordant effects. This proportion reflects the balance of genetic
effects between two traits, with values ranging from 0 (indicating completely discordant effects) to 1
(indicating completely concordant effects), and 0.5 representing an equal balance of both. Specifically, the
MiXeR-estimated proportion of shared 'causal' variants with concordant effects was 0.416 (SD = 0.029)
between LTL and AF, suggesting a balanced mixture of concordant and discordant genetic effects.

LCV analysis was used to infer potential causal relationships between LTL and CVDs, quantified by
estimating the GCP. GCP values range from 0 (indicating no causal relationship) to 1 (indicating a fully
causal relationship), with higher values suggesting a stronger partial causal effect. Notably, none of the trait
pairs showed evidence of partial genetic causation, as all fell below the stringent GCP threshold ($|GCP| >$
$0.6, P < 0.05 / \text{no. of trait pairs} = 0.05 / 6 = 8.33 \times 10^{-3}$). Under the less stringent threshold ($|GCP| > 0.4, P <$
0.05), we observed a weak negative causal relationship between LTL and CAD, that is, genetically
predicted shortening of LTL was associated with an increased incidence of CAD. Although the LCV
analysis is robust, it does not account for factors such as bidirectional causal associations. To address these
limitations, we further employed the MRlap approach to explore bidirectional causal associations between
LTL and CVDs. The MRlap analysis confirmed the previously identified negative causal associations
between LTL and CAD, which is also consistent with the results of traditional MR analysis. Overall, the
MRlap analysis confirmed the reproducibility of the partial causal associations between trait pairs and
indicated that the shared genetic basis between LTL and CVDs cannot be attributed entirely to vertical
pleiotropy. (From Page 10, line 254 to line 270)

**Minor comments**

1. Throughout the paper and especially within the results section, there are plenty of abbreviations that have
not been defined before.

Re: We apologize for the confusion. We have added explanations for the undefined abbreviations in the
revised manuscript and clarified their relevant meanings in advance.

2. Within the results section, the authors should briefly describe the purpose of the MiXeR, LHC-MR,
ANNOVAR, HyPrColoc, LDSC-SEG, and PLACO analyses.

Re: We thank the reviewer for this valuable suggestion. To enhance clarity, we will revise the results section
to include a brief description of the purpose of each analytical method mentioned. Specifically:

To better investigate polygenic overlap beyond genetic correlation, we used the causal mixture modeling
approach (MiXeR) to estimate the number of shared and unique trait-influencing variants necessary to
explain at least 90% of heritability for each trait. These variants are referred to as ‘causal’ variants. Unlike
methods that rely on genome-wide genetic correlation, MiXeR assesses overlap without considering the
direction of each variant’s effect. This approach provides a clearer picture of localized correlations, which
might otherwise cancel each other out at a genome-wide level. (From Page 7, line 171 to line 177)

While LCV estimates causal relationships between traits by leveraging shared genetic architecture, it does
not fully account for latent heritable confounders, which can introduce bias into causal estimates. To address
this limitation, we conducted an additional analysis using the MRlap method to explore the bidirectional
causal associations between LTL and CVDs. (From Page 10, line 261 to line 265)

ANNOVAR was used to identify the functional categories of the lead SNPs shared between LTL and CVDs
based on their locations with respect to genes, including exonic, intronic, 5’ untranslated region, 3’
untranslated region, upstream, downstream, and intergenic. (From Page 11, line 296 to line 298)

We further performed multi-trait colocalization analysis on each FUMA-annotated pleiotropic loci using
the HyPrColoc method to identify potential shared causal variants and provide the posterior probability of
colocalization. (From Page 12, line 325 to line 327)

To pinpoint tissues potentially integral to the biological processes of LTL and six major CVDs, we utilized
LDSC applied to specifically expressed genes (LDSC-SEG) for tissue-specific enrichment analysis using
single trait GWAS summary statistics. By linking genetic heritability to specific functional categories or
tissues, LDSC-SEG evaluates whether the heritability of LTL and CVDs is enriched in SNPs that are active
in specific tissues. (From Page 13, line 359 to line 363)

To identify pleiotropic SNPs within the union set of trait pairs exhibiting significant genetic correlation or
overlap, we applied pleiotropic analysis under composite null hypothesis (PLACO) method. (From Page
11, line 275 to line 277)

3. It is not clear how the input data was selected in the MAGMA analysis.

Re: Thank you for your comment. We mainly conducted the MAGMA analysis based on the PLACO results.

Additionally, we also compared it with MAGMA analyses based on original GWAS data, defining new

associations as genes that had not been reported in original GWAS results.

Reviewer #2 (Remarks to the Author):

Jun Qiao et al have performed genetic correlation analyses and further pleiotropic effect and causality
evaluations between LTL and CVD outcomes. Results suggest no causality between LTL and these CVD
outcomes but rather extensive genetic pleiotropy between LTL and CVD outcomes. They highlight on
numerous potential genes and pathways that may underlie this pleiotropy.

Some clarifications and suggestions.

1) The analyses, particularly the analyses of pleiotropic effects, was extensive. But, it is not clear or hard
to see what are the consistent results or "consensus" among these different results. And, it would also be
important to comment on the "inconsistence" among some of these results.

Re: Thank you for your insightful comments regarding the consistency and inconsistency of results,
especially in terms of the pleiotropic effects we observed.

Consistence results: In our study, we have made an effort to highlight the consistent findings across various
analyses, as they form the foundation of our conclusions. Our study utilized multiple levels of analysis—
SNP-level, gene-level, pathway-level, and protein-target-level analyses—which collectively supported the
consistency of our findings. Specifically, we identified 248 significant loci shared between LTL and CVDs,
such as 16q22.1, 12q24.12, 17p13.3, and 8p21.3. Gene-level analysis further validated several key genes
associated with these loci, including *ACAD10* (12q24.12), *ALDH2* (12q24.12), *HECTD4* (12q24.12),
*MAPKAPK5* (12q24.12), *NAA25* (12q24.12), *SH2B3* (12q24.12), *TMEM116* (12q24.12), *SERPINF1*
(17p13.3), *TMED6* (16q22.1), and *XPO7* (8p21.3). To explore the biological mechanisms of these genes,
pathway enrichment analysis revealed their involvement in crucial processes such as DNA biosynthesis and
telomere maintenance. Additionally, we found that *SH2B3*, one of the genes highlighted in our gene-level
analysis, was further supported by a proteome-wide MR study. Genetically predicted *SH2B3* protein levels
were linked to both LTL-CAD and LTL-VTE, reinforcing its potential role in the shared genetic
architecture of LTL and CVDs.

Inconsistence results: Although genome-wide r_g offered valuable insight into the genetic overlap between
phenotypes, it could not distinguish genetic overlap resulting from a mixture of concordant and discordant
effects from the absence of genetic overlap, potentially yielding an estimated r_g near zero in both scenarios.
Therefore, using multiple methods with different model assumptions to identify and understand this
"missing dimension" of genetic overlap was essential for comprehensively characterizing the shared genetic

foundations across phenotypes. Specifically, the pairwise LDSC analysis identified negative genome-wide
genetic correlations between LTL and several CVDs, including CAD, HF, PAD, Stroke, and VTE, with r_g
ranging from -0.250 to -0.072. Importantly, there was no significant genetic correlation between LTL and
AF, but a notable genetic overlap identified by LAVA and MiXeR was observed, which led us to include
AF in the final union set for further investigation.

2) Furthermore, in the Pleiotropic genomic loci identified for LTL and CVDs and Pleiotropic genes
associated with LTL and multiple CVDs sections the authors attempt to connect the identified SNPs to
potential functional genes for pathway-based evaluations. While it is very difficult to make this connection
without extensive and detailed functional data relevant for both TL attrition and CVDs and the authors
attempted to evaluate multiple scenarios (closes genes through MAGMA analysis, inclusion of QTL and
other tissue level mics data), it makes the results difficult to crystallize. Would suggest including only the
results of genes from either the overlapping list (MAGMA and E-MAGMA) or only MAGMA based results
(it has been suggested recently that usually the closest gene is likely functional gene from GWAS studies
and using the closest gene to the SNP perform relatively well in subsequent pathway evaluations).

Re: Thank you for your insightful feedback. We completely agree with your suggestion to focus on either
the overlapping results (MAGMA and e-MAGMA) or MAGMA-based results alone. Our study indeed
employed the two approaches suggested by the reviewer. To assess the functional relevance of these genes,
we performed a gene-set enrichment analysis on the results from the MAGMA analysis. This identified
several pathways related to chromatin organization, the negative regulation of nucleobase-containing
compound metabolic processes, and the negative regulation of miRNA maturation. Notably, the genes
shared between LTL and AF were significantly associated with the 'negative regulation of nucleobase-
containing compound metabolic processes.' Furthermore, using ToppFun to analyze the overlapping genes
from both MAGMA and e-MAGMA, we found that key biological processes, particularly those related to
telomere maintenance, were enriched. These included pathways like 'telomere maintenance via telomerase'
and 'telomere capping,' suggesting a shared role in the biology of LTL and CVDs.

3) As the authors highlight, it is well known that reduced LTL is an increased risk for various CVDs. Given
this negative correlation would it be necessary to perform subsequent pathway-based enrichment analyses
separately for variants that follow this known discordant relationship (overlapping variants that decrease
LTL and also increase CVD risks) from variants that show a positive correlation (overlapping variants that
either both decrease or increase LTL and CVD risks).

Re: We appreciate the reviewer’s insightful comment. In response, we classified the shared variants into
 two groups: those with positive correlations and those with negative correlations, to explore whether these
 distinct categories of variant-driven genes were enriched in different biological pathways. Our analysis
 revealed that genes driven by positively correlated variants were primarily enriched in pathways related to
 cellular structure regulation, intracellular transport, and biological interactions. Notably, these pathways
 were not identified in our previous analysis of all overlapping variants. Additionally, we highlight that
 MAPKBP1 is specifically enriched in pathways involved in the negative regulation of the response to biotic
 stimuli. At the same time, XPO7 is associated with pathways related to intracellular transport, nuclear
 transport, and nucleocytoplasmic transport. In contrast, genes driven by negatively correlated variants were
 predominantly enriched in telomere-related pathways, including telomere capping, telomere maintenance
 via lengthening, telomerase-mediated maintenance, RNA-templated DNA biosynthesis, and telomere
 organization. Interestingly, genes such as MAPKAPK5 were consistently associated with all these
 pathways. These telomere-related pathways were also observed in our previous analysis of all overlapping
 variants, suggesting that genes driven by negatively correlated variants play a dominant role in pathway
 enrichment.

 4) Following on this, would there be subsets of variants (and subsequent dysfunctional pathways) that
 follow this known negative correlation for a particular trait pair (i.e LTL – CAD pair) but not is another
 pair (i.e LTL – STROKE pair).

 Re: We appreciate the reviewer’s insightful comment. In response, we examined the distribution of
 pleiotropic genes that drive telomere-related pathways across trait pairs. Our analysis found that these
 pleiotropic genes were consistently associated with at least two trait pairs, with the exception of LTL-AF.
 This suggests that telomere-related pathways play a more generalized role in linking LTL with CVDs,
 excluding AF, rather than being specific to any single trait pair.

 **Table: Pleiotropic genes driving telomere-related pathways**

Trait pairs	No loci	locus	SYMBOL	Pathway
LTL-CAD	10q24.33	33	STN1	[1], [2], [3], [4], [5], [6]
LTL-Stroke	10q24.33	13	STN1	
LTL-CAD	12q13.13	39	HNRNPA1	[2], [3], [4], [5], [6]
LTL-Stroke	12q13.13	15	HNRNPA1	
LTL-CAD	12q24.12	40	MAPKAPK5	[1], [2], [3], [4], [5], [6]

LTL-PAD	12q24.12	18	MAPKAPK5	
LTL-Stroke	12q24.12	16	MAPKAPK5	
LTL-VTE	12q24.12	35	MAPKAPK5	
LTL-HF	17p13.3	16	RPA1	[2], [3], [4], [5], [6]
LTL-PAD	17p13.3	29	RPA1	
LTL-CAD	1p13.2	2	DCLRE1B	
LTL-HF	1p13.2	1	DCLRE1B	[1], [2], [3], [4]
LTL-PAD	1p13.2	2	DCLRE1B	
LTL-Stroke	1p13.2	1	DCLRE1B	

Note: [1]: telomere capping, [2]: telomere maintenance via telomere lengthening, [3]: telomere maintenance,
[4]: telomere organization, [5]: telomere maintenance via telomerase , [6]: RNA-templated DNA
biosynthetic process.

5) The authors indicate that additional East-Asian GWAS summary data was utilized for replications, but
it is not clear which datasets were used and which results were replicated in the results section.

Re: Thank you for your insightful comment. Due to word constraints and the primary use of East Asian
data for replication purposes, we have included all related results in the supplementary materials.
Specifically, we used East Asian GWAS data for LTL, AF, CAD, HF, PAD, and Stroke to replicate findings
on genetic overlaps, pleiotropic loci, risk genes, and biological pathways. However, due to the absence of
tissue-specific eQTL reference files for East Asian populations, we did not perform e-MAGMA, TWAS,
or pathway enrichment analyses for tissue-specific genes.

Despite numerous genetic studies on LTL or CVDs in non-European populations, these typically feature
smaller sample sizes than those focusing on European populations. However, Dorajoo et al. conducted a
GWAS for LTL in 23,096 individuals from a Singaporean Southern Han Chinese population, highlighting
key genetic variants for LTL specific to East Asians (Dorajoo R *et al.*, 2019). Ishigaki *et al.* performed a
comprehensive GWAS of 42 common diseases through the BioBank Japan Project (BBJ), which included
approximately 200,000 Japanese individuals and covered major CVDs such as CAD, HF, PAD, and Stroke
(Ishigaki K *et al.*, 2020). Additionally, Low et al. conducted the largest GWAS for AF in the Japanese
population, involving 8,180 cases and 28,612 controls (Low SK *et al.*, 2017), while Zhang et al. performed
the first GWAS for VTE in a Han Chinese cohort (1,268 cases and 17,663 controls) (Zhang Z *et al.*, 2023).
Due to the lack of published GWAS summary statistics for VTE at the time of our analysis, only data for
AF, CAD, HF, PAD, and Stroke were included in our study.

Our analysis revealed no significant genome-wide correlations between LTL and CVDs in East Asian
populations, except PAD. This lack of association is likely due to the small sample sizes and limited
statistical power of the original GWAS studies, which constrain the ability to detect meaningful genetic
correlations. The identification of PAD as an exception suggests a potential unique genetic relationship
between LTL and PAD in East Asians, warranting further investigation. Overall, these findings underscore
the challenges of studying CVD genetics in underrepresented populations and highlight the importance of
larger, well-powered studies.

We identified a total of 11 independent genomic risk loci as pleiotropic in East Asian populations. Of these,
9 loci (e.g., 4q25, 10q24.33, and 1q42.12) were also observed in European populations, while 2 loci (3q26.2
and 12q24.11) appeared to be specific to East Asians. This highlights the significance of incorporating
ancestry-specific loci to uncover associations that may not be present across populations. Furthermore,
three loci originally identified in European GWAS studies of AF, CAD, and Stroke (4q25, 10q24.33, and
2p24.1) were confirmed to be relevant in East Asian populations. These findings emphasize the importance
of cross-population comparisons in understanding shared and unique genetic mechanisms underlying CVDs.
Among the 17 pleiotropic genes identified in East Asian populations, the number was markedly lower than
in European populations ($n = 478$), likely due to the smaller sample sizes and reduced statistical power in
East Asian studies. Most pleiotropic genes were ancestry-specific, reflecting distinct genetic architectures
between populations. For example, *SH2B3* was identified in European datasets but not in East Asian
samples, while three genes (*COL17A1*, *SH3PXD2A*, and *STN1*) were shared across ancestries. For example,
*COL17A1*, a component of hemidesmosomes, has been implicated in immune inflammation and may
contribute to post-myocardial infarction recovery by reducing pro-inflammatory cytokine secretion.
*SH3PXD2A* interacts with NADPH oxidases to facilitate reactive oxygen species (ROS) formation, linking
it to oxidative stress and aging-related disorders. *STN1* plays a critical role in telomere replication and
maintenance, with common polymorphisms associated with longer LTL and protection against CVD-
related mortality. Gene-set analysis using MAGMA revealed distinct pathway enrichment patterns between
East Asian and European ancestry samples. These differences suggest that the genetic architecture
underlying LTL and CVDs may vary significantly across populations, reflecting unique biological
pathways and environmental influences. These findings highlight the importance of population-specific
analyses in capturing the full spectrum of genetic contributions to complex traits and diseases.

**Reference:**

Dorajoo R, Chang X, Gurung RL, et al. Loci for human leukocyte telomere length in the Singaporean
Chinese population and trans-ethnic genetic studies. *Nat Commun.* 2019;10(1):2491. Published 2019 Jun
6. doi:10.1038/s41467-019-10443-2

Ishigaki K, Akiyama M, Kanai M, et al. Large-scale genome-wide association study in a Japanese
population identifies novel susceptibility loci across different diseases. *Nat Genet.* 2020;52(7):669-679.
doi:10.1038/s41588-020-0640-3

Low SK, Takahashi A, Eban Y, et al. Identification of six new genetic loci associated with atrial fibrillation
in the Japanese population. *Nat Genet.* 2017;49(6):953-958. doi:10.1038/ng.3842

Zhang Z, Li H, Weng H, et al. Genome-wide association analyses identified novel susceptibility loci for
pulmonary embolism among Han Chinese population. *BMC Med.* 2023;21(1):153. Published 2023 Apr 19.
doi:10.1186/s12916-023-02844-4

6) The authors correctly highlighted issues with previous studies, especially causal inference studies, that
may have a level of bias due to sample overlap between traits evaluated. Would the causal inference studies
performed in the present study in The Causal inference between LTL and six major CVDs and Shared
causal proteins between LTL and six major CVDs sections also be similarly affected by some level of bias
due to the extensive overlap of the UKBiobank data for LTL, CVDs, proteomics and other traits. This would
be especially important to suggest that lack of causality between these traits.

Re: We appreciate your valuable insights. In standard MR studies, genetic variants serve as instrumental
variables to infer causal relationships. However, if the effects of these instrumental variables overlap
between two sample populations (e.g., when the same individuals are used for both exposure and outcome
variables), it can lead to biased results. This is because the shared genetic background, which is unobserved
in the overlapping samples, may influence the relationship between the exposure and outcome variables,
potentially creating a false causal link.

In our analysis of the causal relationship between LTL and six major CVDs, we considered the potential
effects of sample overlap. To address this, we previously employed LCV and LHC-MR methods, rather
than relying on traditional MR approaches. In response to Reviewer 3's comments, we further investigated
the robustness of LHC-MR by comparing causal inferences using GWAS data with and without overlapping
samples. Our findings revealed that the results obtained from LHC-MR were not robust and may not be
suitable for causal inference in the presence of sample overlap. To mitigate potential biases from sample
overlap, we further conducted an inverse variance weighting (IVW) Mendelian randomization analysis
using the MRlap approach (Mounier *et al.*, 2023). MRlap corrects for biases arising from sample overlap,

weak instruments, and winner's curses, offering more reliable causal estimates. The MRlap analysis
confirmed the previously identified negative causal relationship between LTL and CAD, aligning with the
results of traditional MR analysis. Moreover, there was a high degree of agreement between MRlap and
LCV results, with both methods inferring causality in a manner consistent with existing epidemiological
evidence. Based on these findings, we considered the MRlap method can act as the primary outcome in the
sensitivity analysis of LCV results. In contrast, LHC-MR appeared to overestimate reverse causality,
leading to inconsistent results. (From Page 10, line 254 to line 270)

In addition, we used SMR analysis to analyze protein levels, which is consistent with the basic idea of
traditional MR analysis. SMR analysis can test whether exposure (such as protein expression) and disease
outcomes based on aggregated data share the same genetic variants (co-localization effect). The challenge
of sample overlap is difficult to ignore in this analysis, which has become one of the limitations of our study.
Therefore, our study utilized the largest and most comprehensive GWAS dataset available for LTL and six
CVDs in order to minimize the impact of sample overlap. Despite this, we acknowledge that sample overlap
remains a potential limitation, and we have highlighted this issue in the limitation of our study. (From Page
24, line 668 to line 671)

**Reference:**

Mounier N, Kutalik Z. Bias correction for inverse variance weighting Mendelian randomization. *Genet*
*Epidemiol.* 2023;47(4):314-331. doi:10.1002/gepi.22522

7) Previous work by the ENAGAGE consortium have utilized similar datasets to suggest that insulin and
potential glucose metabolism may be a strong mediator between LTL and CVD (PMID: 28515044 DOI:
10.1161/CIRCRESAHA.116.310517). It was quite striking that this was not picked up strongly in the
present study. Could the authors evaluate if similar genes were identified in the present study and discuss
their findings in relation to these previous data?

Re: We are grateful for your insightful suggestions. We performed HyPrColoc analysis to determine
whether pleiotropic loci co-localize across traits associated with LTL, CVDs, and glycemic-related traits,
including fasting insulin (FI), fasting glucose (FG), two-hour glucose (2hGlu), and glycated hemoglobin
(HbA1c). Our results revealed strong colocalization between LTL, FI, HbA1c, and four CVDs (CAD, VTE,
HF, and Stroke), all sharing a common causal SNP (rs10774625), located within an intron of the *ATXN2*
gene on 12q24.12. Previous research by Zhan et al. demonstrated that shorter telomeres were associated
with lower fasting insulin levels and increased CAD-related indices, suggesting that insulin plays a key role

in the progression from telomere attrition to CAD (Zhan Y *et al.*, 2017). Our findings corroborate this
association and further identify *ATXN2* as a shared genetic locus linking insulin metabolism, telomere
length, and CVD risk. Functionally, *ATXN2* is involved in RNA processing and nutrient receptor
internalization, and its loss has been shown to induce obesity and insulin resistance, further contributing to
lipid and glucose dysregulation (Auburger *et al.*, 2014; Ikram *et al.*, 2010). Collectively, our supplementary
analysis provides compelling evidence of colocalization between insulin metabolism, glucose homeostasis,
LTL, and CVDs, suggesting shared genetic mechanisms underlying these traits.

**Reference:**

Zhan Y, Karlsson IK, Karlsson R, et al. Exploring the Causal Pathway From Telomere Length to Coronary
Heart Disease: A Network Mendelian Randomization Study. *Circ Res.* 2017;121(3):214-219.
doi:10.1161/CIRCRESAHA.116.310517

Auburger G, Gispert S, Lahut S, Omür O, Damrath E, Heck M, Başak N. 12q24 locus association with type
1 diabetes: SH2B3 or ATXN2? *World J Diabetes.* 2014 Jun 15;5(3):316-27. doi: 10.4239/wjd.v5.i3.316

Ikram MK, Sim X, Jensen RA, Cotch MF, Hewitt AW, Ikram MA, Wang JJ, Klein R, Klein BE, Breteler
MM, Cheung N, Liew G, Mitchell P, Uitterlinden AG, Rivadeneira F, Hofman A, de Jong PT, van Duijn
CM, Kao L, Cheng CY, Smith AV, Glazer NL, Lumley T, McKnight B, Psaty BM, Jonasson F, Eiriksdottir
G, Aspelund T; Global BPgen Consortium; Harris TB, Launer LJ, Taylor KD, Li X, Iyengar SK, Xi Q,
Sivakumaran TA, Mackey DA, Macgregor S, Martin NG, Young TL, Bis JC, Wiggins KL, Heckbert SR,
Hammond CJ, Andrew T, Fahy S, Attia J, Holliday EG, Scott RJ, Islam FM, Rotter JI, McAuley AK,
Boerwinkle E, Tai ES, Gudnason V, Siscovick DS, Vingerling JR, Wong TY. Four novel Loci (19q13,
6q24, 12q24, and 5q14) influence the microcirculation in vivo. *PLoS Genet.* 2010 Oct 28;6(10):e1001184.
doi: 10.1371/journal.pgen.1001184

Reviewer #3 (Remarks to the Author):

In this paper Qiao et al. throw lots of methods at a complex question investigating a shared genetic biology
of telomere length and the risk of six major cardiovascular diseases (CVDs). The authors focus on an
investigation of horizontal pleiotropic effects that are likely to reveal shared biology but does not manage
to create a clear narrative when exploring the established TL CVD relationship.

1. Due to the sheer number of methods used a figure detailing the strategy and logical process through these
analyses would help to elucidate the meaning for some of these tests and their findings, which is currently
a bit lost within the text. It is my impression that there are two strategies. The first being correlation leading
to causal inference. The second being colocalization leading to gene/protein set analyses. I believe both
have serious flaws.

Re: We apologize for the confusion. For the genome-wide level, we estimated the genetic correlation (r_g)
between LTL and CVDs using LDSC. A global r_g represents the average of the shared association signals
across the genome. However, it does not differentiate between (i) genetic overlap with a mixture of
concordant and discordant effects and (ii) an absence of genetic overlap, as both return an estimate close to
0. To address this “missing dimension” of genetic overlap, the MiXeR was employed to demonstrate
extensive genetic overlap and mixed effect directions between LTL and CVDs. While global r_g provides a
broad overview, local genetic correlations offer a more effective approach for identifying and capturing
associations with mixed effect directions. We applied LAVA to perform local genetic correlations ($loc-r_{gs}$)
between LTL and six major CVDs across 2,495 semi-independent genetic regions of approximately equal
size (~1 Mb). By charting the landscape of genetic overlap beyond r_g , we provide insights into shared
genetic architectures between LTL and six major CVDs.

These shared genetic bases reflect genetic variants that influence multiple complex phenotypic traits, which
can occur through vertical or horizontal pleiotropy. Horizontal pleiotropy arises when two phenotypes are
affected by the same genetic variant, shedding light on shared biological pathways linking complex traits.
In contrast, vertical pleiotropy occurs when a genetic variant influences one phenotype, which in turn causes
a secondary phenotype—this is the type of relationship primarily targeted by MR analysis. To distinguish
these mechanisms, we conducted extensive analyses of both horizontal and vertical pleiotropy, testing
potential causal genetic effects using LCV and MRlap methods and focusing on elucidating contributions
from vertical pleiotropy.

To capture horizontal pleiotropy, we applied various statistical tools to characterize shared loci and examine
their functional implications for genes, tissues, biological processes, and protein targets. At the SNP level,
PLACO was conducted to identify pleiotropic variants shared between LTL and CVDs, considering only
those reaching genome-wide significance ($P < 5 \times 10^{-8}$). FUMA was applied to characterize potential
pleiotropic loci, based on which a Bayesian colocalization analysis was performed to further identify shared
causal variants in each pleiotropic locus. At the gene level, MAGMA was used to identify candidate
pleiotropic genes. LDSC-SEG identified the tissue types specificity of these genes, which helped refine the
understanding of gene function and expression patterns. Furthermore, e-MAGMA and TWAS were applied
to identify pleiotropic genes specific to particular tissue types. At the pathway level, MAGMA gene-set
analysis was also conducted to explore the biological functions of these pleiotropic genes, testing 7,744
gene sets, including Gene Ontology biological processes from the MSigDB. Additionally, pathway
enrichment analysis of mapped genes was carried out using ToppFun. Finally, to elucidate the biological
underpinnings of the shared genetic predisposition to LTL and six major CVDs, we performed proteome-
wide MR and colocalization analyses to identify potential druggable targets.

In summary, this study addressed five central questions regarding the shared genetic basis of LTL and six
major CVDs: (i) Can we identify shared genetic architectures across LTL and CVDs? (ii) Are these shared
genetic architectures driven by causal associations, specifically vertical pleiotropy? (iii) Can we detect
genomic loci shared by LTL and multiple CVDs, and do some loci exhibit opposite allelic effects between
LTL and CVDs? (iv) Can we identify functional features of pleiotropic loci that explain their broad impact
on LTL and cardiovascular pathology? (v) Can we identify causal plasma proteins as therapeutic targets
affecting LTL and major CVDs?

Finally, we have included a figure that outlines the strategy and logical process behind these analyses
(Supplementary Fig. 1), to provide the readers of Nature Communications with a clearer understanding of
the manuscript's approach and findings.

2. Performed LDSC regression to look at rg. Looking at results in the supplement the authors found no
significant evidence of genome-wide correlation with all but PAD, which was only nominally associated
($p=0.047$) in East Asians (s. table 1). The analysis presented in the rest of the paper is for Europeans only
where there was evidence of a genetic correlation with LTL in all but AF. S Table 1 does not show the
studies included in the presented analysis. This contradictory information puts the strategy in place in a
weak position as it would appear results weren't as expected in East Asians so the analysis side-stepped to
Europeans. These genetic correlations are also not novel.

Re: We appreciate the reviewer's comment and would like to clarify a few key points. Our study mainly
focuses on European populations, as detailed in the main analysis, due to the availability of large, well-
powered GWAS datasets that provide robust and reliable genetic correlation estimates. In contrast, the
analysis of East Asian populations was conducted as an exploratory effort to assess the generalizability of
our findings across different populations. However, the GWAS datasets for East Asians are limited by
smaller sample sizes and less comprehensive coverage, which significantly reduces the power to detect
genetic variants within this population. This limitation highlights the current disparity in available genetic
resources, underscoring the critical need for expanded GWAS efforts in non-European populations to
achieve more globally representative analyses. Due to space constraints, we have placed the results of the
East Asian analysis in the Supplementary file, with the corresponding LDSC results presented in Table 2
of the Supplementary Note Table, instead of Table 1.

3. The LTL data from UK Biobank are of predominantly European ancestry but, importantly, also include
non-Europeans. The same is true of data for many of the CVD GWAS included here and without the
information in ST1 it is unclear which sample sizes were used in this analysis to know if they are European
only (I assume not). This has an impact on many analyses, such as those utilising the 1000g EU reference
panel data.

Re: Thank you for your insightful comment. In our analysis, we carefully considered the potential impact
of ancestry on the results, and as such, we selected exclusively the European subset of the LTL and CVD
GWAS data. For further details on the study populations and sample sizes, please refer to Supplementary
Table 1.

4. Then look at MiXeR that aims to look at overlapping 'causal' variants that is not bound by a constant
shared direction of effect. This highlighted an overlap of signal at numerous locations thus supporting an
analysis of specific loci. LAVA was then applied and HyPrColoc. The threshold for PP of 0.7 is liberal and
non-conclusive but suggestive that there may be a pleiotropic loci present.

Re: We appreciate the reviewer's insightful comment. As noted, LAVA divides the genome into 2,495
semi-independent genetic regions, which, while useful for an initial exploration of pleiotropy, can result in
less precise colocalization analysis. The threshold for posterior probability (PP) of 0.7 is indeed liberal and
should be interpreted as suggestive, rather than definitive. The results obtained from LAVA are intended to
highlight regions that warrant further investigation rather than to provide conclusive evidence. In contrast,

PLACO offers finer granularity by subdividing the genome further, which enables a more accurate analysis
of specific loci.

5. *ATXN2* loci came up as linked to all CVDs using HyPrColoc. Though this is also linked to blood pressure
and lipids suggesting a pleiotropic relationship larger than that which is analysed here as LTL also links to
these cardiovascular risk factors.

Re: Thank you for your insightful feedback. However, in our original manuscript, we did not claim that
hyprcoloc analysis identified colocalization with all CVDs. Instead, our results specifically indicated
colocalization with CAD, HF, VTE, and Stroke. We further conducted HyPrColoc analysis to explore
potential colocalization between LTL, CVDs, and several cardiovascular risk factors, including blood
pressure traits (systolic blood pressure [SBP], diastolic blood pressure [DBP], pulse pressure [PP]) and
blood lipid traits (low-density lipoprotein cholesterol [LDL-C], high-density lipoprotein cholesterol [HDL-
C], triglycerides [TG], and total cholesterol [TC]). Our results showed significant colocalization for LTL,
DBP, SBP, three lipid traits (HDL-C, LDL-C, and TC), and three CVDs (CAD, VTE, and HF), with PP
higher than 0.7, indicating shared causal variants. Specifically, the SNP rs3184504, located in the *ATXN2*
gene on chromosome 12q24.12, emerged as a shared causal variant linking these traits. This suggests that
the *ATXN2* locus may have a broader pleiotropic role, influencing both cardiovascular risk factors (such as
blood pressure and lipids) and LTL, thereby linking LTL to multiple cardiovascular traits.

6. Causal analyses are highly limited and attempt to address the difference between a direct and indirect
causal association in a bi-directional approach that fails to convince in the findings of no association. This
is due to the nature of a genome-wide instrument (importantly, this is not detailed here) and goes against
current knowledge. Without detail it makes it very hard to believe. Comprehensive evidence linking TL to
CVDs utilising non-pleiotropic instruments were shown in the publication where the TL data come from.
Again, supplementary tables are presented for a different East Asian analysis that makes it impossible to
assess this analysis.

Authors heavily utilised genetic correlation methods, and to show “causal” part chose latent causal variable
(LCV) analysis and Latent Heritable Confounder Mendelian Randomization (LHC-MR). Both LCV and
LHC-MR use whole-genome summary statistics to explore bi-directional causal associations between
complex traits and yield effect estimates accounting for bi-directional genetic effects, direct heritability,
and confounder effects. Unfortunately, in LCV and LHC-MR settings it is difficult to assess which genetic
variants are potentially causal, which have direct, indirect, or null effects, which are biologically plausible.
Moreover, LHC-MR is a relatively new method and might not perform as well as advertised (PMID:

35576237). Using LCV and LHC-MR on the whole-genome may incorporate information that may not be
relevant in association of just two traits – potentially, they may be correlated via some other traits, Thus the
first strategy arm, correlation to causal inference, struggles to convince in the final outcome.

Re: Thank you for your valuable suggestions. MR was also used to assess causal relationships between
LTL and disease in publications from LTL data sources, but sample overlap was not considered. This may
be a significant reason why they overestimate the causal relationship between LTL and CVD. In addition,
they use observational studies to supplement the evidence, but they do not essentially prove a causal
association. We provide detailed results on LTL and CVDs causal estimates in European populations for
reference in Supplementary Table 6.

To assess the robustness of LHC-MR in determining causal relationships between GWAS data with
overlapping samples, we tested its performance by including CVD summary statistics that excluded UK
Biobank data and comparing these results with our previous analysis, which used overlapping samples. We
only obtained GWAS summary statistics for CAD (not including UK Biobank data). Our analysis revealed
that LHC-MR demonstrates no putative causal relationship between CAD and LTL (not including UK
Biobank data). Inconsistencies previously observed in reverse causality between LTL and CAD suggest
that LHC-MR's results may lack robustness in the presence of sample overlap. To address this issue and
mitigate potential biases, we performed an inverse variance weighting (IVW) Mendelian randomization
(MR) analysis using the MRlap approach (Mounier *et al.*, 2023). MRlap corrects for biases arising from
sample overlap, weak instruments, and winner's curses, thus providing more reliable causal estimates (From
Page 30, line 849 to line 861). The MRlap analysis confirmed the previously identified negative causal
associations between LTL and CAD, also consistent with the results of traditional MR analysis. Notably,
we observed a high degree of agreement between the MRlap and LCV results, with both methods inferring
causality that aligns with existing epidemiological evidence. Based on these findings, we considered the
MRlap conclusions to be the primary outcome in the sensitivity analysis of LCV results. In contrast, LHC-
MR may have overestimated reverse causality, leading to inconsistent results. (From Page 10, line 265 to
line 270)

7. The authors, on lines 268-271, state it as remarkable that 47.2% of the loci have concordant effects that
increase LTL and decrease CVD risk yet this is entirely within the understood relationship between the
traits.

Re: Thank you for your thoughtful comment. We agree with the reviewer that the observed effect, where
some loci increase LTL and decrease CVD risk, is consistent with the well-established negative relationship
between these traits. While this relationship is indeed known, our analysis provides a more detailed and
specific view by identifying particular loci where the directionality of effects is inconsistent, highlighting
the complexity of the genetic architecture underlying this relationship.

8. The evidence supporting signals as causal for both is not present, as shown in the colocalization results
where very few of these signals (only 22/248) pass a too liberal threshold of 0.7. As a combined score for
2 traits it appears evident that the signals are predominantly driven by 1 of the traits and likely not truly
overlapping signals.

Re: We thank the reviewer for the comment. As we explained in response to Question 10, PLACO is
specifically designed to identify shared genetic variations between traits without requiring strict
colocalization at the genome-wide level. While colocalization is a useful tool for pinpointing genetic
variants that are significant in both traits, it is not definitive; rather, it provides supplementary evidence.
Therefore, relying solely on stringent colocalization to explore the genetic mechanisms between trait pairs
is unreasonable. In conclusion, this approach is consistent with established practices in the field, where
colocalization serves as an additional layer of evidence, rather than a strict criterion for identifying causal
variants. For example, Gong et al. used the same strategy to investigate the shared genetic causes between
gastrointestinal disorders and mental illnesses, successfully identifying common genetic loci, genes, and
pathways (Gong *et al.*, 2023).

**Reference:**

Gong W, Guo P, Li Y, et al. Role of the Gut-Brain Axis in the Shared Genetic Etiology Between
Gastrointestinal Tract Diseases and Psychiatric Disorders: A Genome-Wide Pleiotropic Analysis. *JAMA*
*Psychiatry*. 2023;80(4):360-370. doi:10.1001/jamapsychiatry.2022.4974

9. This is where the second strategy arm, coloc to gene/protein sets, fails to convince as all downstream
analyses were then performed on all loci despite no evidence of a colocalized signal for the huge majority.
This makes the interpretation of any output wildly speculative that does not reflect the hypothesis of a
shared biology and combines genes from different traits together with no strong justification that these
genes are shared.

Re: Thank you for your constructive comments. We understand the concern that not all loci showed strong

evidence of colocalization in our COLOC analysis. However, we intentionally included all loci in the
downstream analyses to capture potential pleiotropic signals, even those with weaker evidence of
colocalization. Our approach aims to ensure that we do not overlook loci that might still contribute to shared
biological mechanisms, as complex genetic relationships often involve loci with varying levels of evidence.
This strategy is consistent with approaches used in prior studies, where loci with weaker evidence were
included to enhance the understanding of complex pleiotropic interactions. For example, Gong et al. used
the same strategy to investigate the shared genetic causes between gastrointestinal disorders and mental
illnesses, successfully identifying common genetic loci, genes, and pathways (Gong *et al.*, 2023).

**Reference:**

Gong W, Guo P, Li Y, et al. Role of the Gut-Brain Axis in the Shared Genetic Etiology Between
Gastrointestinal Tract Diseases and Psychiatric Disorders: A Genome-Wide Pleiotropic Analysis. *JAMA*
*Psychiatry*. 2023;80(4):360-370. doi:10.1001/jamapsychiatry.2022.4974

10. The FUMA analysis only considered results from the PLACO analysis novel if not found in the single
trait GWAS. This is counterintuitive given the hypothesis that the variant/loci is linked to both traits through
shared biology. Colocalization $PP > 0.7$ is only suggestive and is not 'strong evidence' (as stated in line 300),
reserved for more stringent criteria > 0.9 .

Re: We appreciate your valuable insights. PLACO analysis is designed to detect pleiotropic loci between
two traits and assess the genetic mechanisms shared by these traits. It does this by considering a composite
null hypothesis, which assumes that a genetic variant is either unrelated to both traits or associated with
only one of the traits (Ray *et al.*, 2021; Kircher *et al.*, 2014). The null hypothesis is tested by evaluating the
product of the Z statistics of the genetic variants for both traits. The null distribution of the test statistic is
derived from a mixed distribution, which accounts for the fraction of variants associated with either one or
both traits. We first calculate the square of the Z score for each genetic variant, then compute the correlation
matrix of Z scores to account for potential correlations between the traits under investigation. To test the
null hypothesis of no pleiotropy, we employ the intersection union test (IUT). The composite null
hypothesis, H_0 , is expressed as $H_0: H_{00} \cup H_{01} \cup H_{02}$, while the alternative hypothesis is H_1 . The
hypotheses are further defined as follows: $H_{00}: \beta_{\text{trait1}} = 0$ and $\beta_{\text{trait2}} = 0$ (no association with either
trait); $H_{01}: \beta_{\text{trait1}} = 0$ and $\beta_{\text{trait2}} \neq 0$ (association with trait 2 only); $H_{02}: \beta_{\text{trait1}} \neq 0$ and $\beta_{\text{trait2}} = 0$
(association with trait 1 only). Here, β represents the effect size of the phenotype, and h_c refers to the

complement of H (the opposite of the null hypothesis). The final p-value for the test is determined by taking
the maximum p-value from H0 and H1, and replacing it with the following asymptotic approximation:

$$954 \quad P_{Z_{trait1}Z_{trait2}} = F\left(\frac{Z_{trait1}Z_{trait2}}{\sqrt{Var(Z_{trait1})}}\right) + F\left(\frac{Z_{trait1}Z_{trait2}}{\sqrt{Var(Z_{trait2})}}\right) - F(Z_{trait1}Z_{trait2})$$

Z_{trait1} and Z_{trait2} represent the observed Z scores of the two conditions/diseases given the genetic variant. $F(y)$
is the two-sided tail probability of the normal product distribution of y. $Var(x)$ is the expected value of the
squared deviation of x from the mean.

According to this approach, it is not counterintuitive for PLACO analysis to uncover significant findings,
even when no significant associations are found in a single-trait GWAS. This is because the interaction
between multiple traits may help uncover shared genetic factors that might otherwise remain undetected.
In some instances, single-trait GWAS may fail to identify genetic effects that are weak or insignificant for
one trait, but these effects may become more pronounced when analyzed across multiple traits
simultaneously. Furthermore, in single-trait GWAS, a large number of statistical tests are performed, which
often results in the application of multiple comparison corrections. This process can lead to weak signals
being ignored or considered non-significant. In contrast, PLACO analysis integrates data from multiple
traits, allowing for the more powerful analysis of genetic variants that may have been overlooked in single-
trait studies. By accounting for the interactions between traits, PLACO analysis can identify potential
genetic associations that might have been missed.

In conclusion, the discovery of new signals in PLACO analysis—despite the absence of significant findings
in single-trait GWAS—should not be viewed as counterintuitive. Instead, it indicates that certain genetic
variants may play an essential role in the interaction of multiple traits, offering valuable insights into the
shared genetic architecture of complex diseases.

In addition, we have modified the inappropriate description of colocalization in the original text with ‘A
Posterior Probability for Hypothesis 4 (PP.H4) greater than 0.7 was considered evidence for colocalization,
suggesting the presence of shared causal variants at the locus.’ (From Page 32, line 922 to line 924)

**Reference:**
Ray D, Venkataraghavan S, Zhang W, Leslie EJ, Hetmanski JB, Weinberg SM, Murray JC, Marazita ML,
Ruczinski I, Taub MA, Beaty TH. Pleiotropy method reveals genetic overlap between orofacial clefts at

multiple novel loci from GWAS of multi-ethnic trios. PLoS Genet. 2021 Jul 9;17(7):e1009584. doi:
10.1371/journal.pgen.1009584

Kircher M, Witten DM, Jain P, O’Roak BJ, Cooper GM, Shendure J. A general framework for estimating
the relative pathogenicity of human genetic variants. Nat Genet. 2014 Mar;46(3):310-5. doi:
10.1038/ng.2892

11. In identifying *SH2B3* as a shared ‘causal’ protein and *ATXN2* as a shared pleiotropic loci I am reminded
that these are 2 top key drivers of hypertension (PMID:25882670) and wonder if they are simply 2 genes
involved in other complex pleiotropic associations that are unrelated to the LTL CVD relationship and these
are simply shadows of other mechanisms.

Re: Thank you for your careful review and feedback. To address your concern about whether *SH2B3* and
*ATXN2* merely reflect other pleiotropic mechanisms rather than a direct link between LTL and CVD, we
conducted the HyPrColoc analysis. This result revealed strong colocalization signals at 12q24.12 locus
across LTL, key cardiovascular risk factors (SBP, DBP, LDL-C, HDL-C, and TC), and three CVDs (CAD,
VTE, and HF), with posterior probabilities exceeding 0.7. Specifically, rs3184504, an SNP located between
the *SH2B3* and *ATXN2* genes within 12q24.12, is a shared causal variant linking these traits.

Multiple biological mechanisms support the relationship between the aforementioned gene loci and key
cardiovascular risk factors, telomere length, and cardiovascular disease. *SH2B3* is a key adaptor protein
that regulates cytokine signaling and immune responses. *SH2B3* has a well-established causal relationship
with blood pressure regulation. Previous studies by Saleh et al. demonstrated that *SH2B3* deficiency
exacerbates low-dose angiotensin II elevation via inflammatory pathways and T-cell activation, leading to
hypertension (Saleh *et al.*, 2006). It is well known that inflammatory regulation plays a central role in the
progression of non-ischemic heart failure and cardiac fibrosis (Dale *et al.*, 2016). *SH2B3* has been shown
to enhance the production of proinflammatory cytokines, thereby mediating inflammation and fibrosis
following myocardial infarction (Keefe *et al.*, 2019). However, the specific biological connection pathways
between telomeres and *SH2B3* remain unknown. Only studies have shown that the enzyme encoding the
elongation of telomeric DNA interacts with a missense variant in the *SH2B3* gene, which has been identified
as a key genetic determinant of human lifespan, disrupting the subcellular localization and function of
*SH2B3*, potentially altering hematopoietic homeostasis and inflammatory responses, both of which are
closely related to aging and lifespan regulation. Given the important role of telomere length in cellular aging
and disease susceptibility, this interaction highlights a potential mechanistic link between telomere
dynamics and *SH2B3* function in regulating human lifespan. On the other hand, *ATXN2* is a polyglutamine

protein that plays an important role in RNA metabolism and translation regulation. *ATXN2* has been
proposed as a new obesity candidate gene that regulates the adaptor protein GRB2, an amplifier of insulin
receptor signaling (Auburger *et al.*, 2014). *ATXN2* deficiency leads to a reduction in insulin receptors in the
liver and brain, thereby affecting blood lipids. Moreover, *ATXN2* may cause cardiovascular events by
affecting microcirculation, with evidence suggesting that this effect arises from changes in retinal vascular
caliber. However, the underlying biological mechanisms between *ATXN2* and telomeres remain to be
elucidated.

Thus, rather than being mere "shadows" of other mechanisms, *SH2B3* and *ATXN2* likely serve as genetic
nodes integrating inflammation, metabolism, telomere biology, and cardiovascular risk. Further
mechanistic studies are warranted to disentangle the direct impact of these genes on LTL dynamics beyond
their roles in blood pressure and/or lipids regulation.

**Reference:**

Saleh MA, McMaster WG, Wu J, Norlander AE, Funt SA, Thabet SR, Kirabo A, Xiao L, Chen W, Itani
HA, Michell D, Huan T, Zhang Y, Takaki S, Titze J, Levy D, Harrison DG, Madhur MS. Lymphocyte
adaptor protein LNK deficiency exacerbates hypertension and end-organ inflammation. *J Clin Invest*. 2015
Mar 2;125(3):1189-202. doi: 10.1172/JCI76327

Keefe JA, Hwang SJ, Huan T, Mendelson M, Yao C, Courchesne P, Saleh MA, Madhur MS, Levy D.
Evidence for a Causal Role of the SH2B3- β 2M Axis in Blood Pressure Regulation. *Hypertension*. 2019
Feb;73(2):497-503. doi: 10.1161/HYPERTENSIONAHA.118.12094

Dale BL, Madhur MS. Linking inflammation and hypertension via LNK/SH2B3. *Curr Opin Nephrol*
*Hypertens*. 2016 Mar;25(2):87-93. doi: 10.1097/MNH.0000000000000196

Auburger G, Gispert S, Lahut S, Omür O, Damrath E, Heck M, Başak N. 12q24 locus association with type
1 diabetes: SH2B3 or ATXN2? *World J Diabetes*. 2014 Jun 15;5(3):316-27. doi: 10.4239/wjd.v5.i3.316

12. Missing reference 16, where shorter genetic TL and its association with CVDs is discussed. Possibly
PMID: 32109421.

Re: Thank you for the reviewer's supplement. We have inserted this reference in the references of the
revised manuscript.

In conclusion, lots of analyses are performed that do not create a narrative of clear shared biology and
through their own investigation do not see a causal relationship. I am open to the concept of other factors

linking TL to CVD and TL being a surrogate but these analyses ventured through analysis toolkits that fall
down at key points (no observed colocalization and no detail on key MR analyses) and is thus certainly not
‘proved’ as stated in the abstract and conclusion.

Re: Thank you for your careful review and feedback. We hope that through improvements and further
clarifications, we can offer you new insights into our paper.

Reviewer #4 (Remarks to the Author):

I co-reviewed this manuscript with one of the reviewers who provided the listed reports. This is part of the
Nature Communications initiative to facilitate training in peer review and to provide appropriate recognition
for Early Career Researchers who co-review manuscripts.

Re: Thank you for sharing this information. We greatly appreciate the collaborative effort and the valuable
feedback provided by both reviewers.

Re: NCOMMS-24-58910

Title: Contribution of leukocyte telomere length to major cardiovascular diseases onset: phenotypic and genetic insights from a large-scale genome-wide cross-trait analysis

We thank the Reviewers and the Editor for the helpful feedback and insightful suggestions. The response from the reviewers is encouraging and constructive. Below, we provide point-by-point responses that address all the comments with additional analyses. By incorporating these new data and findings, our manuscript has been improved substantially.

Reviewer #1:

Reviewer #1 (Remarks to the Author):

1. The introduction is still out of focus and contains multiple unnecessary, nearly out of context content such as “Hayflick limit”, “fibrous cap”, “matrix degradation” that are never used again anywhere in the manuscript and do not help the flow of the argument. The authors should consider rewriting a more to-the-point Introduction to ease the flow of the section.

Re: We sincerely thank the reviewer for this important suggestion. In response, we have carefully revised the introduction section, eliminating a few out-of-context terms such as 'Hefflick limit' and 'matrix degradation'. In response to the second question, we have provided a clear and concise description of the fibrous cap to enhance the coherence of the text. The first paragraph of the introduction is now more focused and concise, highlighting the key biological mechanisms between telomere attrition and cardiovascular disease.

Here is the revised first paragraph of the Introduction: “Telomeres are repetitive DNA-protein complexes that cap the ends of linear chromosomes, safeguarding genomic stability by preventing degradation and fusion. Due to the end-replication problem, telomeres progressively shorten with each cell division, ultimately triggering cellular senescence or

apoptosis once a critical length is reached. Telomere attrition is a hallmark of biological aging and has been linked to several age-related diseases, including cardiovascular diseases (CVDs). In the vascular system, accelerated telomere shortening promotes the senescence of smooth muscle cells and macrophages, contributing to the formation of atherosclerotic plaques. These plaques, composed of a necrotic core surrounded by a fibrous cap made up of extracellular matrix, become unstable when senescent cells drive chronic inflammation and degradation of extracellular matrix, leading to thrombosis upon plaque rupture and elevating the risk of serious cardiovascular events. Endothelial dysfunction—an early feature of atherosclerosis—can further accelerate telomere erosion, establishing a vicious cycle of cellular aging and vascular damage. Despite growing interest, the association between leukocyte telomere length (LTL) and CVD risk remains unclear. While some studies report that shorter LTL is linked to a higher risk of coronary artery disease (CAD) and peripheral artery disease (PAD), others find no association, particularly in stroke or PAD outcomes. These discrepancies highlight the need for further research to clarify the underlying relationship between telomere biology and CVD pathogenesis.” (From Page 5, line 97 to line 114)

2. An example problem within the Introduction is the following sentence “These changes contribute to the thinning of the fibrous cap, thereby heightening the risk of CVDs.” The authors should provide context into what fibrous cap is and why its thinning increases risk of CVD.

Re: We appreciate the reviewer's helpful comment regarding the need to clarify the role of the fibrous cap. Accordingly, we have added a concise explanation in the Introduction, defining the fibrous cap as a critical structural layer covering atherosclerotic plaques. Its thinning or rupture compromises plaque stability, which significantly increases the risk of acute cardiovascular events such as myocardial infarction and stroke. We have integrated this explanation into the first paragraph of the Introduction. (From Page 5, line 102 to line 107)

3. The Introduction superficially presents various arguments and jumps back and forth between pure biology and data science studies without an obvious logic that links these different arguments. Thus create a disjointed and unpleasant reading experience.

Re: We thank the reviewer for highlighting the structural concerns. To improve flow and readability, we have reorganized the the first paragraph of Introduction to first describe the biological mechanisms by which telomere attrition contributes to cardiovascular pathology. Subsequently, we present supporting evidence from epidemiological and experimental studies. (From Page 5, line 97 to line 114)

4. In another place within the Introduction section, the authors mention “although the relationship with other CVDs remains more ambiguous”. What other CVDs are considered and what is the reason for this ambiguity? And what the authors exactly mean by ambiguous? Is it because these “Other CVDs” have not been studied, or the results were not exactly conclusive, and if so why?

Re: We thank the reviewer for requesting clarification regarding the term “ambiguity” in relation to other CVDs. While the inverse association between LTL and coronary artery disease is well-established, findings for other CVDs—such as ischemic stroke and PAD—are less consistent. For example, large studies including the UK Biobank and meta-analyses have confirmed the link between shortened LTL and increased cardiovascular mortality and coronary disease. However, some studies have reported no significant association between LTL and ischemic stroke or PAD. These discrepancies likely arise from differences in study design, population heterogeneity, and endpoints assessed.

Specifically, a prospective study of 768 patients followed up for 6 years showed that telomere shortening was associated with an increased incidence of cardiovascular events, even after adjusting for CVD risk factors (Baragetti *et al.*, 2015). A meta-analysis involving 43,725 people showed that LTL was inversely associated with coronary heart disease, but not significantly associated with cerebrovascular disease (Haycock *et al.*, 2014). Similarly, a

cross-sectional study reported an association between LTL and PAD, in which the risk of PAD increased by 44% for every standard deviation decrease in LTL, suggesting that shorter telomeres may play a role in the development of peripheral vascular disease (Raschenberger *et al.*, 2013). In contrast, some studies reported no significant association between LTL and certain cardiovascular outcomes. In a prospective cohort study of 14,916 healthy American men, no association was observed between relative LTL and the risk of ischemic stroke (Zee *et al.*, 2010). Similarly, in a longitudinal study, LTL shortening was not associated with the incidence of PAD (Willeit *et al.*, 2010).

Reference:

Baragetti A, Palmen J, Garlaschelli K, Grigore L, Pellegatta F, Tragni E, Catapano AL, Humphries SE, Norata GD, Talmud PJ. Telomere shortening over 6 years is associated with increased subclinical carotid vascular damage and worse cardiovascular prognosis in the general population. *J Intern Med.* 2015 Apr;277(4):478-87. doi: 10.1111/joim.12282

Haycock PC, Heydon EE, Kaptoge S, Butterworth AS, Thompson A, Willeit P. Leucocyte telomere length and risk of cardiovascular disease: systematic review and meta-analysis. *BMJ.* 2014 Jul 8;349:g4227. doi: 10.1136/bmj.g4227

Raschenberger J, Kollerits B, Hammerer-Lercher A, Rantner B, Stadler M, Haun M, Klein-Weigel P, Fraedrich G, Kronenberg F. The association of relative telomere length with symptomatic peripheral arterial disease: results from the CAVASIC study. *Atherosclerosis.* 2013 Aug;229(2):469-74. doi: 10.1016/j.atherosclerosis.2013.05.027

Zee RY, Castonguay AJ, Barton NS, Ridker PM. Relative leukocyte telomere length and risk of incident ischemic stroke in men: a prospective, nested case-control approach. *Rejuvenation Res.* 2010 Aug;13(4):411-4. doi: 10.1089/rej.2009.0975

Willeit P, Willeit J, Brandstätter A, Ehrlenbach S, Mayr A, Gasperi A, Weger S, Oberhollenzer F, Reindl M, Kronenberg F, Kiechl S. Cellular aging reflected by leukocyte telomere length predicts advanced atherosclerosis and cardiovascular disease risk. *Arterioscler Thromb Vasc Biol.* 2010 Aug;30(8):1649-56. doi: 10.1161/ATVBAHA.110.205492

5. Within the Introduction, lines 99 to 104 provides several claims without mentioning a scientific source.

Re: We sincerely thank the reviewer for highlighting this issue. We fully agree that all claims should be supported by appropriate scientific references. In response, we carefully revised lines 99–104 of the original introduction, adding two relevant citations to ensure that each statement is properly substantiated by literature.

6. The authors stated: “This body of evidence highlighted the potential impact of LTL on the development and progression of various CVDs “. The evidence is presented vaguely and is a brief mention of previous meta-analyses linking ITL and CVD. The authors do not present how these meta-analyses have an impact of ITL on both development and progression of ITL. The authors should provide clear, relevant, explicit evidence for their claims within the Introduction section.

Re: Thank you for the reviewer's comment regarding the vagueness of the evidence supporting the potential effect of LTL on the development and progression of CVDs. To address this issue, we have revised the introduction to provide clearer evidence from the studies mentioned. (From Page 5, line 109 to line 114)

7. Towards the end of the first paragraph within the Introduction section, the authors introduce a new angle of “inflammation and oxidative stress” that is presented superficially and vague making it a difficult reading experience and hard to pinpoint what the authors are suggesting as their central research question.

Re: Thank you for the reviewer’s insightful comment. To address this, we revised the paragraph by removing these vague elements and refocusing the discussion on the well-established link between cardiovascular endothelial dysfunction and telomere attrition, thereby improving coherence and clarity.

The revised paragraph now reads: “Endothelial dysfunction—an early feature of atherosclerosis—can further accelerate telomere erosion, establishing a vicious cycle of cellular aging and vascular damage.” (From Page 5, line 107 to line 109)

8. In the second paragraph of the Introduction, the authors start by repeating the same arguments as the first paragraph, this time instead of meta-analyses linking LTL and CVD, they vaguely and broadly mention epidemiological studies linking LTL and CVD. Without presenting these epidemiological studies, the authors start discussing genetics as the explanation for this link and briefly mention various genetic dimensions such as GWAs, vertical and horizontal pleiotropy. Yet at this point, it is unclear what a central research question is. All these issues and unclarity remain unaddressed from the previous version and the authors should consider a serious revision of this section to improve the readability, flow, and clarity.

Re: We thank the reviewer for their valuable and constructive feedback. In response, we have made substantial revisions to improve the structure, flow, and clarity of the Introduction. First, we have added specific references to relevant epidemiological studies in the first paragraph to provide a stronger empirical foundation for investigating shared genetic mechanisms. Second, we have refined the second paragraph to present a clearer transition from observational associations to a genetic perspective. In particular, we now explicitly define and distinguish the concepts of vertical and horizontal pleiotropy, and articulate our core research question more clearly.

Our central research question is whether the observed associations between LTL and various CVDs are underpinned by a shared genetic basis, and to what extent vertical and horizontal pleiotropy contribute to this overlap.

The revised second paragraph now reads as follows:

A plausible explanation for the observed association between LTL and CVDs is a shared genetic architecture. LTL exhibits considerable inter-individual variability across the human

lifespan but is highly heritable, with estimates ranging from 44% to 86%. The largest genome-wide association study (GWAS) of LTL, based on UK Biobank data, has identified 138 associated loci. Similarly, GWAS of various CVD phenotypes have revealed numerous risk loci, underscoring the contribution of common genetic variants to disease susceptibility. These findings offer an opportunity to investigate the genetic overlap between LTL and CVDs. Shared genetic architecture may be mediated by vertical or horizontal pleiotropy. Vertical pleiotropy refers to genetic variants influencing one trait that causally affects another—a framework often examined using Mendelian randomization (MR). Previous MR studies have suggested potential causal associations between shorter LTL and CAD or stroke, whereas findings for atrial fibrillation (AF) and heart failure (HF) remain inconsistent. In contrast, horizontal pleiotropy describes genetic variants that independently affect multiple traits, pointing to shared genes and biological pathways. Recent work by Gong et al. has demonstrated that horizontal pleiotropy plays a critical role in explaining the genetic architecture of complex traits. However, studies of horizontal pleiotropy between LTL and CVDs are limited, with most efforts focused on psychiatric disorders. Given the extensive epidemiological evidence and emerging insights from genetic studies, this study aims to systematically evaluate the shared genetic architecture and underlying mechanisms between LTL and multiple CVD phenotypes through a genome-wide pleiotropic analysis.

9. In line 134, please explain the study by Gong et al. and its relevance to the research question. Also ensure to clarify your central research question. What are you seeking to study between LTL and CVD and why? This is unclear throughout the Introduction section.

Re: We thank the reviewer for this helpful comment. The cited study by Gong *et al.* applied advanced genomic statistical methods to investigate the shared genetic architecture between gastrointestinal diseases and psychiatric disorders, emphasizing the role of the gut-brain axis (Gong *et al.*, 2023). Their work underscored the importance of distinguishing between vertical and horizontal pleiotropy when interpreting genetic overlap between complex traits. Inspired by this framework, our study aimed to explore whether LTL and CVDs share a common genetic architecture, and to disentangle the contributions of vertical (causal relationship) and

horizontal (shared genes and biological pathway) pleiotropy. Specifically, we asked whether the genetic variants associated with LTL are also implicated in CVD risk, and if so, through which pleiotropic mechanisms. This question is central to understanding the biological pathways that link aging (as reflected by telomere length) with cardiovascular outcomes. We have revised the Introduction to more clearly articulate this research objective.

Reference:

Gong W, Guo P, Li Y, Liu L, Yan R, Liu S, Wang S, Xue F, Zhou X, Yuan Z. Role of the Gut-Brain Axis in the Shared Genetic Etiology Between Gastrointestinal Tract Diseases and Psychiatric Disorders: A Genome-Wide Pleiotropic Analysis. *JAMA Psychiatry*. 2023 Apr 1;80(4):360-370. doi: 10.1001/jamapsychiatry.2022.4974

10. In line 136, the authors mention: "Therefore, in light of the widespread associations reported in epidemiological studies and the existing evidence of a shared genetic basis between LTL and CVDs, the necessity for a comprehensive, large-scale analysis is clear." large scale analysis on what topic precisely? Please explicitly state what large-scale analysis you are seeking to perform and to prove what hypothesis?

Re: We thank the reviewer for this important comment. We have clarified that the "comprehensive, large-scale analysis" refers to a genome-wide pleiotropy analysis aimed at systematically investigating the shared genetic architecture between LTL and multiple CVDs. This analysis leveraged the most comprehensive and up-to-date GWAS summary statistics available for LTL and six CVD phenotypes, all from individuals of European ancestry. The central hypothesis we aimed to test is that LTL and CVDs share common genetic architectures, mediated through both vertical (causal relationship) and horizontal (shared genes and biological pathway) pleiotropy. (From Page 6, line 133 to line 136)

11. The six major CVDs mentioned in line 153, should be explicitly listed in brackets.

Re: Thank you very much for pointing this out. To improve clarity, we have revised the sentence in line 153 to explicitly list the six major CVDs in brackets. The revised sentence now reads: ‘Following the harmonization and filtering of SNPs shared across GWAS summary statistics, we employed cross-trait linkage disequilibrium (LD) score regression (LDSC) to assess genome-wide genetic correlation (r_g) between LTL and six major CVDs, namely AF, CAD, venous thromboembolism (VTE), HF, PAD, and stroke.’ (From Page 6, line 149 to line 152)

12. The abbreviation VTE in line 168 is not previously defined. Also PAD in line 155.

Re: We appreciate the reviewer pointing this out. We have now defined both abbreviations—PAD (peripheral artery disease) and VTE (venous thromboembolism)—at their first mention in the manuscript to ensure clarity for all readers.

13. Line 161-164 states: “Although genome-wide r_g offered valuable insight into the genetic overlap between phenotypes, it could not distinguish genetic overlap resulting from a mixture of concordant and discordant effects from the absence of genetic overlap, potentially yielding an estimated r_g near zero in both scenarios.” which two scenarios the authors are referring to?

Re: Thank you for your insightful comment. The two scenarios referred to in lines 161–164 are: 1) A genuine absence of genetic overlap between the phenotypes; 2) The presence of concordant and discordant genetic effects at different genomic loci, which offset each other and result in an overall genome-wide genetic correlation (r_g) close to zero. Although both scenarios can yield similarly low r_g values, they imply fundamentally different biological mechanisms (Frei *et al.*, 2019; Hindley *et al.*, 2022; Hindley *et al.*, 2021; O’Connell *et al.*, 2021; Røddevand *et al.*, 2021; Werme *et al.*, 2022).

To address this issue, we supplemented the LDSC analysis with MiXeR and LAVA. MiXeR quantifies the number of shared causal variants irrespective of their effect direction. For example, despite a low genome-wide r_g between LTL and AF, MiXeR estimated that 18.30%

of LTL-associated variants also influence AF (Dice coefficient = 0.157, SD = 0.041), suggesting polygenic overlap despite the low r_g . This is likely due to heterogeneous effect directions. Additionally, LAVA's local genetic correlation analyses further supported this pattern (9 positively correlated and 4 negatively correlated regions), revealing mixed effect directions. These findings highlight the limitations of relying solely on r_g as a metric and emphasize the importance of integrating multiple complementary methods, such as MiXeR and LAVA, to better understand the complexity of genetic architecture.

Reference:

Frei O, Holland D, Smeland OB, et al. Bivariate causal mixture model quantifies polygenic overlap between complex traits beyond genetic correlation. *Nat Commun.* 2019;10(1):2417. Published 2019 Jun 3. doi:10.1038/s41467-019-10310-0

Hindley G, Frei O, Shadrin AA, et al. Charting the Landscape of Genetic Overlap Between Mental Disorders and Related Traits Beyond Genetic Correlation. *Am J Psychiatry.* 2022;179(11):833-843. doi:10.1176/appi.ajp.21101051

Hindley G, Bahrami S, Steen NE, et al. Characterising the shared genetic determinants of bipolar disorder, schizophrenia and risk-taking. *Transl Psychiatry.* 2021;11(1):466. Published 2021 Sep 8. doi:10.1038/s41398-021-01576-4

O'Connell KS, Frei O, Bahrami S, et al. Characterizing the Genetic Overlap Between Psychiatric Disorders and Sleep-Related Phenotypes. *Biol Psychiatry.* 2021;90(9):621-631. doi:10.1016/j.biopsych.2021.07.007

Rødevand L, Bahrami S, Frei O, et al. Extensive bidirectional genetic overlap between bipolar disorder and cardiovascular disease phenotypes. *Transl Psychiatry.* 2021;11(1):407. Published 2021 Jul 23. doi:10.1038/s41398-021-01527-z

Werme J, van der Sluis S, Posthuma D, de Leeuw CA. An integrated framework for local genetic correlation analysis. *Nat Genet.* 2022;54(3):274-282. doi:10.1038/s41588-022-01017-y

14. In line 170, the authors mentioned “These variants are referred to as ‘causal’ variants”. It is not clear what “these” refer to. Also a citation might be necessary here for any criteria used to refer to a variant a “causal variant”.

Re: We thank the reviewer for this valuable observation. We agree that the phrase “these variants” was ambiguous. To clarify, we have revised the manuscript to explicitly define “causal variants” as genetic variants with non-zero additive effects on the trait (From Page 7, line 169 to line 170). Specifically, a variant is considered “causal” under the additive model if it contributes to heritable variation via a non-zero effect size (Frei *et al.*, 2019; Holland *et al.*, 2020). We have now introduced this definition earlier in the text and added relevant citations to avoid any ambiguity. Additionally, we clarify that in our study, “causal variants” refer to the total number of such variants estimated by MiXeR, which does not aim to localize individual variants, but rather models polygenicity by estimating the number of variants contributing to genetic architecture across the genome. This modeling framework is biologically more realistic than assuming a sparse distribution of large-effect variants and allows for the estimation of shared polygenic architecture across traits.

Reference:

Frei O, Holland D, Smeland OB, et al. Bivariate causal mixture model quantifies polygenic overlap between complex traits beyond genetic correlation. *Nat Commun.* 2019;10(1):2417. Published 2019 Jun 3. doi:10.1038/s41467-019-10310-0

Holland D, Frei O, Desikan R, et al. Beyond SNP heritability: Polygenicity and discoverability of phenotypes estimated with a univariate Gaussian mixture model. *PLoS Genet.* 2020;16(5):e1008612. Published 2020 May 19. doi:10.1371/journal.pgen.1008612

15. In line 172, the authors mention “This approach provides a clearer picture of localized correlations, which might otherwise cancel each other out at a genome-wide level.”. What do the author mean by “cancel each other out”. The authors should be explicit about the phrase “cancel each other out”.

Re: Thank you very much for your insightful comment. We agree that the phrase “cancel each other out” requires clarification. Here, it refers to genetic variants having opposite directional effects on two traits—meaning some variants increase risk for one trait while decreasing risk for another. When these discordant effects are widespread, they offset the concordant effects at the genome-wide level, resulting in an overall genetic correlation close to zero despite substantial shared genetic architecture. This phenomenon can mask important localized genetic correlations. We have revised the manuscript sentence accordingly for clarity: “This approach provides a clearer picture of localized correlations, which might otherwise be masked in genome-wide analyses due to opposing directional effects of shared variants that cancel each other out.” (From Page 7, line 172 to line 174)

16. In line 173, the authors should state the input and output used in the MiXeR analysis.

Re: Thank you for your valuable comment. The MiXeR analysis uses raw GWAS summary statistics from two traits as input, including SNP-level z-scores (or effect sizes with standard errors), allele information, and a set of quality-controlled SNPs. Additionally, an external linkage disequilibrium (LD) reference panel, typically the 1000 Genomes Project Phase 3, is used to model SNP correlations. The outputs from MiXeR include: (1) Estimates of polygenicity for each trait (number of SNPs with non-zero genetic effects); (2) Variance of effect sizes; (3) Estimates of the number of shared and trait-specific causal variants; (4) Visualizations such as Venn diagrams illustrating genetic overlap; (5) Model fit statistics (e.g., log-likelihood) indicating model performance.

17. Explain calculation of the Dice Coefficient under the methods section.

Re: Thank you for this valuable suggestion. We have now revised the Methods section to clarify the calculation of the Dice Coefficient within the MiXeR framework. Specifically, the Dice Coefficient quantifies the proportion of shared causal variants between two traits relative to their total number of causal variants, and is calculated as:

$$\text{Dice Coefficient} = \frac{2 \times N_{\text{shared}}}{N_{\text{trait1}} + N_{\text{trait2}}},$$

where: N_{shared} is the number of causal variants inferred to affect both traits, N_{trait1} and N_{trait2} are the total numbers of causal variants inferred for each trait individually. This metric captures the degree of polygenic overlap regardless of effect direction, providing complementary insight to genetic correlation estimates. We have added this definition to the revised manuscript to improve clarity and methodological transparency. (From Page 28, line 791 to line 793)

18. In line 181, the authors should present the results that revealed distinct patterns of polygenic overlap. What were these patterns of polygenic overlap and how was it concluded that they were distinct.

Re: We thank the reviewer for this insightful comment. In our study, distinct polygenic overlap patterns between LTL and various CVDs was quantified based on multiple aspects: polygenicity levels, the number and proportion of shared versus unique "causal" variants, Dice coefficients, and the concordance of effect directions. Specifically, our results revealed the following patterns:

1) Low polygenicity and symmetric overlap (e.g., LTL–AF, LTL–PAD, LTL–VTE): LTL exhibited low polygenicity (380 variants), and similarly low polygenicity was observed for AF, PAD, and VTE (ranging from 308 to 504). Due to this, the absolute number of shared variants with these traits was small (e.g., 69 for AF, 86 for PAD), corresponding to a lower proportion of shared variants (e.g., 18.30% for AF, 22.63% for PAD). However, the Dice coefficients suggest relatively large genetic overlaps between LTL and these CVDs, particularly between LTL and PAD (Dice coefficient = 0.229, SD = 0.015).

2) High polygenicity and asymmetric overlap (e.g., LTL–HF, LTL–CAD, LTL–Stroke): HF, CAD, and Stroke were more polygenic (up to 2,305 variants for HF). They shared more variants with LTL in absolute number (e.g., 185 for HF), and a large proportion of

LTL-influencing variants overlapped with these traits (e.g., 48.69% of LTL-associated variants also influence HF). However, due to the high polygenicity of the CVD traits, these shared variants represented only a small proportion of their own trait-influencing variants (e.g., only 8.02% for HF). This asymmetric overlap highlights a distinct sharing pattern.

3) Overlap with mixed directionality (especially evident in AF): For LTL-AF, even with a detectable proportion of overlap (69 shared variants, 18.30% of LTL-influencing variants, 13.79% of AF-influencing variants), the weak genome-wide genetic correlation and the low effect-direction concordance (41.65%) strongly suggest predominant mixed directional effects. This pattern, of substantial overlap despite weak r_g , implies that genome-wide correlation underestimates the extent of overlap when effect directions cancel each other out.

In summary, we identified three distinct polygenic overlap patterns: (1) symmetric overlap under low polygenicity, (2) asymmetric overlap driven by trait-specific polygenicity differences, and (3) substantial overlap masked by mixed effect directions. These consistent patterns across trait pairs not only underscore the heterogeneity in genetic architecture among CVD traits but also highlight the limitations of traditional correlation metrics in capturing the full extent of polygenic sharing.

19. Sometimes the authors use one and sometimes 2 decimal points within the manuscript.

Re: Thank you for pointing this out. We apologize for the inconsistency in decimal formatting. We have now carefully reviewed the manuscript and ensured that all numerical values follow a consistent format, using two decimal places as appropriate based on the context and standard conventions in our field. The revised version reflects these corrections.

20. In line 382, the authors mentioned we successfully replicated 1,117 genes (60.57%) using FUMA eQTL mapping. What is the outcome variable for these replications? Did the author perform any meta-analysis? Was there any overlap between the original cohort where these genes were found and the UKbiobank?

Re: We appreciate the reviewer’s thoughtful questions. In this analysis, the 1,117 “replicated” genes refer to those initially prioritized via e-MAGMA based on PLACO-identified loci, and subsequently supported by significant cis-eQTL associations using FUMA’s eQTL mapping module. The outcome variable for replication was whether each gene showed statistically significant cis-eQTL support in independent eQTL resources, such as GTEx v8, as implemented in FUMA. No meta-analysis was performed during this replication step, as the goal was not to aggregate results but rather to evaluate reproducibility across complementary functional annotations. Regarding sample overlap, although part of the GWAS summary statistics used for e-MAGMA (e.g., LTL) were derived from the UK Biobank, the eQTL data used in FUMA are obtained from independent datasets (i.e., GTEx), which do not overlap with UK Biobank samples, ensuring no direct sample overlap between discovery and replication stages. We have now revised the Methods section to clarify these aspects in the manuscript. (From Page 32, line 915)

21. In line 289, the authors say: “289 pleiotropic genes (207 unique) were jointly identified by MAGMA and e-MAGMA analysis” in what sense these genes were unique? If you mean that they were only found in association with one trait, then pleiotropy doesn’t make sense.

Re: We thank the reviewer for this important comment. In this context, the 207 “unique” genes refer to a non-redundant count of genes across all LTL–CVD trait pairs. Specifically, a total of 289 gene-trait associations were identified by MAGMA and e-MAGMA, but some genes were implicated in multiple trait pairs. After removing duplicates, 207 distinct genes remained. Importantly, many of these 207 genes were associated with more than one LTL–CVD pair, indicating pleiotropic effects. Thus, the term “unique” refers only to gene identity (non-redundancy), not to association with a single trait.

Reviewer #2 (Remarks to the Author):

Most of my comments have been addressed. Particularly, they have 1) acknowledged potential bias in MR due to sample overlaps as a limitation in the discussion section (and used newer methods to account for these bias), 2) performed coloc with glucose related traits to highlight a ATXN2 variant as a shared genetic locus for LTL, CVD and glyceimic traits (this will be a validation of previous work by ENGAGE consortium), 3) added in the replications using East-Asian data in supplementary note and 4) showed that if they focused on overlapping genes from Magma and e-magma gene based evaluations, they will have a smaller set of 50 genes that explain much of the pleiotropic effects between the LTL-CVD trait pairs. And, they have modified the conclusions slightly, stating that there is some level of causal effects and vertical pleiotropy between LTL-CVD traits (mainly CAD), however much of these associations are due to horizontal pleiotropic effects.

Re: We sincerely thank the reviewer for their thoughtful comments and for acknowledging the key revisions made in our manuscript. We are glad that our updates—particularly those related to MR bias correction, locus colocalization, East Asian replication, and pleiotropic gene prioritization—addressed the concerns raised. We also appreciate the recognition of our revised conclusions, which now more clearly distinguish between causal and pleiotropic relationships between LTL and CVDs.

Reviewer #3 (Remarks to the Author):

Authors made several adjustments, improved clarity and performed additional analyses. This improves the paper a lot. However, due to a broad and quite liberal approach that includes all possible connections despite the weak signals the strength of the evidence linking these traits is still a bit lost to me. The authors have kindly added a figure (Supplementary Figure 1) that somewhat details the research questions and methods applied for each this is helpful and helps to improve the readers understanding of the authors strategy. However, the results are hard to follow and, while presented in detail (Figures and Supplementary tables), I believe that the paper lacks clarity in a final summary of findings that would create the authors stated “atlas of the shared genetic associations”.

1. One suggestion for improvement is a figure or a table that summarises the key findings making clear connections between the results, i.e. the most important genetic variants/genes alongside the methods they were detected and confirmed with.

Re: Thank you for your insightful suggestion. We will include a new figure in the revised manuscript that provides a concise overview of the shared genetic variants/genes identified, along with the methods employed for their detection and validation (Supplementary Fig. 1).

2. The reference you give in Supp Table 1 for the LTL GWAS was run on the whole of UKB where multiple ethnicities were included. Therefore you have not “selected exclusively the European subset of the LTL and CVD GWAS data”. Please correct this in the text.

Re: We thank the reviewer for this important and constructive comment. You are correct that the referenced LTL GWAS (Supplementary Table 1) was conducted using the full UK Biobank cohort, which includes individuals from multiple ancestral backgrounds. However, participants of European ancestry constitute approximately 95% of the cohort, and thus, the resulting summary statistics predominantly reflect European ancestry. Nonetheless, we acknowledge that the original phrasing may have overstated the exclusivity of the European

subset. To address this, we have revised the wording in the Methods section and Supplementary Table 1 to indicate that the LTL GWAS was based on the full UK Biobank cohort, with results largely reflecting European ancestry. We have also reviewed the manuscript to ensure consistency in the description of ancestry across all relevant sections. (From Page 25, line 701-702)

3. Your colocalisation at *ATXN2* justifies the previous comment that this locus is involved in other mechanisms and is therefore pleiotropic. Perhaps this should be made clearer in the text as a limitation, that the *ATXN2* loci (and similar) that are linked to multiple CVDs, blood pressure, and lipids, may have larger pleiotropic relationships that are not covered in this study.

Re: We sincerely thank the reviewer for this important and insightful comment. We agree that loci such as *ATXN2*, which colocalise with multiple cardiovascular traits—including blood pressure, lipid, and glycemic traits—likely reflect broader and more complex pleiotropic mechanisms that go beyond the specific LTL–CVD relationships explored in our study. We acknowledge this as a limitation of our current approach, which focuses primarily on shared genetic architecture between LTL and CVDs. To address this, we have now explicitly incorporated this point into the Discussion section, noting that certain loci with extensive cross-trait colocalisation, such as *ATXN2*, may influence CVD risk via multiple distinct biological pathways. Future studies integrating additional cardiometabolic traits may be necessary to fully disentangle these effects. (From Page 24, line 669-675)

4. I believe you are overstating the impact of sample overlap, especially when there is good evidence that this does not introduce bias in two-sample MR methods (except for MR Egger) when using Biobank sized GWAS (Minneli et al. PMID: 33899104).

Re: Thank you for your thoughtful comment. We appreciate your reference to the work by Minneli *et al.*, which provides strong evidence that sample overlap does not introduce significant bias in most two-sample MR methods, except for MR Egger, when using large-scale datasets such as those from the UK Biobank.

Therefore, We re-performed the two-sample MR analysis independently using standard parameters to provide a more reliable causal inference. In this updated analysis, we identified 144 independent genome-wide significant SNPs ($P < 5 \times 10^{-8}$) associated with LTL to serve as instrumental variables for subsequent causal inference. We found nominally significant associations ($P < 0.05$) between genetically determined LTL and risk of CAD or PAD. That is, genetically inferred LTL shortening was linked to a higher risk of CAD (OR = 0.942, 95% CI: 0.896–0.990) and PAD (OR = 0.857, 95% CI: 0.745–0.986). To assess the robustness of these findings, we conducted a sensitivity analysis using an alternative set of 130 independent genome-wide significant SNPs for LTL, as defined in the original GWAS of LTL. The results only support a causal relationship between genetically predicted LTL shortening and risk of PAD (OR = 0.806, 95% CI: 0.682–0.952) at a nominally significant level ($P < 0.05$). The lack of concordance in these results may reflect either the presence of sample overlap or insufficient statistical power in the two-sample MR analyses.

Reference:

Minelli C, Del Greco M F, van der Plaat DA, Bowden J, Sheehan NA, Thompson J. The use of two-sample methods for Mendelian randomization analyses on single large datasets. *Int J Epidemiol.* 2021;50(5):1651-1659. doi:10.1093/ije/dyab084

5. Supplementary Table 6 shows your MR analysis with LTL as the exposure uses an instrument with only 51 SNPs associated with LTL. The referenced paper these data come from generated an exact conditionally independent instrument that excluded pleiotropic variants and contained 130 variants. Why is this instrument not being used?

Re: Thank you for your valuable comments. The previous two-sample MR analysis was originally derived as a sensitivity analysis during the LHC-MR calculation process, using only 51 SNPs associated with LTL as genetic instruments. However, because LHC-MR results are less robust for causal inference when there is sample overlap between traits, we remove all LHC-MR-related analyses from the current version of our manuscript.

We re-performed the two-sample MR analysis independently using standard parameters to provide a more reliable causal inference. In this updated analysis, we identified 144 independent genome-wide significant SNPs ($P < 5 \times 10^{-8}$) associated with LTL to serve as instrumental variables for subsequent causal inference. We found nominally significant associations ($P < 0.05$) between genetically determined LTL and risk of CAD or PAD. That is, genetically inferred LTL shortening was linked to a higher risk of CAD (OR = 0.942, 95% CI: 0.896–0.990) and PAD (OR = 0.857, 95% CI: 0.745–0.986). To assess the robustness of these findings, we conducted a sensitivity analysis using an alternative set of 130 independent genome-wide significant SNPs for LTL, as defined in the original GWAS of LTL. The results only support a causal relationship between genetically predicted LTL shortening and risk of PAD (OR = 0.806, 95% CI: 0.682–0.952) at a nominally significant level ($P < 0.05$). The lack of concordance in these results may reflect either the presence of sample overlap or insufficient statistical power in the two-sample MR analyses.

6. I appreciate the argument defending the use of PLACO despite weak evidence being based on a published strategy. I would still comment, without requesting a response, that just because it wasn't picked up for a previous paper it doesn't make the strategy correct. I thank the authors for their detailed response to my previous comment (R3.10), it was very helpful.

Re: Thank you very much for your thoughtful feedback. We sincerely appreciate your constructive engagement with our work throughout the review process.

Reviewer #4 (Remarks to the Author):

Re: We are grateful for the thoughtful and constructive feedback provided by both reviewers.

Re: NCOMMS-24-58910C

Title: Contribution of leukocyte telomere length to cardiovascular disease onset from genome-wide cross-trait analysis

We thank the Reviewers and the Editor for the helpful feedback and insightful suggestions. The response from the reviewers is encouraging and constructive. Below, we provide point-by-point responses that address all the comments with additional analyses. By incorporating these new data and findings, our manuscript has been improved substantially.

REVIEWERS' COMMENTS

We thank the authors for their considered responses and believe the paper has greatly improved. We have a small number of additional minor comments.

1. -In response to R3.4 the authors re-performed some MR analysis to show the effects of overlap bias, and in lines 513-527 make a point about sample overlap, however, most of the previous papers they used to source genetic instruments were based on large datasets suggesting sample overlap bias is highly unlikely and thus this point has little value.

Re: Thank you for your insightful comment. We have removed lines 513-527 regarding sample overlap as requested.

2. -Lines 144-146 Previous MR studies do not contain the reference to the key paper by Codd et al. 2021 (your ref 15, from where the instrument was taken, where several CVDs have been assessed). It should also add a reference to Li et al. (PMID: 32109421); they use an instrument from an entirely independent meta-GWAS (also showing several CVD MR estimates).

Re: Thank you for your suggestion. We have now included the reference to Codd et al. 2021 (Ref. 15) and added the reference to Li et al. (PMID: 32109421) as recommended (Codd *et al.*, 2021; Li *et al.*, 2020).

Reference:

Codd V, Wang Q, Allara E, et al. Polygenic basis and biomedical consequences of telomere length variation. *Nat Genet.* 2021;53(10):1425-1433. doi:10.1038/s41588-021-00944-6

Li C, Stoma S, Lotta LA, et al. Genome-wide Association Analysis in Humans Links Nucleotide Metabolism to Leukocyte Telomere Length. *Am J Hum Genet.* 2020;106(3):389-404. doi:10.1016/j.ajhg.2020.02.006

3. -Line 531 suggests that all previous research on TL and CVDs is overstated - perhaps this should be supported with additional references.

Re: Thank you for your valuable feedback. We have now included further references to support our assessment of the limitations in previous studies on LTL and CVDs (Larsson *et al.*, 2023; Kang *et al.*, 2025; Hartwig *et al.*, 2021).

Reference:

Larsson SC, Butterworth AS, Burgess S. Mendelian randomization for cardiovascular diseases: principles and applications. *Eur Heart J.* 2023;44(47):4913-4924. doi:10.1093/eurheartj/ehad736

Kang H, Guo Z, Liu Z, Small D. Identification and Inference with Invalid Instruments. *Annu Rev Stat Appl.* 2025;12:385-405. doi:10.1146/annurev-statistics-112723-034721

Hartwig FP, Tilling K, Davey Smith G, Lawlor DA, Borges MC. Bias in two-sample Mendelian randomization when using heritable covariable-adjusted summary associations. *Int J Epidemiol.* 2021;50(5):1639-1650. doi:10.1093/ije/dyaa266

4. -The authors have conducted HyPrColoc but we are unable to find the results, which should be added to Supplementary Data and referenced in the paper where appropriate.

Re: Thank you for pointing this out. We apologize for the oversight. The results of the HyPrColoc analysis have now been added to the Supplementary Data 19, and we have included the appropriate references to them in the main text.